# CERTIFIABLY BYZANTINE-ROBUST FEDERATED CONFORMAL PREDICTION

## ABSTRACT

Conformal prediction has shown impressive capacity in constructing statistically rigorous prediction sets for machine learning models with exchangeable data samples. The siloed datasets, coupled with the escalating privacy concerns related to local data sharing, have inspired recent innovations extending conformal prediction into federated environments with distributed data samples. However, this framework for distributed uncertainty quantification is susceptible to Byzantine failures. A minor subset of malicious clients can significantly compromise the practicality of coverage guarantees. To address this vulnerability, we introduce a new algorithm Rob-FCP to execute robust federated conformal prediction, effectively countering malicious clients capable of reporting arbitrary statistics with the conformal calibration process. We theoretically provide the conformal coverage bound of Rob-FCP in the Byzantine setting and show that the coverage of Rob-FCP is asymptotically close to the desired coverage level under mild conditions in both IID and non-IID settings. We also propose a malicious client number estimator to tackle a more challenging setting where the number of malicious clients is unknown to the defender and theoretically shows its effectiveness. We empirically demonstrate the robustness of Rob-FCP against diverse proportions of malicious clients under a variety of Byzantine attacks on five realistic benchmark and real-world healthcare datasets.

## 1 INTRODUCTION

As deep neural networks (DNNs) achieved great success across multiple fields (He et al., 2016; Vaswani et al., 2017; Li et al., 2022b), quantifying the uncertainty of model predictions has become essential, especially in safety-conscious domains such as healthcare and medicine (Ahmad et al., 2018; Erickson et al., 2017; Kompa et al., 2021). For example, in sleep medicine, accurately classifying sleep stages (typically on EEG recordings) is crucial for understanding sleep disorders. Just like a human specialist who oftentimes offers a set of possible interpretations for one recording, a DNN should not only provide the point prediction but preferably a *prediction set* (of possible sleep stages), whose cardinality conveys the level of uncertainty in a natural way. In constructing such prediction sets, we often seek frequentist coverage guarantees: The prediction set should contain the truth with a pre-specified probability (e.g. 90%). Recently, Conformal prediction (Shafer & Vovk, 2008; Balasubramanian et al., 2014; Romano et al., 2020) demonstrates the capacity to provide statistical guarantees for any black-box DNN with exchangeable data.

Meanwhile, the demand for training machine learning models on large-scale and diverse datasets necessitates model training across multiple sites and institutions. Federated learning (Konečný et al., 2016; Smith et al., 2017; McMahan et al., 2017; Bonawitz et al., 2019; Yang et al., 2019; Kairouz et al., 2021) offers an effective approach to collaboratively train a global model while preserving data privacy, as it enables training with distributed data samples without the requirement of sharing the raw data. For example, multiple hospitals ("clients") could jointly train a global clinical risk prediction model without sharing raw patient data. However, this introduces several unique challenges: Heterogeneous clients might not satisfy the exchangeability assumption, and the existence of malicious or negligent clients could negatively affect the quality of the prediction sets.

Recently, federated conformal prediction (FCP) methods (Lu & Kalpathy-Cramer, 2021; Lu et al., 2023; Plassier et al., 2023; Humbert et al., 2023) provide rigorous bounds on the coverage rate with distributed and not globally exchangeable data samples. However, FCP demonstrates vulnerability

to Byzantine failures (Lamport et al., 2019), which are caused by uncontrollable behaviors of malicious clients. For example, a hospital's data could be corrupted with incorrect or even fabricated medical information due to human negligence or deliberate manipulation of data statistics (such as age, gender, or disease prevalence). In the Byzantine federated setting, the prediction coverage guarantees of FCP are broken, and the empirical marginal coverage is downgraded severely with a small portion of malicious clients as Figure 1.

In this paper, we aim to restore the coverage rate compromised by malicious clients in Byzantine federated learning settings using a robust federated conformal prediction algorithm, Rob-FCP. The Rob-FCP algorithm calculates conformity scores, sketches them with characterization vectors, and identifies malicious clients based on averaged vector distance. Clients with high maliciousness scores are excluded during calibration. We propose a method to estimate the number of malicious clients when unknown, by maximizing the likelihood of characterization vectors. The coverage bounds show that the coverage of Rob-FCP is asymptotically close to the desired coverage level as long as the number of malicious clients is less than that of be-

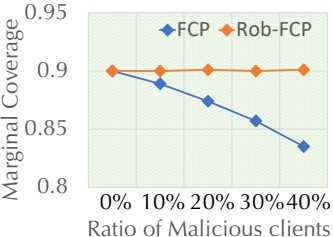

Figure 1: Coverage with different ratios of malicious clients on SHHS dataset. The desired coverage is 0.9.

nign clients and the sample sizes of benign clients are sufficiently large. We also derive the coverage bounds in the non-IID setting and show that the coverage of Rob-FCP can be arbitrarily close to the desired coverage level as long as the heterogeneity among benign clients is bounded.

We empirically evaluate Rob-FCP against multiple Byzantine attackers in both IID and non-IID settings. Rob-FCP outperforms FCP by a large margin and achieves comparable prediction coverage and efficiency as the benign settings on *five* realistic datasets covering multiple fields. We demonstrate the validity and tightness of the bounds of prediction coverage with different ratios of malicious clients and conduct a set of ablation studies.

**Technical Contributions:** Our contributions span both theoretical and empirical aspects.

- We provide the first certifiably robust federated conformal prediction framework (Rob-FCP) in the Byzantine setting where malicious clients can report arbitrary conformity score statistics.
- We propose a maliciousness score to identify Byzantine clients in the space of characterization vectors and a malicious client number estimator to predict the number of malicious clients.
- We theoretically provide the coverage guarantees of Rob-FCP in both IID and non-IID settings. We also theoretically analyze the accuracy of malicious client number estimator.
- We empirically demonstrate the robustness of Rob-FCP in federated Byzantine settings across multiple datasets including two real-world medical datasets and validate the coverage guarantees.

## 2 PRELIMINARIES

### 2.1 CONFORMAL PREDICTION

Suppose that we have $n$ data samples $\{(X_i, Y_i)\}_{i=1}^n$ with features $X_i \in \mathbb{R}^d$ and labels $Y_i \in \mathcal{Y} := \{1, 2, ..., C\}$. Assume that the data samples are drawn IID (thereby exchangeably) from some unknown distribution $\mathcal{P}_{XY}$. Given a desired coverage $1 - \alpha \in (0, 1)$, conformal prediction methods construct a prediction set $\hat{C}_{n,\alpha} \subseteq \mathcal{Y}$ for a new data sample $(X_{n+1}, Y_{n+1}) \sim \mathcal{P}_{XY}$ with the guarantee of *marginal prediction coverage*: $\mathbb{P}[Y_{n+1} \in \hat{C}_{n,\alpha}(X_{n+1})] \geq 1 - \alpha$.

In this work, we focus on the split conformal prediction setting (Papadopoulos et al., 2002), where the data samples are randomly partitioned into two disjoint sets: a training set $\mathcal{I}_{tr}$ and a calibration (hold-out) set $\mathcal{I}_{cal} = [n] \backslash \mathcal{I}_{tr}$. [1] We fit a classifier $h(x) : \mathbb{R}^d \mapsto \mathcal{Y}$ to the training set $\mathcal{I}_{tr}$ to estimate the conditional class probability $\pi : \mathbb{R}^d \mapsto \Delta^C$ with the $y$-th element denoted as $\pi_y(x) = \mathbb{P}[Y = y | X = x]$. Using the estimated probabilities that we denote by $\hat{\pi}(x)$, we then compute a non-conformity score $S_{\hat{\pi}}(X_i, Y_i)$ for each sample in the calibration set $\mathcal{I}_{cal}$. The non-conformity score measures how much non-conformity each sample has with respect to its ground truth label. A small non-conformity score $S_{\hat{\pi}}(X_i, Y_i)$ indicates that the estimated class probability $\hat{\pi}(X_i)$ aligns well with the ground truth label $Y_i$ for the data sample $(X_i, Y_i)$. A simple and commonly used non-conformity score (Sadinle et al., 2019) is: $S_{\hat{\pi}}(x, y) = 1 - \hat{\pi}_y(x)$.

---

[1]In here and what follows, $[n] := \{1, \cdots, n\}$.

Given a desired coverage $1 - \alpha$, the prediction set of a new data point $X_{n+1}$ is formulated as:

$$\hat{C}_{n,\alpha}(X_{n+1}) = \{y \in \mathcal{Y} : S_{\hat{\pi}}(X_{n+1}, y) \leq Q_{1-\alpha}\left(\{S_{\hat{\pi}}(X_i, Y_i)\}_{i \in \mathcal{I}_{\text{cal}}}\right)\}, \tag{1}$$

where $Q_{1-\alpha}(\{S_{\hat{\pi}}(X_i, Y_i)\}_{i \in \mathcal{I}_{\text{cal}}})$ is the $\lceil(1 - \alpha)(1 + |\mathcal{I}_{\text{cal}}|)\rceil$-th largest value of the set $\{S_{\hat{\pi}}(X_i, Y_i)\}_{i \in \mathcal{I}_{\text{cal}}}$. The prediction set $\hat{C}_{n,\alpha}(X_{n+1})$ includes all the labels with a smaller non-conformity score than the $(1 - \alpha)$-quantile of scores in the calibration set. Since we assume the data samples are exchangeable, the marginal coverage of the prediction set $\hat{C}_{n,\alpha}(X_{n+1})$ is no less than $1 - \alpha$. We refer to (Vovk et al., 2005) for a more rigorous analysis of the prediction coverage.

## 2.2 FEDERATED CONFORMAL PREDICTION

In federated learning, multiple clients own their private data and collaboratively develop a global model. Let $K$ be the number of clients. We denote the data distribution of the $k$-th client $(k \in [K])$ by $\mathcal{P}^{(k)}$. Let $\{(X_i^{(k)}, Y_i^{(k)})\}_{i \in [n_k]} \sim \mathcal{P}^{(k)}$ be $n_k$ calibration (held-out) samples of the $k$-th client. We denote $(X_{\text{test}}, Y_{\text{test}})$ as the future test point sampled from the global distribution $\mathcal{Q}_{\text{test}} = \mathcal{Q}_\lambda$ for some probability vector $\lambda \in \Delta^K$: $(X_{\text{test}}, Y_{\text{test}}) \overset{IID}{\sim} \mathcal{Q}_\lambda$, $\mathcal{Q}_\lambda = \sum_{k=1}^{K} \lambda_k \mathcal{P}^{(k)}$.

Let $N = \sum_{k=1}^{K} n_k$ be the total sample size and $\hat{q}_\alpha$ be the $\lceil(1 - \alpha)(N + K)\rceil$ largest value in $\{(X_i^{(k)}, Y_i^{(k)})\}_{i \in [n_k], k \in [K]}$. Suppose that $\hat{\pi} : \mathbb{R}^d \mapsto \Delta^C$ is the conditional class probability estimator trained with federated learning algorithms. FCP (Lu et al., 2023) proves that under the assumption of partial exchangeability (Carnap & Jeffrey, 1980) and $\lambda_k \propto (n_k + 1)$, the prediction set $\hat{C}_\alpha(X_{\text{test}}) = \{y \in \mathcal{Y} : S_{\hat{\pi}}(X_{\text{test}}, y) \leq \hat{q}_\alpha\}$ is a valid conformal prediction set with the guarantee:

$$1 - \alpha \leq \mathbb{P}\left[Y_{\text{test}} \in \hat{C}_\alpha(X_{\text{test}})\right] \leq 1 - \alpha + \frac{K}{N + K}. \tag{2}$$

Due to the concerns of privacy and communication costs, it is not practical for all the agents to upload the local non-conformity scores to the server for quantile value computation. Therefore, FCP (Lu et al., 2023) leverages data sketching algorithms such as T-digest (Dunning, 2021) for distributed quantile estimation. They prove that if the rank of quantile estimate $\hat{q}_\alpha$ is between $(1 - \alpha - \epsilon)(N + K)$ and $(1 - \alpha + \epsilon)(N + K)$, then the guarantee in Equation (2) can be corrected as the following:

$$1 - \alpha - \frac{\epsilon N + 1}{N + K} \leq \mathbb{P}\left[Y_{\text{test}} \in \hat{C}_\alpha(X_{\text{test}})\right] \leq 1 - \alpha + \epsilon + \frac{K}{N + K}. \tag{3}$$

## 3 BYZANTINE-ROBUST FEDERATED CONFORMAL PREDICTION

### 3.1 PROBLEM SETUP

We follow the standard setup of FCP illustrated in Section 2.2 and consider the following Byzantine threat model. Suppose that among $K$ clients, there exist $K_b$ benign clients and $K_m$ ($K_m = K - K_b$) malicious clients. Without loss of generality, let the clients indexed by $[K_b] = \{1, ..., K_b\}$ be benign clients and the clients indexed by $[K] \backslash [K_b] = \{K_b + 1, ..., K\}$ be malicious clients. The $k$-th benign clients ($k \in [K_b]$) leverage the collaboratively trained global model $\hat{\pi}$ to compute the conformity scores on its local calibration data set $\{(X_i^{(k)}, Y_i^{(k)})\}_{i \in [n_k]}$ and then report the conformity score statistics to the server. However, $K_m$ malicious clients can submit arbitrary score statistics to the server. For the threat model, we attempt to develop a Byzantine robust FCP framework (Rob-FCP) with which the coverage and prediction efficiency are not affected by malicious clients. We also attempt to provide rigorous coverage guarantees of Rob-FCP in the Byzantine setting.

### 3.2 ROB-FCP ALGORITHM

Rob-FCP first identifies the set of malicious clients, excludes their score statistics, then computes the empirical quantile of conformity scores, and finally performs conformal prediction.

**Characterization of conformity scores.** Let $\{s_j^{(k)}\}_{j \in [n_k]}$ be the conformity scores computed by the $k$-th client ($k \in [K]$). Since it is challenging to detect abnormal behavior from the unstructured and unnormalized score samples, we first need to characterize the conformity scores $\{s_j^{(k)}\}_{j \in [n_k]}$ with a vector $\mathbf{v}^{(k)} \in \mathbb{R}^H$ for client $k$, where $H \in \mathbb{Z}^+$ is the dimension of the vector and implicates the granularity of the characterization. Specifically, we can partition the range of conformity score values $[0, 1]$ into $H$ subintervals $\{[a_h, a_{h+1})\}_{0 \leq h \leq H-2} \cup \{[a_{H-1}, a_H]\}$, where $a_h$ denotes the $h$-th cut point. For simplicity, we abuse the last interval $[a_{H-1}, a_H]$ as $[a_{H-1}, a_H)$ in the future

discussions. The $h$-th element of the characterization vector $(\mathbf{v}_h^{(k)})$ equals the probability that the conformity score lies in the corresponding subinterval $[a_{h-1}, a_h)$ and can be estimated as:

$$\mathbf{v}_h^{(k)} = \mathbb{P}_{s \sim \left\{ s_j^{(k)} \right\}_{j \in [n_k]}} [a_{h-1} \leq s < a_h] = \frac{1}{n_k} \sum_{j=1}^{n_k} \mathbb{I}\left[ a_{h-1} \leq s_j^{(k)} < a_h \right]. \quad (4)$$

The characterization vector $\mathbf{v}^{(k)}$ basically presents the histogram statistics of the score samples and follows an underlying multinomial distribution. The conformity scores of data samples from the same distribution have high similarity, and thus the characterization vectors of benign clients with homogeneous data also have high similarity. Therefore, the characterization vectors of malicious clients can be distinguishable. Rob-FCP also flexibly allows for alternative approaches to characterizing the empirical conformity score samples with a real-valued vector $\mathbf{v}$. Alternatives include kernel density estimation (Terrell & Scott, 1992) as a smooth generalization of histogram statistics, parametric model fitting (e.g., Gaussian model), and exemplar representations by clustering algorithms (e.g., KMeans). We empirically show that the histogram statistic in Equation (4) sufficiently outperforms parametric models and clustering approaches and mainly adopt it in Rob-FCP.

**Benefits of the characterization vector to the federated setting**. The characterization vector $\mathbf{v}^{(k)}$ also benefits the federated setting to reduce the privacy leakage of local scores and reduce communication costs. Concretely, in the federated conformal prediction context, each client $k$ computes the conformity scores of its $n_k$ local data samples $\{s_j^{(k)}\}$. However, sending all the conformity scores $\{s_j^{(k)}\}$ to the server is expensive and leaks much information about local data samples. Therefore, we characterize the scores $\{s_j^{(k)}\}$ with a histogram statistics vector $\mathbf{v}^{(k)}$, which computes score frequency in $H$ equally partitioned subintervals. The characterization vector $\mathbf{v}^{(k)}$ has a much smaller dimension than the universal scores $\{s_j^{(k)}\}$ ($H << n_k$) and thus leaks much less private information during the communication between the server and clients. Thus, the characterization vector distance computation on the server side also obeys the principle of federated learning for privacy-preserving and communication efficiency.

**Maliciousness score computation**. We identify the malicious clients via a maliciousness score in the space of characterization vectors. First, we compute the pairwise $\ell_p$ ($p \in \mathbb{Z}^+$) vector distance:

$$d_{k_1, k_2} = \|\mathbf{v}^{(k_1)} - \mathbf{v}^{(k_2)}\|_p, \ \forall k_1, k_2 \in [K]. \quad (5)$$

Denote $N_{ear}(k, t)$ as the $t$-nearest neighbors of client $k$ (excluding itself), with the distance between two clients $k_1$ and $k_2$ given by Equation (5). We define the maliciousness score $M(\cdot) : [K] \mapsto \mathbb{R}$ of client $k$ ($k \in [K]$) as the averaged distance to the clients in the $K_b - 1$ nearest neighbors, where $K_b$ is the number of benign clients:

$$M(k) = \frac{1}{K_b - 1} \sum_{k' \in N_{ear}(k, K_b-1)} d_{k,k'}. \quad (6)$$

Then, we let the benign set identified by Rob-FCP $\mathcal{B}_{\text{Rob-FCP}}$ be the set of the index of the clients with the lowest $K_b$ maliciousness scores in $\{M(k)\}_{k=1}^K$. Finally, we can perform quantile estimation $\hat{q}_\alpha$ with the statistics of the clients in the benign set $\mathcal{B}_{\text{Rob-FCP}}$ and do conformal prediction with $\hat{q}_\alpha$ on the distributedly trained global model. We provide the overview of Rob-FCP in Figure 4 in Appendix B and detailed pseudocode of the algorithm of malicious clients identification in Algorithm 1 in Appendix G.

In order to downgrade the global conformal prediction performance, malicious clients tend to report conformity score statistics deviating from benign statistics. Accordingly, the characterization vectors of malicious clients are also separable from the cluster of benign characterization vectors. Since the maliciousness scores compute the averaged vector distance to $K_b - 1$ nearest neighbors, the maliciousness scores of malicious clients are larger than those of benign clients when $K_b > K_m$ holds (a general condition in Byzantine analysis (Blanchard et al., 2017)). Therefore, Rob-FCP can effectively exclude the malicious statistics during conformal calibration and robustly generate the conformal prediction set. We rigorously analyze Rob-FCP and provide the coverage bound of Rob-FCP in the existence of a certain ratio of malicious clients in Section 3.3.

### 3.3 COVERAGE GUARANTEE OF ROB-FCP

In this section, we rigorously analyze the coverage bounds of Rob-FCP in the Byzantine setting where there exist $K_m$ malicious clients among $K$ clients. In Theorem 1, we provide the coverage

guarantees in the IID federated setting where each client owns data sampled from the same global distribution. We then adapt to a non-IID federated setting where clients have heterogeneous data with bounded disparity and provide the coverage guarantees of Rob-FCP in Corollary 1. The theoretical results demonstrate that Rob-FCP asymptotically approaches the desirable coverage level with sufficiently large sample sizes in both IID and non-IID settings. We keep the proof sketches and defer the complete proofs to Appendix D.

**Theorem 1** (Coverage guarantees of Rob-FCP in the IID setting)**.** *For $K$ clients including $K_b$ benign clients and $K_m := K - K_b$ malicious clients, each client reports a characterization vector $\mathbf{v}^{(k)} \in \Delta^H$ ($k \in [K]$) and a quantity $n_k \in \mathbb{Z}^+$ ($k \in [K]$) to the server. Suppose that the reported characterization vectors of benign clients are sampled from the same underlying multinomial distribution $\mathcal{D}$, while those of malicious clients can be arbitrary. We bound the concentration of the characterization vectors with the binomial proportion confidence interval Wallis (2013). Let $\epsilon$ be the estimation error of the data sketching by characterization vectors as illustrated in Equation (3). Under the assumption that $K_m < K_b$, the following holds with probability $1 - \beta$:*

$$\mathbb{P}\left[Y_{test} \in \hat{C}_\alpha(X_{test})\right] \geq 1 - \alpha - \frac{\epsilon n_b + 1}{n_b + K_b} - \frac{H\Phi^{-1}(1 - \beta/2HK_b)}{2\sqrt{n_b}}\left(1 + \frac{N_m}{n_b}\frac{2}{1-\tau}\right),$$
$$\mathbb{P}\left[Y_{test} \in \hat{C}_\alpha(X_{test})\right] \leq 1 - \alpha + \epsilon + \frac{K_b}{n_b + K_b} + \frac{H\Phi^{-1}(1 - \beta/2HK_b)}{2\sqrt{n_b}}\left(1 + \frac{N_m}{n_b}\frac{2}{1-\tau}\right), \quad (7)$$

*where $\tau = K_m/K_b$ is the ratio of the number of malicious clients and the number of benign clients, $N_m := \sum_{k \in [K] \setminus [K_b]} n_k$ is the total sample size of malicious clients, $n_b := \min_{k' \in [K_b]} n_{k'}$ is the minimal sample size of benign clients, and $\Phi^{-1}(\cdot)$ denotes the inverse of the cumulative distribution function (CDF) of standard normal distribution.*

*Proof sketch.* We first leverage statistical confidence intervals and union bounds to conduct concentration analysis of the characterization vectors $\mathbf{v}^{(k)}$ for benign clients ($1 \leq k \leq K_b$). Then we consider the maliciousness scores of special points and relax the histogram statistics error of Rob-FCP. We finally translate the error of aggregated statistics to the error of the coverage bounds.

*Remark.* In Equation (7), the lower bound and upper bound of $\mathbb{P}[Y_{\text{test}} \in \hat{C}_\alpha(X_{\text{test}})]$ are asymptotically close to the desired coverage level $1 - \alpha$ as the minimal benign sample size $n_b$ is sufficiently large and estimation error of data sketching $\epsilon$ is sufficiently small. This implies the robustness of Rob-FCP since the guaranteed coverage level is still valid and tight in the Byzantine setting under these mild conditions. Note that we assume $\tau < 1$ ($i.e., K_m < K_b$), which requires that the number of malicious clients is smaller than benign clients. The assumption aligns with the break point of $\lceil K/2 \rceil$ in Byzantine analysis (Blanchard et al., 2017; Yin et al., 2018; Guerraoui et al., 2018). From Equation (7), we can also conclude that a larger ratio of malicious clients $\tau$ and a larger sample size of malicious clients $N_m$ can induce a larger gap between the upper bound and the lower bound, which implies a larger prediction set and higher quantification uncertainty in practice. $\epsilon$ quantifies the approximation error of the quantile computation induced by the data sketch and can be controlled by adjusting the granularity of the partitions $H$. Note that the appearance of $1/n_b$ is indeed due to the artifact of the proof. One can replace $n_b$ with the summation of the smallest $K_b - K_m$ benign sample sizes according to the proof. In practice, if a benign client has an extremely small sample size, then the characterization vector is almost useless for malicious client identification, and the influence on the conformal calibrated quantile is also negligible as FCP performs a weighted calibration based on sample sizes of clients. Therefore, for the extreme case of some benign clients with extremely small sample sizes (e.g., less than 10), it is more meaningful to discard their scores and apply Rob-FCP to the remaining clients, which is more beneficial to malicious client detection and only leads to negligible error for conformal calibration. We also provide results with more advanced concentration bounds DKW inequality (Dvoretzky et al., 1956) in Theorem 5.

Next, we provide the coverage bound of Rob-FCP in the non-IID setting. To achieve it, we need to assume a bounded disparity of benign clients.

**Assumption 3.1** (Bounded distribution disparity of benign clients)**.** *Suppose that the characterization vector $\mathbf{v}^{(k)}$ is sampled from multinomial distribution $\mathcal{D}_k$ with the event probability $\overline{\mathbf{v}}^{(k)}$ for the $k$-th client ($k \in [K]$). We assume bounded distribution disparity among benign clients:*

$$\|\overline{\mathbf{v}}^{(k_1)} - \overline{\mathbf{v}}^{(k_2)}\|_p \leq \sigma, \ \forall k_1, k_2 \in [K_b]. \quad (8)$$

Assumption 3.1 requires that the mean characterization vector of benign clients is not far away from each other with the metric of $\ell_p$ norm distance. The assumption is quite standard, typical,

and not simplified across the literature of non-IID federated learning analysis (Xiang et al., 2020) and Byzantine analysis (Park et al., 2021; Data & Diggavi, 2021). Concretely, Xiang et al.; Park et al.; Data & Diggavi quantify the non-iid degree (i.e., disparity) of local loss or gradients of clients and then provide the convergence guarantee of federated optimization as a function of the disparity quantity. Similarly, we assume a bounded disparity of score statistics quantified with $\sigma$ and will accordingly provide the coverage guarantees of Rob-FCP in the non-IID setting as a function of $\sigma$. The assumption is also essential since, without a quantity to restrict the disparity in local behavior, the local behavior can vary arbitrarily, and we can not control global performance quantitatively.

Then we can provide the statement of the coverage bound of Rob-FCP in the non-IID setting.

**Corollary 1** (Coverage guarantees of Rob-FCP in the non-IID setting). *Under the same definitions and conditions in Theorem 1 and with Assumption 3.1, the following holds with probability $1 - \beta$:*

$$\mathbb{P}\left[Y_{test} \in \hat{C}_\alpha(X_{test})\right] \geq 1 - \alpha - \frac{\epsilon n_b + 1}{n_b + K_b} - \frac{H\Phi^{-1}(1 - \beta/2HK_b)}{2\sqrt{n_b}}\left(1 + \frac{N_m}{n_b}\frac{2}{1-\tau}\right) - \frac{N_m}{n_b}\frac{\sigma}{1-\tau},$$

$$\mathbb{P}\left[Y_{test} \in \hat{C}_\alpha(X_{test})\right] \leq 1 - \alpha + \epsilon + \frac{K_b}{n_b + K_b} + \frac{H\Phi^{-1}(1 - \beta/2HK_b)}{2\sqrt{n_b}}\left(1 + \frac{N_m}{n_b}\frac{2}{1-\tau}\right) + \frac{N_m}{n_b}\frac{\sigma}{1-\tau}.$$

$$(9)$$

*Proof sketch.* The sketch follows the proof of Theorem 1 except that we further consider bounded distribution disparity of benign clients during concentration analysis of the characterization vectors.

*Remark.* The lower and upper bound of coverage in Equation (9) also asymptotically approaches the desired coverage level $1-\alpha$ as the minimal benign sample size $n_b$ is sufficiently large and estimation error of data sketching $\epsilon$ is sufficiently small, which implies the robustness of Rob-FCP in the non-IID setting. Furthermore, a larger distribution disparity $\sigma$ can induce a larger gap between the lower bound and upper bound and thus a larger prediction set with high quantification uncertainty. This shows that high heterogeneity among clients will pose more difficulty in Byzantine-robust federated conformal prediction, but as long as the disparity is bounded, the coverage bound is still asymptotically valid and tight with Rob-FCP. We also provide results with more advanced concentration bounds DKW inequality (Dvoretzky et al., 1956) in Corollary 3.

## 4 ROB-FCP WITH UNKNOWN NUMBERS OF MALICIOUS CLIENTS

### 4.1 MALICIOUS CLIENT NUMBER ESTIMATOR

The number of malicious clients $K_m$ is typically a known quantity for the defender in the standard Byzantine setting (Blanchard et al., 2017; Park et al., 2021; Liu et al., 2023). The number of malicious clients is a critical quantity for the defense. An overestimation of the quantity will involve malicious clients that attempt to deteriorate the global performance, while an underestimation of the quantity will exclude benign clients and induce a global distribution shift with heterogeneous data in the non-IID setting. However, the number of malicious clients is usually agnostic to the server in practice. Therefore, we propose a malicious client number estimator for Rob-FCP to unleash its potential in a more challenging Byzantine setting with unknown numbers of malicious clients.

To estimate the number of malicious clients $K_m$, it is sufficient to estimate the number of benign clients $K_b$ as we assume the total number of clients $K$ is a known quantity. Towards that, we should maximize the likelihood of benign characterization vectors while minimizing the likelihood of malicious characterization vectors over the number of benign clients $\hat{K}_b$. The discrete optimization can be tackled by traversing the finite feasible set, but the challenge lies in the computation of the likelihood, which requires a specified distribution of benign characterization vectors. Since the benign characterization vector $\mathbf{v}^{(k)}$ ($k \in [K_b]$) follows a multinomial distribution and the multinomial distribution can be approximated with multivariate normal distribution with large sample sizes (Severini, 2005), we assume the benign characterization vectors are drawn from a multivariate normal distribution $\mathcal{N}(\mu, \Sigma)$ with mean $\mu \in \mathbb{R}^H$ and covariance $\Sigma \in \mathbb{R}^{H \times H}$. For a given benign client number $\tilde{K}_b$, we can leverage the Rob-FCP algorithm illustrated in Section 3.2 to identify the set of benign clients, which enables different likelihood computations for benign clients and malicious clients individually. Let $I(\cdot) : [K] \mapsto [K]$ be the mapping from the rank of clients sorted by the maliciousness scores to the original index. The estimate of the benign client number $\hat{K}_b$ is given by:

$$\hat{K}_b = \underset{z \in [K]}{\arg\max}\left[\frac{1}{z}\sum_{k=1}^{z}\log p(\mathbf{v}^{(I(k))}; \hat{\mu}(z), \hat{\Sigma}(z)) - \frac{1}{K-z}\sum_{k=z+1}^{K}\log p(\mathbf{v}^{(I(k))}; \hat{\mu}(z), \hat{\Sigma}(z))\right], \quad (10)$$

where $\hat{\mu}(z) = \frac{1}{z} \sum_{k \in [z]} \mathbf{v}^{(I(k))}$, $\hat{\Sigma} = \mathbb{E}_{k \in [z]} \left[ (\mathbf{v}^{(I(k))} - \hat{\mu}(z))^T (\mathbf{v}^{(I(k))} - \hat{\mu}(z)) \right]$ are the empirical mean and covariance, and $p(\mathbf{v}; \mu, \Sigma)$ computes the likelihood of multivariate normal distribution as $p(\mathbf{v}; \mu, \Sigma) = \exp\left(-1/2(\mathbf{v} - \mu)^T \Sigma^{-1} (\mathbf{v} - \mu)\right) / \sqrt{(2\pi)^H |\Sigma|}$. The discrete optimization in Equation (10) essentially searches for $\hat{K}_b$ such that the vectors of $\hat{K}_b$ clients with the lowest maliciousness scores (i.e., more likely to be benign clients) have a high likelihood to conform the normal distribution, while the vectors of remaining clients (i.e., more likely to be malicious clients) have a low likelihood to conform the normal distribution. Note that we can use the estimated $\hat{K}_b$ as the input parameter $\tilde{K}_b$ in Rob-FCP in the next step and repeat the optimization iteratively.

## 4.2 ANALYSIS OF MALICIOUS CLIENT NUMBER ESTIMATOR

In this part, we theoretically show the precision of benign client number estimate in Equation (10).

**Theorem 2** (Probability of correct malicious client number estimate). *Assume $\mathbf{v}^{(k)}$ ($k \in [K_b]$) are IID sampled from Gaussian $\mathcal{N}(\mu, \Sigma)$ with mean $\mu \in \mathbb{R}^H$ and positive definite covariance matrix $\Sigma \in \mathbb{R}^{H \times H}$. Let $d := \min_{k \in [K] \setminus [K_b]} \|\mathbf{v}^{(k)} - \mu\|_2$. Suppose that we use $\ell_2$ norm to measure vector distance and leverage the malicious client number estimator with an initial guess of a number of benign clients $\tilde{K}_b$ such that $K_m < \tilde{K}_b < K_b$. Then we have:*

$$\mathbb{P}\left[\hat{K}_b = K_b\right] \geq 1 - \frac{(3\tilde{K}_b - K_m - 2)^2 Tr(\Sigma)}{(\tilde{K}_b - K_m)^2 d^2} - \frac{2(K + K_b) Tr(\Sigma) \sigma_{max}^2(\Sigma^{-1/2})}{\sigma_{min}^2(\Sigma^{-1/2}) d^2}, \quad (11)$$

*where $\sigma_{max}(\Sigma^{-1/2})$, $\sigma_{min}(\Sigma^{-1/2})$ denote the maximal and minimal eigenvalue of matrix $\Sigma^{-1/2}$, and $Tr(\Sigma)$ denotes the trace of matrix $\Sigma$.*

*Proof sketch.* We first analyze the tail bound of the multivariate normal distribution as (Vershynin, 2018), and then derive the probabilistic relationships between the maliciousness scores of benign clients and those of malicious clients using the tail bounds. We finally upper bound the probability of overestimation and underestimation by opening up the probability formulations.

*Remark.* The lower bound in Equation (11) positively correlates with the minimal distance between the malicious characterization vector to the mean $\mu$ and is asymptotically close to 1 when the minimal distance $d$ is sufficiently large. It implies that when the malicious characterization vector is far away from the benign cluster (i.e., a large $d$), the malicious client number estimator has a high probability of accurate estimation. The lower bound in Equation (11) also shows that when the initial guess $\tilde{K}_b$ is closer to $K_b$, the lower bound of accurate estimation probability is higher, demonstrating the effectiveness of iterative optimization using Equation (10). Note that the condition of the initial guess $K_m < \tilde{K}_b < K_b$ is satisfiable by simply setting $\tilde{K}_b = \lceil K/2 \rceil$.

## 5 EXPERIMENTS

### 5.1 EXPERIMENT SETUP

**Datasets**. We evaluate Rob-FCP on computer vision datasets including MNIST (Deng, 2012), CIFAR-10 (Krizhevsky et al.), and Tiny-ImageNet (T-ImageNet) (Le & Yang, 2015). We additionally evaluate Rob-FCP on two realistic healthcare datasets, including SHHS (Zhang et al., 2018) and PathMNIST (Yang et al., 2023).

**Non-IID data construction**. We follow the standard evaluation setup of non-IID federated learning by sampling different label ratios for different clients from the Dirichlet distribution as the literature (Yurochkin et al., 2019; Lin et al., 2020; Wang et al., 2020; Gao et al., 2022). Concretely, we sample $p_{c,j} \sim \text{Dir}(\beta)$ and allocate a $p_{c,j}$ proportion of the instances with class $c$ to the client $j$. Here $\text{Dir}(\cdot)$ denotes the Dirichlet distribution and $\beta$ is a concentration parameter ($\beta > 0$). An advantage of this approach is that we can flexibly change the imbalance level by varying the concentration parameter $\beta$. If $\beta$ is set to a smaller value, then the partition is more unbalanced. We fixed the imbalance level $\beta$ as 0.5 without specification. We also consider alternative approaches to construct non-IID data with demographic differences in federated learning. Concretely, we split the SHHS dataset by sorting five different attributes wake time, N1, N2, N3, REM.

**Byzantine attacks.** To evaluate the robustness of Rob-FCP in the Byzantine setting, we compare Rob-FCP with the baseline FCP (Lu et al., 2023) under three types of Byzantine attacks: (1) *coverage attack* (CovAttack) with which malicious clients report the upper bound th conformity scores

Table 1: Marginal coverage / average set size under different Byzantine attacks with 40% ($K_m/K = 40\%$) malicious clients. The desired marginal coverage is 0.9. The results whose coverage rates closer to the all-benign-client scenario (shown in Table 6) are in bold. Note that here, it is not always the case that a smaller prediction set is better, as our goal is to identify the prediction set that would have been issued, when no client is malicious.

| Attack Method | Coverage Attack | | Efficiency Attack | | Gaussian Attack | |
|---|---|---|---|---|---|---|
| | FCP | Rob-FCP | FCP | Rob-FCP | FCP | Rob-FCP |
| **IID** | | | | | | |
| MNIST | 0.832 / 0.834 | **0.899** / 0.903 | 1.000 / 10.00 | **0.901** / 0.907 | 0.979 / 1.025 | **0.908** / 0.913 |
| CIFAR-10 | 0.831 / 1.189 | **0.906** / 1.641 | 1.000 / 10.00 | **0.902** / 1.617 | 0.916 / 1.733 | **0.899** / 1.609 |
| T-ImageNet | 0.830 / 12.97 | **0.898** / 21.65 | 1.000 / 200.0 | **0.903** / 22.63 | 0.918 / 25.69 | **0.906** / 24.15 |
| SHHS | 0.834 / 1.093 | **0.899** / 1.354 | 1.000 / 6.000 | **0.900** / 1.359 | 0.937 / 1.611 | **0.901** / 1.366 |
| PathMNIST | 0.840 / 0.997 | **0.907** / 1.259 | 1.000 / 9.000 | **0.901** / 1.228 | 1.000 / 6.632 | **0.909** / 1.275 |
| **non-IID** | | | | | | |
| MNIST | 0.805 / 1.284 | **0.899** / 1.783 | 1.000 / 10.00 | **0.902** / 1.804 | 0.941 / 2.227 | **0.923** / 2.182 |
| CIFAR-10 | 0.829 / 1.758 | **0.897** / 2.319 | 1.000 / 10.00 | **0.892** / 2.351 | 0.970 / 3.863 | **0.921** / 2.623 |
| T-ImageNet | 0.825 / 27.84 | **0.903** / 43.47 | 1.000 / 200.0 | **0.904** / 43.68 | 0.942 / 61.50 | **0.928** / 54.91 |
| SHHS | 0.835 / 1.095 | **0.901** / 1.365 | 1.000 / 6.000 | **0.901** / 1.366 | 0.937 / 1.609 | **0.900** / 1.359 |
| PathMNIST | 0.837 / 1.055 | **0.900** / 1.355 | 1.000 / 9.000 | **0.900** / 1.344 | 1.000 / 6.935 | **0.926** / 1.585 |

(a) CIFAR-10      (b) Tiny-ImageNet

Figure 2: Results of malicious client number estimation and conformal prediction performance in the setting with unknown numbers of malicious clients. The green horizontal line denotes the benign conformal performance. Rob-FCP estimates the number of malicious clients faithfully, and generally provides an empirical coverage rate matching the target.

(e.g., 1 for the mostly used LAC score) to induce a larger conformity score at the desired quantile and a lower coverage accordingly, (2) *efficiency attack* (EffAttack) with which malicious clients report the lower bound of the conformity scores (e.g., 0 for the mostly used LAC score) to induce a lower conformity score at the quantile and a larger prediction set, and (3) Gaussian Attack (GauAttack) with which malicious clients inject random Gaussian noises with standard deviation 0.5 to the scores to perturb the conformal calibration.

**Evaluation metric.** Consider the global test data set $\mathcal{D}_{\text{test}} = \{(X_i, Y_i)\}_{i=1}^{N_{\text{test}}}$. Let $C_\alpha(X_i)$ be the conformal prediction set given test sample $X_i$ and desired coverage level $1-\alpha$. We consider the metrics of *marginal coverage* $\sum_{i=1}^{N_{\text{test}}} \mathbb{I}[Y_i \in C_\alpha(X_i)] / N_{\text{test}}$ and the *average set size* $\sum_{i=1}^{N_{\text{test}}} |C_\alpha(X_i)| / N_{\text{test}}$. Without specification, the desired coverage level $1 - \alpha$ is set 0.9 across evaluations. We provide more details of experiment setups in Appendix H.1.

## 5.2 EVALUATION RESULTS

**Byzantine robustness of Rob-FCP in the IID and non-IID settings.** We evaluate the marginal coverage and average set size of Rob-FCP under coverage, efficiency, and Gaussian Attack and compare the results with the baseline FCP. We present results of FCP and Rob-FCP in existence of 40% ($K_m/K = 40\%$) malicious clients on MNIST, CIFAR-10, Tiny-ImageNet (T-ImageNet), SHHS, and PathMNIST in Table 1. The coverage of FCP deviates drastically from the desired coverage level 0.9 under Byzantine attacks, along with a deviation from the benign set size. In contrast, Rob-FCP achieves comparable marginal coverage and average set size in both IID and non-IID settings. Note that while in general a smaller prediction set is preferred, here the underlying global model is the same, so the same coverage always corresponds to the same set size. The goal here is, however, to identify the correct threshold that achieves our original coverage target (90%). We demonstrate the Byzantine robustness of Rob-FCP with different ratios of malicious clients (30%, 20%, 10%) in Table 7 in Appendix H.2.

**Rob-FCP with unknown numbers of malicious clients.** In Section 4, we consider a more challenging Byzantine setting where the number of malicious clients is unknown to the defender and

propose the malicious client number estimator to predict the size. Now we directly evaluate the precision of the malicious client number estimator and conformal prediction performance of Rob-FCP in this setting. The results in Figure 2 demonstrate that the malicious client number estimate ($\hat{K}_m$) matches the true number of malicious clients ($K_m$) very well, inducing a more desirable marginal coverage and average set size (i.e., closer to the benign performance) than FCP in the IID setting under coverage attack. We also demonstrate the conclusion in both IID and non-IID settings on all five datasets with various Byzantine attacks in Table 8 in Appendix H.2.

**Validation of coverage bounds of Rob-FCP.** In Theorem 1, we provide the lower bound and upper bound of the coverage rate of Rob-FCP as a function regarding the malicious client number ratio $\tau = K_m/K_b$ and sample sizes of clients. In Figure 3, we validate the coverage bounds in the IID settings on Tiny-ImageNet by comparing the theoretical bound and empirical marginal coverage under Gaussian attacks with different variances. The results show that the coverage bound is valid and tight, especially for a small maliciousness ratio $\tau$. By Equation (7), a large ratio $\tau$ amplifies the finite-sample error and induces inflation of the bounds, but the bounds asymptotically approach the desired coverage 0.9 with sufficiently large sample sizes.

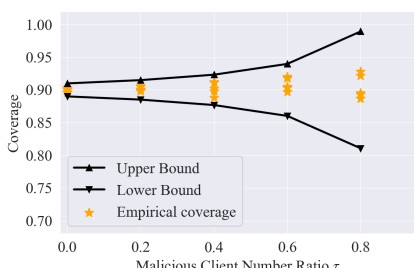

Figure 3: Coverage bounds of Rob-FCP (Theorem 1) in the IID settings on Tiny-ImageNet.

**Robustness of Rob-FCP with different conformity scores.** Besides applying LAC nonconformity scores, we also evaluate Rob-FCP with APS scores (Romano et al., 2020). The results in Figures 5 to 10 in Appendix H.2 demonstrate the Byzantine robustness of Rob-FCP with APS scores.

**Ablation study of different conformity score distribution characterization.** One key step in Rob-FCP is to characterize the conformity score distribution based on empirical observations. We adopt the histogram statistics approach as Equation (4). One can also sketch the score samples with cluster centers by clustering algorithms such as KMeans. Another alternative is to use a parametric approach such as fitting a Gaussian distribution to the score samples and characterizing them with the Gaussian mean and variances. We empirically compare different approaches in Figures 11 and 12 in Appendix H.2 and show that the histogram statistics approach achieves the best performance. We also provide ablation studies of different distance measurements in Figure 13 in Appendix H.2.

We included more evaluations of Rob-FCP on results with different non-IID construction, runtime of quantile computation, and results with incorrect numbers of malicious clients in Appendix H.2.

## 6    RELATED WORK

**Conformal prediction** is a statistical tool to construct the prediction set with guaranteed prediction coverage (Jin et al., 2023; Solari & Djordjilović, 2022; Yang & Kuchibhotla, 2021; Romano et al., 2020; Barber et al., 2021), assuming exchangeable data. Recently, **federated conformal prediction** (FCP) (Lu & Kalpathy-Cramer, 2021; Lu et al., 2023) adapts the conformal prediction to the federated learning and provides a rigorous guarantee of the distributed uncertainty quantification framework. DP-FCP (Plassier et al., 2023) proposes federated CP with differential privacy guarantees and provides valid coverage bounds under label shifting among clients. Humbert et al. propose a quantile-of-quantiles estimator for federated conformal prediction with a one-round communication and provide a locally differentially private version. WFCP (Zhu et al., 2023) applies FCP to wireless communication. However, no prior works explore the robustness of FCP against Byzantine agents which can report malicious statistics to downgrade the conformal prediction performance. We are the first to propose a robust FCP method with valid and tight coverage guarantees.

## 7    CONCLUSION

In this paper, we propose Rob-FCP, a certifiably Byzantine-robust federated conformal prediction algorithm with rigorous coverage guarantees in both IID and non-IID settings. Rob-FCP sketches the local samples of conformity scores with characterization vectors and detects the malicious clients with distance measurements in the vector space. We empirically show the robustness of Rob-FCP against Byzantine failures on five datasets and validate the theoretical coverage bounds in practice.

**Ethics statement.** We do not see potential ethical issues about Rob-FCP. In contrast, Rob-FCP is a robust framework against malicious clients in the federated conformal prediction settings and can safeguard the applications of FCP in safety-critical scenarios such as healthcare and medical diagnosis.

**Reproducibility statement.** The reproducibility of Rob-FCP span theoretical and experimental perspectives. We provide complete proofs of all the theoretical results in Appendix D and include insightful proof sketches in the main texts. We include implementation details in Appendix H.1 and upload the source codes for implementing Rob-FCP and reproducing experiment results as supplementary materials.

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

## Contents

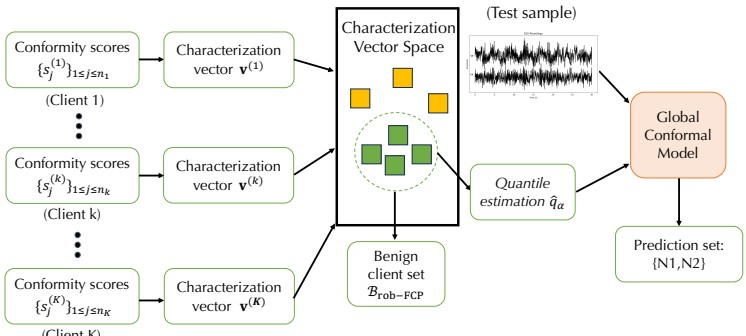

Figure 4: Overview of Rob-FCP. In this example, we have a non-REM stage 1 (N1) EEG recording, which is a hard and rare class often confused with non-REM stage 2 (N2).

## A    LIMITATIONS AND FUTURE WORKS

One possible limitation of Rob-FCP may lie in the restriction of the targeted Byzantine threat model. We mainly consider the Byzantine setting where a certain ratio of malicious clients reports arbitrary conformity score statistics. In such a Byzantine case, the break point is $\lceil K/2 \rceil$, indicating that any algorithm cannot tolerate $\lceil K/2 \rceil$ or more malicious clients. However, in practice, malicious clients have the flexibility of only manipulating partial conformity scores. In this case, the potential break point is a function of the maximal ratio of manipulated scores for each client and can be larger than $\lceil K/2 \rceil$. Therefore, it is interesting for future work to analyze the break point of robust FCP algorithms with respect to the total manipulation sizes and budgets of manipulation sizes for each client. Another threat model worthy of exploration in future work is the adversarial setting in FCP. In the adversarial setting, malicious clients can only manipulate the data samples instead of the conformity scores to downgrade the FCP performance. Therefore, potential defenses can consider adversarial conformal training procedures to collaboratively train a robust FCP model against perturbations in the data space.

To provide differential privacy guarantees of Rob-FCP, one practical approach is to add privacy-preserving noises to the characterization vectors before uploading them to the server. Essentially, we can view the characterization vector as the gradient in the setting of FL with differential privacy (DP) and add Gaussian noises to the characterization vector with differential privacy guarantees as a function of the scale of noises, which can be achieved by drawing analogy from the FL with DP setting (Zheng et al., 2021; Andrew et al., 2021; Zhang et al., 2022). Therefore, practically implementing the differential-private version of Rob-FCP is possible and straightforward.

## B    OVERVIEW OF ROB-FCP

We provide an overview of Rob-FCP in Figure 4.

## C    MORE RELATED WORK

**Byzantine learning** (Driscoll et al., 2003; Awerbuch et al., 2002; Lamport et al., 2019) refers to methods that can robustly aggregate updates from potentially malicious or faulty worker nodes in the distributed setting. Specifically, a line of works (Guerraoui et al., 2018; Pillutla et al., 2022; Data & Diggavi, 2021; Karimireddy et al., 2020; Yi et al., 2022) studies the resilience to Byzantine failures of distributed implementations of Stochastic Gradient Descent (SGD) and proposes different metrics to identify malicious gradients such as gradient norm (Blanchard et al., 2017) and coordinate-wise trimmed mean (Yin et al., 2018). However, the metrics are designed for the stability and convergence of distributed optimization and cannot be applied to the Byzantine FCP setting to provide rigorous coverage guarantees. In contrast, we propose Rob-FCP to perform Byzantine-robust distributed uncertainty quantification and provide valid and tight coverage bounds theoretically.

# D OMITTED PROOFS

## D.1 PROOF OF THEOREM 1

**Theorem 3** (Restatement of Theorem 1). *For $K$ clients including $K_b$ benign clients and $K_m := K - K_b$ malicious clients, each client reports a characterization vector $\mathbf{v}^{(k)} \in \Delta^H$ ($k \in [K]$) and a quantity $n_k \in \mathbb{Z}^+$ ($k \in [K]$) to the server. Suppose that the reported characterization vectors of benign clients are sampled from the same underlying multinomial distribution $\mathcal{D}$, while those of malicious clients can be arbitrary. Let $\epsilon$ be the estimation error of the data sketching by characterization vectors as illustrated in Equation (3). Under the assumption that $K_m < K_b$, the following holds with probability $1 - \beta$:*

$$\mathbb{P}\left[Y_{test} \in \hat{C}_\alpha(X_{test})\right] \geq 1 - \alpha - \frac{\epsilon n_b + 1}{n_b + K_b} - \frac{H\Phi^{-1}(1 - \beta/2HK_b)}{2\sqrt{n_b}}\left(1 + \frac{N_m}{n_b}\frac{2}{1-\tau}\right),$$

$$\mathbb{P}\left[Y_{test} \in \hat{C}_\alpha(X_{test})\right] \leq 1 - \alpha + \epsilon + \frac{K_b}{n_b + K_b} + \frac{H\Phi^{-1}(1 - \beta/2HK_b)}{2\sqrt{n_b}}\left(1 + \frac{N_m}{n_b}\frac{2}{1-\tau}\right).$$

(12)

*where $\tau = K_m/K_b$ is the ratio of the number of malicious clients and the number of benign clients, $N_m := \sum_{k \in [K] \setminus [K_b]} n_k$ is the total sample size of malicious clients, $n_b := \min_{k' \in [K_b]} n_{k'}$ is the minimal sample size of benign clients, and $\Phi^{-1}(\cdot)$ denotes the inverse of the cumulative distribution function (CDF) of standard normal distribution.*

*Proof.* The proof consists of 3 parts: (a) concentration analysis of the characterization vectors $\mathbf{v}^{(k)}$ for benign clients ($1 \leq k \leq K_b$), (b) analysis of the algorithm of the identification of malicious clients, and (c) analysis of the error of the coverage bound.

**Part (a)**: concentration analysis of the characterization vectors $\mathbf{v}^{(k)}$ for benign clients ($1 \leq k \leq K_b$).

Let $\mathbf{v}_h^{(k)}$ be the $h$-th element of vector $\mathbf{v}^{(k)}$. By definition, since $\mathbf{v}^{(k)}$ is sampled from a multinomial distribution, $\mathbf{v}_h^{(k)}$ denotes the success rate estimate of a Bernoulli distribution. We denote the event probabilities of the multinomial distribution $\mathcal{D}$ as $\overline{\mathbf{v}}$. Therefore, the true success rate of the Bernoulli distribution at the $h$-th position is $\overline{\mathbf{v}}_h$. According to the binomial proportion confidence interval Wallis (2013), we have:

$$\mathbb{P}\left[\left|\mathbf{v}_h^{(k)} - \overline{\mathbf{v}}_h\right| > \Phi^{-1}(1 - \beta/2HK_b)\frac{\sqrt{n_{ks}n_{kf}}}{n_k\sqrt{n_k}}\right] \leq \beta/HK_b,$$

(13)

where $\beta/HK_b$ is the probability confidence, $\Phi^{-1}(\cdot)$ denotes the inverse of the CDF of the standard normal distribution, and $n_{ks}$ and $n_{kf} := n_k - n_{ks}$ are the number of success and failures in $n_k$ Bernoulli trials, respectively. Applying the inequality $n_{ks}n_{kf} \leq n_k^2/4$ in Equation (13), the following holds:

$$\mathbb{P}\left[\left|\mathbf{v}_h^{(k)} - \overline{\mathbf{v}}_h\right| > \frac{\Phi^{-1}(1 - \beta/2HK_b)}{2\sqrt{n_k}}\right] \leq \beta/HK_b.$$

(14)

Applying the union bound for $H$ elements in vector $\mathbf{v}^{(k)}$ and $K_b$ characterization vectors of benign clients, the following holds with probability $1 - \beta$:

$$\left|\mathbf{v}_h^{(k)} - \overline{\mathbf{v}}_h\right| \leq \frac{\Phi^{-1}(1 - \beta/2HK_b)}{2\sqrt{\min_{k' \in [K_b]} n_{k'}}}, \ \forall k \in [K_b], \ \forall h \in [H],$$

(15)

from which we can derive the bound of difference for $\ell_1$ norm distance as:

$$\left\|\mathbf{v}^{(k)} - \overline{\mathbf{v}}\right\|_1 \leq r(\beta) := \frac{H\Phi^{-1}(1 - \beta/2HK_b)}{2\sqrt{\min_{k' \in [K_b]} n_{k'}}}, \ \forall k \in [K_b],$$

(16)

where $r(\beta)$ is the perturbation radius of random vector $\mathbf{v}$ given confidence level $1 - \beta$. $\forall k_1, k_2 \in [K_b]$, the following holds with probability $1 - \beta$ due to the triangular inequality:

$$\left\|\mathbf{v}^{(k_1)} - \mathbf{v}^{(k_2)}\right\|_1 \leq \left\|\mathbf{v}^{(k_1)} - \overline{\mathbf{v}}\right\|_1 + \left\|\mathbf{v}^{(k_2)} - \overline{\mathbf{v}}\right\|_1 \leq 2r(\beta).$$

(17)

Furthermore, due to the fact that $\|\mathbf{v}\|_p \leq \|\mathbf{v}\|_1$ for any integer $p \geq 1$, the following holds with probability $1 - \beta$:

$$\left\|\mathbf{v}^{(k)} - \overline{\mathbf{v}}\right\|_p \leq \left\|\mathbf{v}^{(k)} - \overline{\mathbf{v}}\right\|_1 \leq r(\beta), \tag{18}$$

$$\left\|\mathbf{v}^{(k_1)} - \mathbf{v}^{(k_2)}\right\|_p \leq \left\|\mathbf{v}^{(k_1)} - \mathbf{v}^{(k_2)}\right\|_1 \leq 2r(\beta). \tag{19}$$

**Part (b)**: analysis of the algorithm of the identification of malicious clients.

Let $N(k,n)$ be the set of the index of $n$ nearest clients to the $k$-th client based on the metrics of $\ell_p$ norm distance in the space of characterization vectors. Then the maliciousness scores $M(k)$ for the $k$-th client ($k \in [K]$) can be defined as:

$$M(k) := \frac{1}{K_b - 1} \sum_{k' \in N(k, K_b - 1)} \left\|\mathbf{v}^{(k)} - \mathbf{v}^{(k')}\right\|_p. \tag{20}$$

Let $\mathcal{B}$ be the set of the index of benign clients identified by Algorithm 1 by selecting the clients associated with the lowest $K_b$ maliciousness scores. We will consider the following cases separately: (1) $\mathcal{B}$ contains exactly $K_b$ benign clients, and (2) $\mathcal{B}$ contains at least one malicious client indexed by $m$.

*Case (1)*: $\mathcal{B}$ ($|\mathcal{B}| = K_b$) contains exactly $K_b$ benign clients. We can derive as follows:

$$\left\|\sum_{k=1}^{K_b} \frac{n_k}{N_b} \mathbf{v}^{(k)} - \overline{\mathbf{v}}\right\|_p \leq \sum_{k=1}^{K_b} \frac{n_k}{N_b} \left\|\mathbf{v}^{(k)} - \overline{\mathbf{v}}\right\|_p \qquad \text{[triangular inequality]} \tag{21}$$

$$\leq \sum_{k=1}^{K_b} \frac{n_k}{N_b} r(\beta) \qquad \text{[by Equation (18)]} \tag{22}$$

$$= r(\beta), \tag{23}$$

where $N_b := \sum_{k \in [K_b]} n_k$ is the total sample size of benign clients.

*Case (2)*: $\mathcal{B}$ ($|\mathcal{B}| = K_b$) contains at least one malicious client indexed by $m$. Since we assume $K_m < K_b$, there are at most $K_b - 1$ malicious clients in $\mathcal{B}$. Therefore, there is at least 1 benign client in $[K] \backslash \mathcal{B}$ indexed by $b$. We can derive the lower bound of the maliciousness score for the $m$-th client $M(m)$ as:

$$M(m) = \frac{1}{K_b - 1} \sum_{k' \in N(m, K_b - 1)} \left\|\mathbf{v}^{(m)} - \mathbf{v}^{(k')}\right\|_p \tag{24}$$

$$\geq \frac{1}{K_b - 1} \sum_{k' \in N(m, K_b - 1), k' \in [K_b]} \left\|\mathbf{v}^{(m)} - \mathbf{v}^{(k')}\right\|_p. \tag{25}$$

Since there are at least $K_b - K_m$ benign clients in $\mathcal{B}$ (there are at most $K_m$ malicious clients in $\mathcal{B}$), there exists one client indexed by $b_b$ ($b_b \in \mathcal{B}$) such that:

$$\left\|\mathbf{v}^{(m)} - \mathbf{v}^{(b_b)}\right\|_p \leq \frac{(K_b - 1)M(m)}{K_b - K_m} \tag{26}$$

We can derive the upper bound of the maliciousness score for the $b$-th benign client $M(b)$ as:

$$M(b) = \frac{1}{K_b - 1} \sum_{k' \in N(b, K_b - 1)} \left\|\mathbf{v}^{(b)} - \mathbf{v}^{(k')}\right\|_p \tag{27}$$

$$\leq 2r(\beta) \qquad \text{[by Equation (19)]} \tag{28}$$

Since the $m$-th client is included in $\mathcal{B}$ and identified as a benign client, while the $b$-th client is not in $\mathcal{B}$, the following holds according to the procedure in Algorithm 1:

$$M(b) \geq M(m), \tag{29}$$

from which we can derive the following by combining Equation (26) and Equation (28):

$$\left\| \mathbf{v}^{(m)} - \mathbf{v}^{(b_b)} \right\|_p \leq \frac{(K_b - 1)2r(\beta)}{K_b - K_m} \tag{30}$$

Then, we can derive the upper bound of $\left\| \mathbf{v}^{(m)} - \overline{\mathbf{v}} \right\|_p$, $\forall m \in \mathcal{B}$ and $K_b < m \leq K$ as follows:

$$\left\| \mathbf{v}^{(m)} - \overline{\mathbf{v}} \right\|_p \leq \left\| \mathbf{v}^{(m)} - \mathbf{v}^{(b_b)} \right\|_p + \left\| \mathbf{v}^{(b_b)} - \overline{\mathbf{v}} \right\|_p \tag{31}$$

$$\leq \frac{2(K_b - 1)r(\beta)}{K_b - K_m} + r(\beta) \tag{32}$$

Finally, we can derive as follows:

$$\left\| \sum_{k \in \mathcal{B}} \frac{n_k}{N_{\mathcal{B}}} \mathbf{v}^{(k)} - \overline{\mathbf{v}} \right\|_p \leq \sum_{k \in \mathcal{B}} \frac{n_k}{N_{\mathcal{B}}} \left\| \mathbf{v}^{(k)} - \overline{\mathbf{v}} \right\|_p \tag{33}$$

$$\leq \sum_{k \in \mathcal{B}, k \in [K_b]} \frac{n_k}{N_{\mathcal{B}}} \left\| \mathbf{v}^{(k)} - \overline{\mathbf{v}} \right\|_p + \sum_{k \in \mathcal{B}, k \in [K] \setminus [K_b]} \frac{n_k}{N_{\mathcal{B}}} \left\| \mathbf{v}^{(k)} - \overline{\mathbf{v}} \right\|_p \tag{34}$$

$$\leq \sum_{k \in \mathcal{B}, k \in [K_b]} \frac{n_k}{N_{\mathcal{B}}} r(\beta) + \sum_{k \in \mathcal{B}, k \in [K] \setminus [K_b]} \frac{n_k}{N_{\mathcal{B}}} \left[ \frac{2(K_b - 1)r(\beta)}{K_b - K_m} + r(\beta) \right] \tag{35}$$

$$\leq r(\beta) + \sum_{k \in \mathcal{B}, k \in [K] \setminus [K_b]} \frac{n_k}{N_{\mathcal{B}}} \frac{2(K_b - 1)r(\beta)}{K_b - K_m} \tag{36}$$

$$\leq r(\beta) \left( 1 + \frac{N_m}{\min_{k' \in [K_b]} n_{k'}} \frac{2}{1 - \tau} \right), \tag{37}$$

where $N_m := \sum_{k \in [K] \setminus [K_b]} n_k$ is the total sample size of malicious clients, $N_{\mathcal{B}}$ is the total sample size of clients in $\mathcal{B}$, and $\tau := \frac{K_m}{K_b}$ is the ratio of the number of malicious clients to the number of benign clients.

Combining *case (1)* and *case (2)*, we can conclude that:

$$\left\| \sum_{k \in \mathcal{B}} \frac{n_k}{N_{\mathcal{B}}} \mathbf{v}^{(k)} - \overline{\mathbf{v}} \right\|_p \leq \max \left\{ 1, 1 + \frac{N_m}{\min_{k' \in [K_b]} n_{k'}} \frac{2}{1 - \tau} \right\} r(\beta) \tag{38}$$

$$= \left( 1 + \frac{N_m}{\min_{k' \in [K_b]} n_{k'}} \frac{2}{1 - \tau} \right) r(\beta) \tag{39}$$

**Part (c)**: analysis of the error of the coverage bound. In this part, we attempt to translate the error of aggregated vectors induced by malicious clients to the error of the bound of marginal coverage. Let $F_1(q, \mathbf{v}) := \sum_{j=1}^{H} \mathbb{I}[a_j < q] \mathbf{v}_j$, where $q \in [0, 1]$ and $a_j$ is the $j$-th partition point used to construct the characterization vector $\mathbf{v} \in \Delta^H$. Let $F_2(q, \mathbf{v}) := \sum_{j=1}^{H} \mathbb{I}[a_{j-1} < q] \mathbf{v}_j$. Then by definition, we know that $F_1(q_\alpha, \overline{\mathbf{v}}) \leq \mathbb{P}\left[ Y_{\text{test}} \in \hat{C}_\alpha(X_{\text{test}}) \right] \leq F_2(q_\alpha, \overline{\mathbf{v}})$, where $q_\alpha$ is the true $(1 - \alpha)$ quantile value of the non-conformity scores, $\overline{\mathbf{v}}$ is the event probability of the multinormial distribution $\mathcal{D}$, and $\hat{C}_\alpha(X_{\text{test}})$ is the conformal prediction set of input $X_{\text{test}}$ using the true benign calibrated conformity score $q_\alpha$ and statistics of score distribution $\overline{\mathbf{v}}$.

Let $\hat{q}_\alpha$ be the quantile estimate during calibration. FCP (Lu et al., 2023) proves that if the rank of quantile estimate $\hat{q}_\alpha$ is between $(1 - \alpha - \epsilon)(N + K)$ and $(1 - \alpha + \epsilon)(N + K)$, then we have:

$$F_1(\hat{q}_\alpha, \overline{\mathbf{v}}) \geq 1 - \alpha - \frac{\epsilon N_{\mathcal{B}} + 1}{N_{\mathcal{B}} + K_b}, \quad F_2(\hat{q}_\alpha, \overline{\mathbf{v}}) \leq 1 - \alpha + \epsilon + \frac{K_b}{N_{\mathcal{B}} + K_b}. \tag{40}$$

Now we start deriving the error of $F_1(\cdot, \cdot)$ induced by the malicious clients. Let $\hat{\mathbf{v}} := \sum_{k \in \mathcal{B}} \frac{n_k}{N} \mathbf{v}^{(k)}$ be the estimated mean of characterization vector. Based on the results in part (b), we can derive as

follows:

$$|F_1(\hat{q}_\alpha, \overline{\mathbf{v}}) - F_1(\hat{q}_\alpha, \hat{\mathbf{v}})| = \left| \sum_{j=1}^{H} \mathbb{I}\left[a_j < \hat{q}_\alpha\right] \overline{\mathbf{v}}_j - \sum_{j=1}^{H} \mathbb{I}\left[a_j < \hat{q}_\alpha\right] \hat{\mathbf{v}}_j \right| \tag{41}$$

$$\leq \sum_{j=1}^{H} \mathbb{I}\left[a_j < \hat{q}_\alpha\right] |\overline{\mathbf{v}}_j - \hat{\mathbf{v}}_j| \tag{42}$$

$$\leq \|\overline{\mathbf{v}} - \hat{\mathbf{v}}\|_1 \tag{43}$$

$$\leq \left(1 + \frac{N_m}{\min_{k' \in [K_b] n_{k'}}} \frac{2}{1 - \tau}\right) r(\beta) \tag{44}$$

From triangular inequalities, we have:

$$F_1(\hat{q}_\alpha, \overline{\mathbf{v}}) - |F_1(\hat{q}_\alpha, \overline{\mathbf{v}}) - F_1(\hat{q}_\alpha, \hat{\mathbf{v}})| \leq F_1(\hat{q}_\alpha, \hat{\mathbf{v}}) \leq \mathbb{P}\left[Y_{\text{test}} \in \hat{C}_\alpha(X_{\text{test}})\right]. \tag{45}$$

Similarly, we can derive that $|F_2(\hat{q}_\alpha, \overline{\mathbf{v}}) - F_2(\hat{q}_\alpha, \hat{\mathbf{v}})| \leq \left(1 + \frac{N_m}{\min_{k' \in [K_b] n_{k'}}} \frac{2}{1 - \tau}\right) r(\beta)$ and have:

$$F_2(\hat{q}_\alpha, \overline{\mathbf{v}}) + |F_2(\hat{q}_\alpha, \overline{\mathbf{v}}) - F_2(\hat{q}_\alpha, \hat{\mathbf{v}})| \geq F_2(\hat{q}_\alpha, \hat{\mathbf{v}}) \geq \mathbb{P}\left[Y_{\text{test}} \in \hat{C}_\alpha(X_{\text{test}})\right]. \tag{46}$$

Plugging in the terms in Equations (40) and (44) and leveraging the fact $N_{\mathcal{B}} \geq n_b$, we finally conclude that the following holds with probability $1 - \beta$:

$$\mathbb{P}\left[Y_{\text{test}} \in \hat{C}_\alpha(X_{\text{test}})\right] \geq 1 - \alpha - \frac{\epsilon n_b + 1}{n_b + K_b} - \frac{H\Phi^{-1}(1 - \beta/2HK_b)}{2\sqrt{n_b}}\left(1 + \frac{N_m}{n_b}\frac{2}{1 - \tau}\right),$$

$$\mathbb{P}\left[Y_{\text{test}} \in \hat{C}_\alpha(X_{\text{test}})\right] \leq 1 - \alpha + \epsilon + \frac{K_b}{n_b + K_b} + \frac{H\Phi^{-1}(1 - \beta/2HK_b)}{2\sqrt{n_b}}\left(1 + \frac{N_m}{n_b}\frac{2}{1 - \tau}\right). \tag{47}$$

where $\tau = K_m/K_b$ is the ratio of the number of malicious clients and the number of benign clients, $N_m := \sum_{k \in [K] \setminus [K_b]} n_k$ is the total sample size of malicious clients, $n_b := \min_{k' \in [K_b]} n_{k'}$ is the minimal sample size of benign clients, and $\Phi^{-1}(\cdot)$ denotes the inverse of the cumulative distribution function (CDF) of standard normal distribution.

$\square$

## D.2  PROOF OF COROLLARY 1

**Corollary 2** (Restatement of Corollary 1). *For $K$ clients including $K_b$ benign clients and $K_m := K - K_b$ malicious clients, each client reports a characterization vector $\mathbf{v}^{(k)} \in \Delta^H$ ($k \in [K]$) and a quantity $n_k \in \mathbb{Z}^+$ ($k \in [K]$) to the server. Suppose that the reported characterization vectors of benign client $k$ ($k \in [K_b]$) are sampled from a multinomial distribution $\mathcal{D}_k$, while those of malicious clients are arbitrary. Under the assumption that the disparity of multinomial distribution $\mathcal{D}_k$ ($k \in [K_b]$) is bounded as in Assumption 3.1. Let $\epsilon$ be the estimation error of the data sketching by characterization vectors as illustrated in Equation (3). Then with probability $1 - \beta$, the following holds:*

$$\mathbb{P}\left[Y_{test} \in \hat{C}_\alpha(X_{test})\right] \geq 1 - \alpha - \frac{\epsilon n_b + 1}{n_b + K_b} - \frac{H\Phi^{-1}(1 - \beta/2HK_b)}{2\sqrt{n_b}}\left(1 + \frac{N_m}{n_b}\frac{2}{1 - \tau}\right) - \frac{N_m}{n_b}\frac{\sigma}{1 - \tau},$$

$$\mathbb{P}\left[Y_{test} \in \hat{C}_\alpha(X_{test})\right] \leq 1 - \alpha + \epsilon + \frac{K_b}{n_b + K_b} + \frac{H\Phi^{-1}(1 - \beta/2HK_b)}{2\sqrt{n_b}}\left(1 + \frac{N_m}{n_b}\frac{2}{1 - \tau}\right) + \frac{N_m}{n_b}\frac{\sigma}{1 - \tau}. \tag{48}$$

*where $\tau = K_m/K_b$ is the ratio of the number of malicious clients and the number of benign clients, $N_m := \sum_{k \in [K] \setminus [K_b]} n_k$ is the total sample size of malicious clients, $n_b := \min_{k' \in [K_b]} n_{k'}$ is the minimal sample size of benign clients, and $\Phi^{-1}(\cdot)$ denotes the inverse of CDF of the standard normal distribution.*

*Proof.* The general structure of the proof follows the proof of Theorem 1. We will omit similar derivation and refer to the proof of Theorem 1 for details. The proof consists of 3 parts: (a) concentration analysis of the characterization vectors $\mathbf{v}^{(k)}$ for benign clients ($1 \leq k \leq K_b$), (b) analysis of the algorithm of the identification of malicious clients, and (c) analysis of the error of the coverage bound.

**Part (a)**: concentration analysis of the characterization vectors $\mathbf{v}^{(k)}$ for benign clients ($1 \leq k \leq K_b$).

Let $\overline{\mathbf{v}}^{(k)}$ be the event probability of the multinormial distribution $\mathcal{D}^{(k)}$ for $k \in [K_b]$. By applying binomial proportion approximate normal confidence interval and union bound as in Part (a) in the proof of Theorem 1, with confidence $1 - \beta$, we have:

$$\left\| \mathbf{v}^{(k)} - \overline{\mathbf{v}}^{(k)} \right\|_1 \leq r(\beta) := \frac{H \Phi^{-1}(1 - \beta/2HK_b)}{2\sqrt{\min_{k' \in [K_b]} n_{k'}}}, \ \forall k \in [K_b], \tag{49}$$

where $r(\beta)$ is the perturbation radius of random vector $\mathbf{v}^{(k)}$ given confidence level $1 - \beta$. $\forall k_1, k_2 \in [K_b]$, we can upper bound the $\ell_p$ norm distance between $\mathbf{v}^{(k_1)}$ and $\mathbf{v}^{(k_2)}$ as:

$$\left\| \mathbf{v}^{(k_1)} - \mathbf{v}^{(k_2)} \right\|_p \leq \left\| \mathbf{v}^{(k_1)} - \overline{\mathbf{v}}^{(k_1)} \right\|_p + \left\| \overline{\mathbf{v}}^{(k_1)} - \overline{\mathbf{v}}^{(k_2)} \right\|_p + \left\| \mathbf{v}^{(k_2)} - \overline{\mathbf{v}}^{(k_1)} \right\|_p \tag{50}$$

$$\leq \left\| \mathbf{v}^{(k_1)} - \overline{\mathbf{v}}^{(k_1)} \right\|_1 + \left\| \overline{\mathbf{v}}^{(k_1)} - \overline{\mathbf{v}}^{(k_2)} \right\|_p + \left\| \mathbf{v}^{(k_2)} - \overline{\mathbf{v}}^{(k_1)} \right\|_1 \tag{51}$$

$$\leq 2r(\beta) + \sigma, \tag{52}$$

where Equation (52) holds by Equation (49) and Assumption 3.1.

**Part (b)**: analysis of the algorithm of the identification of malicious clients.

Let $N(k, n)$ be the set of the index of $n$ nearest clients to the $k$-th client based on the metrics of $\ell_p$ norm distance in the space of characterization vectors. Then the maliciousness scores $M(k)$ for the $k$-th client ($k \in [K]$) can be defined as:

$$M(k) := \frac{1}{K_b - 1} \sum_{k' \in N(k, K_b - 1)} \left\| \mathbf{v}^{(k)} - \mathbf{v}^{(k')} \right\|_p. \tag{53}$$

Let $\mathcal{B}$ be the set of the index of benign clients identified by Algorithm 1 by selecting the clients associated with the lowest $K_b$ maliciousness scores. We will consider the following cases separately: (1) $\mathcal{B}$ contains exactly $K_b$ benign clients, and (2) $\mathcal{B}$ contains at least one malicious client indexed by $m$.

*Case (1)*: $\mathcal{B}$ ($|\mathcal{B}| = K_b$) contains exactly $K_b$ benign clients. We can derive as follows:

$$\left\| \sum_{k=1}^{K_b} \frac{n_k}{N_b} \mathbf{v}^{(k)} - \sum_{k=1}^{K_b} \frac{n_k}{N_b} \overline{\mathbf{v}}^{(k)} \right\|_p \leq \sum_{k=1}^{K_b} \frac{n_k}{N_b} \left\| \mathbf{v}^{(k)} - \overline{\mathbf{v}}^{(k)} \right\|_p \tag{54}$$

$$\leq \sum_{k=1}^{K_b} \frac{n_k}{N_b} r(\beta) \tag{55}$$

$$= r(\beta), \tag{56}$$

where $N_b := \sum_{k \in [K_b]} n_k$ is the total sample size of benign clients.

*Case (2)*: $\mathcal{B}$ ($|\mathcal{B}| = K_b$) contains at least one malicious client indexed by $m$. Since we assume $K_m < K_b$, there are at most $K_b - 1$ malicious clients in $\mathcal{B}$. Therefore, there is at least 1 benign client in $[K] \backslash \mathcal{B}$ indexed by $b$. From the fact that $M(m) \leq M(b)$ and expanding the definitions the maliciousness score as Part (b) in the proof of Theorem 1, we get that $\exists b_b \in \mathcal{B}, b_b \in [K_b]$:

$$\left\| \mathbf{v}^{(m)} - \mathbf{v}^{(b_b)} \right\|_p \leq \frac{(K_b - 1)(2r(\beta) + \sigma)}{K_b - K_m} \tag{57}$$

Therefore, we can upper bound the distance between the estimated global event probability vector $\sum_{k \in \mathcal{B}} \frac{n_k}{N_{\mathcal{B}}} \mathbf{v}^{(k)}$ and the benign global event probability vector $\sum_{k \in [K_b]} \frac{n_k}{N_b} \overline{\mathbf{v}}^{(k)}$.

We first show that $\forall k \in [K_b]$, we have:

$$\left\| \mathbf{v}^{(k)} - \sum_{k \in [K_b]} \frac{n_k}{N_b} \overline{\mathbf{v}}^{(k)} \right\|_p \leq \sum_{k \in [K_b]} \frac{n_k}{N_b} \left\| \mathbf{v}^{(k)} - \overline{\mathbf{v}}^{(k)} \right\|_p \leq r(\beta). \tag{58}$$

Then, we can derive as follows:

$$\left\| \sum_{k \in \mathcal{B}} \frac{n_k}{N_{\mathcal{B}}} \mathbf{v}^{(k)} - \sum_{k \in [K_b]} \frac{n_k}{N_b} \overline{\mathbf{v}}^{(k)} \right\|_p \tag{59}$$

$$\leq \sum_{k \in \mathcal{B}, k \in [K_b]} \frac{n_k}{N_{\mathcal{B}}} \left\| \mathbf{v}^{(k)} - \sum_{k \in [K_b]} \frac{n_k}{N_b} \overline{\mathbf{v}}^{(k)} \right\|_p + \sum_{k \in \mathcal{B}, k \in [K] \setminus [K_b]} \frac{n_k}{N_{\mathcal{B}}} \left\| \mathbf{v}^{(k)} - \sum_{k \in [K_b]} \frac{n_k}{N_b} \overline{\mathbf{v}}^{(k)} \right\|_p \tag{60}$$

$$\leq \sum_{k \in \mathcal{B}, k \in [K_b]} \frac{n_k}{N_{\mathcal{B}}} r(\beta) + \sum_{k \in \mathcal{B}, k \in [K] \setminus [K_b]} \frac{n_k}{N_{\mathcal{B}}} \left[ \left\| \mathbf{v}^{(k)} - \mathbf{v}^{(b_b)} \right\|_p + \left\| \mathbf{v}^{(b_b)} - \sum_{k \in [K_b]} \frac{n_k}{N_b} \overline{\mathbf{v}}^{(k)} \right\|_p \right] \tag{61}$$

$$\leq \sum_{k \in \mathcal{B}, k \in [K_b]} \frac{n_k}{N_{\mathcal{B}}} r(\beta) + \sum_{k \in \mathcal{B}, k \in [K] \setminus [K_b]} \frac{n_k}{N_{\mathcal{B}}} \left[ \frac{(K_b - 1)(2r(\beta) + \sigma)}{K_b - K_m} + r(\beta) \right] \tag{62}$$

$$\leq r(\beta) + \sum_{k \in \mathcal{B}, k \in [K] \setminus [K_b]} \frac{n_k}{N_{\mathcal{B}}} \frac{(K_b - 1)(2r(\beta) + \sigma)}{K_b - K_m} \tag{63}$$

$$\leq r(\beta) \left( 1 + \frac{N_m}{n_b} \frac{2}{1 - \tau} \right) + \frac{N_m}{n_b} \frac{\sigma}{1 - \tau}. \tag{64}$$

**Part (c)**: analysis of the error of the coverage bound.

Let $F_1(q, \mathbf{v}) := \sum_{j=1}^H \mathbb{I}[a_j < q] \mathbf{v}_j$, where $q \in [0, 1]$ and $a_j$ is the $j$-th partition point used to construct the characterization vector $\mathbf{v} \in \Delta^H$. Let $F_2(q, \mathbf{v}) := \sum_{j=1}^H \mathbb{I}[a_{j-1} < q] \mathbf{v}_j$. This part follows the same procedure to translate the error of aggregated vectors induced by malicious clients to the error of the bound of marginal coverage. The only difference is that in the non-iid setting, the error of aggregated vectors formulated in Equation (64) is different than the iid setting. Therefore, by analyzing the connection between characterization vector and coverage similarly in Part (3) in the proof of Theorem 1, we have:

$$|F_1(\hat{q}_\alpha, \overline{\mathbf{v}}) - F_1(\hat{q}_\alpha, \hat{\mathbf{v}})| \leq r(\beta) \left( 1 + \frac{N_m}{n_b} \frac{2}{1 - \tau} \right) + \frac{N_m}{n_b} \frac{\sigma}{1 - \tau}, \tag{65}$$

$$|F_2(\hat{q}_\alpha, \overline{\mathbf{v}}) - F_2(\hat{q}_\alpha, \hat{\mathbf{v}})| \leq r(\beta) \left( 1 + \frac{N_m}{n_b} \frac{2}{1 - \tau} \right) + \frac{N_m}{n_b} \frac{\sigma}{1 - \tau}, \tag{66}$$

where $\overline{\mathbf{v}} := \sum_{k \in [K_b]} \frac{n_k}{N_b} \overline{\mathbf{v}}^{(k)}$ and $\hat{\mathbf{v}} := \sum_{k \in \mathcal{B}} \frac{n_k}{N_{\mathcal{B}}} \mathbf{v}^{(k)}$. On the other hand, from triangular inequalities, we have:

$$F_1(\hat{q}_\alpha, \overline{\mathbf{v}}) - |F_1(\hat{q}_\alpha, \overline{\mathbf{v}}) - F_1(\hat{q}_\alpha, \hat{\mathbf{v}})| \leq F_1(\hat{q}_\alpha, \hat{\mathbf{v}}) \leq \mathbb{P}\left[ Y_{\text{test}} \in \hat{C}_\alpha(X_{\text{test}}) \right], \tag{67}$$

$$F_2(\hat{q}_\alpha, \overline{\mathbf{v}}) + |F_2(\hat{q}_\alpha, \overline{\mathbf{v}}) - F_2(\hat{q}_\alpha, \hat{\mathbf{v}})| \geq F_2(\hat{q}_\alpha, \hat{\mathbf{v}}) \geq \mathbb{P}\left[ Y_{\text{test}} \in \hat{C}_\alpha(X_{\text{test}}) \right]. \tag{68}$$

Plugging in the terms, we finally conclude that the following holds with probability $1 - \beta$:

$$\mathbb{P}\left[ Y_{\text{test}} \in \hat{C}_\alpha(X_{\text{test}}) \right] \geq 1 - \alpha - \frac{\epsilon n_b + 1}{n_b + K_b} - \frac{H\Phi^{-1}(1 - \beta/2HK_b)}{2\sqrt{n_b}} \left( 1 + \frac{N_m}{n_b} \frac{2}{1 - \tau} \right) - \frac{N_m}{n_b} \frac{\sigma}{1 - \tau},$$

$$\mathbb{P}\left[ Y_{\text{test}} \in \hat{C}_\alpha(X_{\text{test}}) \right] \leq 1 - \alpha + \epsilon + \frac{K_b}{n_b + K_b} + \frac{H\Phi^{-1}(1 - \beta/2HK_b)}{2\sqrt{n_b}} \left( 1 + \frac{N_m}{n_b} \frac{2}{1 - \tau} \right) + \frac{N_m}{n_b} \frac{\sigma}{1 - \tau}. \tag{69}$$

where $\tau = K_m/K_b$ is the ratio of the number of malicious clients and the number of benign clients, $N_m := \sum_{k \in [K] \setminus [K_b]} n_k$ is the total sample size of malicious clients, $n_b := \min_{k' \in [K_b]} n_{k'}$ is the minimal sample size of benign clients, and $\Phi^{-1}(\cdot)$ denotes the inverse of CDF of the standard normal distribution.

$\square$

### D.3 PROOF OF THEOREM 2

**Theorem 4** (Restatement of Theorem 2). *Assume $\mathbf{v}^{(k)}$ ($k \in [K_b]$) are IID sampled from Gaussian $\mathcal{N}(\mu, \Sigma)$ with mean $\mu \in \mathbb{R}^H$ and positive definite covariance matrix $\Sigma \in \mathbb{R}^{H \times H}$. Let $d := \min_{k \in [K] \setminus [K_b]} \|\mathbf{v}^{(k)} - \mu\|_2$. Suppose that we use $\ell_2$ norm to measure vector distance and leverage the malicious client number estimator with an initial guess of a number of benign clients $\tilde{K}_b$ such that $K_m < \tilde{K}_b < K_b$. Then we have:*

$$\mathbb{P}\left[\hat{K}_b = K_b\right] \geq 1 - \frac{(3\tilde{K}_b - K_m - 2)^2 Tr(\Sigma)}{(\tilde{K}_b - K_m)^2 d^2} - \frac{2(K + K_b)Tr(\Sigma)\sigma_{max}^2(\Sigma^{-1/2})}{\sigma_{min}^2(\Sigma^{-1/2})d^2}, \quad (70)$$

*where $\sigma_{max}(\Sigma^{-1/2})$, $\sigma_{min}(\Sigma^{-1/2})$ denote the maximal and minimal eigenvalue of matrix $\Sigma^{-1/2}$, and $Tr(\Sigma)$ denotes the trace of matrix $\Sigma$.*

*Proof.* From the concentration inequality of multivariate Gaussian distribution (Vershynin, 2018), the following holds for $\mathbf{v}^{(k)} \sim \mathcal{N}(\mu, \Sigma)$:

$$\mathbb{P}\left[\|\mathbf{v}^{(k)} - \mu\|_2 \leq \sqrt{\frac{1}{\delta}Tr(\Sigma)}\right] \geq 1 - \delta. \quad (71)$$

Applying union bound for all benign clients $k \in [K_b]$, the following concentration bound holds:

$$\mathbb{P}\left[\|\mathbf{v}^{(k)} - \mu\|_2 \leq \sqrt{\frac{K_b}{\delta}Tr(\Sigma)}, \ \forall k \in [K_b]\right] \geq 1 - \delta, \quad (72)$$

Let the perturbation radius $r := \frac{\tilde{K}_b - K_m}{3\tilde{K}_b - K_m - 2}d$. Then we can derive that:

$$\mathbb{P}\left[\|\mathbf{v}^{(k)} - \mu\|_2 \leq r := \frac{\tilde{K}_b - K_m}{3\tilde{K}_b - K_m - 2}d, \ \forall k \in [K_b]\right] \geq 1 - \frac{(3\tilde{K}_b - K_m - 2)^2 Tr(\Sigma)}{(\tilde{K}_b - K_m)^2 d^2} := 1 - \delta. \quad (73)$$

The following discussion is based on the fact that $\|\mathbf{v}^{(k)} - \mu\|_2 \leq r := \frac{\tilde{K}_b - K_m}{3\tilde{K}_b - K_m - 2}d, \ \forall k \in [K_b]$, and the confidence $1 - \delta$ will be incorporated in the final statement. Let $N(k, n)$ be the index set of $n$ nearest neighbors of client $k$ in the characterization vector space with the metric of $\ell_2$ norm distance. We consider the maliciousness score $M(b)$ of any benign client $b \in [K_b]$:

$$M(b) = \frac{1}{\tilde{K}_b - 1} \sum_{k' \in N(b, \tilde{K}_b - 1)} \left\|\mathbf{v}^{(b)} - \mathbf{v}^{(k')}\right\|_2 \quad (74)$$

$$\leq \max_{k' \in [K_b]} \left\|\mathbf{v}^{(b)} - \mathbf{v}^{(k')}\right\|_2 \quad (75)$$

$$\leq \max_{k' \in [K_b]} \left\{\left\|\mathbf{v}^{(b)} - \mu\right\|_2 + \left\|\mu - \mathbf{v}^{(k')}\right\|_2\right\} \quad (76)$$

$$\leq \frac{2(\tilde{K}_b - K_m)}{3\tilde{K}_b - K_m - 2}d. \quad (77)$$

Equation (75) holds since the average of distances to $\tilde{K}_b - 1$ nearest vectors is upper bounded by the average of distances to arbitrary $\tilde{K}_b - 1$ benign clients, which is upper bounded by the maximal distance to benign clients. Equation (77) holds by plugging in the results in Equation (73).

We consider the maliciousness score $M(m)$ of any malicious client $m \in [K]\backslash[K_b]$:

$$M(m) = \frac{1}{\tilde{K}_b - 1} \sum_{k' \in N(m, \tilde{K}_b - 1)} \left\| \mathbf{v}^{(m)} - \mathbf{v}^{(k')} \right\|_2 \tag{78}$$

$$\geq \frac{1}{\tilde{K}_b - 1} \sum_{k' \in N(m, \tilde{K}_b - 1), k' \in [K_b]} \left\| \mathbf{v}^{(m)} - \mathbf{v}^{(k')} \right\|_2 \tag{79}$$

$$\geq \frac{1}{\tilde{K}_b - 1} \sum_{k' \in N(m, \tilde{K}_b - 1), k' \in [K_b]} \left[ \left\| \mathbf{v}^{(m)} - \mu \right\|_2 - \left\| \mu - \mathbf{v}^{(k')} \right\|_2 \right] \tag{80}$$

$$\geq \frac{1}{\tilde{K}_b - 1} (\tilde{K}_b - K_m) \left( d - \frac{\tilde{K}_b - K_m}{3\tilde{K}_b - K_m - 2} d \right) \tag{81}$$

$$\geq \frac{2(\tilde{K}_b - K_m)}{3\tilde{K}_b - K_m - 2} d. \tag{82}$$

Equation (81) holds since $d := \min_{k \in [K]\backslash[K_b]} \|\mathbf{v}^{(k)} - \mu\|_2$ by definition. Therefore, from Equation (77) and Equation (82), we can conclude that with probability $1 - \delta$, $M(m) \geq M(b)$, $\forall b \in [K_b], m \in [K]\backslash[K_b]$, which implies that $\forall k \in [K_b], I(k) \in [K_b]$ and $\forall k \in [K] - [K_b], I(k) \in [K]\backslash[K_b]$.

Recall that the estimate of the number of benign clients $\hat{K}_b$ is given by:

$$\hat{K}_b = \underset{z \in [K]}{\arg \max} \left[ \frac{1}{z} \sum_{k=1}^{z} \log p(\mathbf{v}^{(I(k))}; \mu, \Sigma) - \frac{1}{K - z} \sum_{k=z+1}^{K} \log p(\mathbf{v}^{(I(k))}; \mu, \Sigma) \right]. \tag{83}$$

For ease of notation, let $T(z) := \frac{1}{z} \sum_{k=1}^{z} \log p(\mathbf{v}^{(I(k))}; \mu, \Sigma) - \frac{1}{K - z} \sum_{k=z+1}^{K} \log p(\mathbf{v}^{(I(k))}; \mu, \Sigma)$ for $z \in [K]$ and $d_k := \mathbf{v}^{(I(k))} - \mu$ for $k \in [K]$. Then we can upper bound the probability of an

underestimate of the number of malicious clients $\mathbb{P}\left[\hat{K}_b < K_b\right]$ as follows:

$$\mathbb{P}\left[\hat{K}_b < K_b\right] \tag{84}$$

$$=\mathbb{P}\left[T(\hat{K}_b) > T(K_b)\right] \tag{85}$$

$$\leq\mathbb{P}\left[\frac{-(K_b - \hat{K}_b)}{K_b\hat{K}_b}\sum_{k=1}^{\hat{K}_b}\log p(\mathbf{v}^{(I(k))}) + \frac{K - \hat{K}_b + K_b}{K_b(K - \hat{K}_b)}\sum_{k=\hat{K}_b+1}^{K_b}\log p(\mathbf{v}^{(I(k))})\right.$$

$$\left. < \frac{K_b - \hat{K}_b}{(K - K_b)(K - \hat{K}_b)}\sum_{k=K_b+1}^{K}\log p(\mathbf{v}^{(I(k))})\right] \tag{86}$$

$$\leq\mathbb{P}\left[\frac{K - \hat{K}_b + K_b}{K_b(K - \hat{K}_b)}\sum_{k=\hat{K}_b+1}^{K_b}-d_k^T\Sigma^{-1}d_k < \frac{K_b - \hat{K}_b}{(K - K_b)(K - \hat{K}_b)}\sum_{k=K_b+1}^{K}-d_k^T\Sigma^{-1}d_k\right] \tag{87}$$

$$\leq\mathbb{P}\left[\frac{K_b - \hat{K}_b}{(K - K_b)}\sum_{k=K_b+1}^{K}\|d_k^T\Sigma^{-1/2}\|_2^2 < \frac{K - \hat{K}_b + K_b}{K_b}\sum_{k=\hat{K}_b+1}^{K_b}\|d_k^T\Sigma^{-1/2}\|_2^2\right] \tag{88}$$

$$\leq\mathbb{P}\left[\frac{K_b - \hat{K}_b}{(K - K_b)}\sum_{k=K_b+1}^{K}\sigma_{\min}^2(\Sigma^{-1/2})\|d_k^T\|_2^2 < \frac{K - \hat{K}_b + K_b}{K_b}\sum_{k=\hat{K}_b+1}^{K_b}\sigma_{\max}^2(\Sigma^{-1/2})\|d_k^T\|_2^2\right] \tag{89}$$

$$\leq\mathbb{P}\left[\sigma_{\min}^2(\Sigma^{-1/2})d^2 < \frac{K - \hat{K}_b + K_b}{K_b}\sigma_{\max}^2(\Sigma^{-1/2})\max_{k\in[K_b]}\|d_k^T\|_2^2\right] \tag{90}$$

$$\leq\mathbb{P}\left[\max_{k\in[K_b]}\|d_k^T\|_2 > \sqrt{\frac{K_b}{K + K_b}}\frac{\sigma_{\min}(\Sigma^{-1/2})d}{\sigma_{\max}(\Sigma^{-1/2})}\right] \tag{91}$$

$$\leq\frac{(K + K_b)\text{Tr}(\Sigma)\sigma_{\max}^2(\Sigma^{-1/2})}{\sigma_{\min}^2(\Sigma^{-1/2})d^2} \tag{92}$$

Equation (86) holds by plugging in the definitions in Equation (83) and rearranging the terms. Equation (87) holds by dropping the positive term $\frac{-(K_b - \hat{K}_b)}{K_b\hat{K}_b}\sum_{k=1}^{\hat{K}_b}\log p(\mathbf{v}^{(I(k))})$ and rearranging log-likelihood terms of multivariate Gaussian with $d_k$. Equation (89) holds by leveraging the fact that $\sigma_{\min}(\Sigma^{-1/2})\|d_k^T\|_2 \leq \|d_k^T\Sigma^{-1/2}\|_2 \leq \sigma_{\max}(\Sigma^{-1/2})\|d_k^T\|_2$.

Similarly, we can upper bound the probability of overestimation of the number of malicious clients $\mathbb{P}\left[\hat{K}_b > K_b\right]$ as:

$$\mathbb{P}\left[\hat{K}_b > K_b\right] \leq \frac{(K + K_b)\text{Tr}(\Sigma)\sigma_{\max}^2(\Sigma^{-1/2})}{\sigma_{\min}^2(\Sigma^{-1/2})d^2}. \tag{93}$$

We can finally conclude that:

$$\mathbb{P}\left[\hat{K}_b = K_b\right] \geq 1 - \frac{(3\tilde{K}_b - K_m - 2)^2\text{Tr}(\Sigma)}{(\tilde{K}_b - K_m)^2d^2} - \frac{2(K + K_b)\text{Tr}(\Sigma)\sigma_{\max}^2(\Sigma^{-1/2})}{\sigma_{\min}^2(\Sigma^{-1/2})d^2}. \tag{94}$$

$$\square$$

# E  IMPROVEMENTS WITH DKW INEQUALITY

## E.1  IMPROVEMENT OF THEOREM 1 WITH DKW INEQUALITY

**Theorem 5** (Improvement of Theorem 1). *For $K$ clients including $K_b$ benign clients and $K_m := K - K_b$ malicious clients, each client reports a characterization vector $\mathbf{v}^{(k)} \in \Delta^H$ ($k \in [K]$) and a quantity $n_k \in \mathbb{Z}^+$ ($k \in [K]$) to the server. Suppose that the reported characterization vectors of benign clients are sampled from the same underlying multinomial distribution $\mathcal{D}$, while those of malicious clients can be arbitrary. Let $\epsilon$ be the estimation error of the data sketching by characterization vectors as illustrated in Equation (3). Under the assumption that $K_m < K_b$, the following holds with probability $1 - \beta$:*

$$
\begin{aligned}
\mathbb{P}\left[Y_{test} \in \hat{C}_\alpha(X_{test})\right] &\geq 1 - \alpha - \frac{\epsilon n_b + 1}{n_b + K_b} - H\sqrt{\frac{\ln(2K_b/\beta)}{2n_b}}\left(1 + \frac{N_m}{n_b}\frac{2}{1-\tau}\right), \\
\mathbb{P}\left[Y_{test} \in \hat{C}_\alpha(X_{test})\right] &\leq 1 - \alpha + \epsilon + \frac{K_b}{n_b + K_b} + H\sqrt{\frac{\ln(2K_b/\beta)}{2n_b}}\left(1 + \frac{N_m}{n_b}\frac{2}{1-\tau}\right),
\end{aligned}
\tag{95}
$$

*where $\tau = K_m/K_b$ is the ratio of the number of malicious clients and the number of benign clients, $N_m := \sum_{k \in [K] \setminus [K_b]} n_k$ is the total sample size of malicious clients, and $n_b := \min_{k' \in [K_b]} n_{k'}$ is the minimal sample size of benign clients.*

*Proof.* The proof structure follows the proof of Theorem 1 and consists of 3 parts: (a) concentration analysis of the characterization vectors $\mathbf{v}^{(k)}$ for benign clients ($1 \leq k \leq K_b$), (b) analysis of the algorithm of the identification of malicious clients, and (c) analysis of the error of the coverage bound. Part (b) and (c) are exactly the same as the proof Theorem 1 and the only difference lies in the use of a more advanced concentration bound in part (a), which provides concentration analysis of the characterization vectors $\mathbf{v}^{(k)}$ for benign clients ($1 \leq k \leq K_b$). Let $\mathbf{v}^{(k)}_h$ be the $h$-th element of vector $\mathbf{v}^{(k)}$. According to the Dvoretzky–Kiefer–Wolfowitz (DKW) inequality, we have:

$$
\mathbb{P}\left[\left|\mathbf{v}^{(k)}_h - \overline{\mathbf{v}}_h\right| > \beta\right] \leq 2\exp\left\{-2H\beta^2\right\}, \ \forall h \in \{1, 2, .., H\}.
\tag{96}
$$

Applying the union bound for $K_b$ characterization vectors of benign clients, the following holds with probability $1 - \beta$:

$$
\left|\mathbf{v}^{(k)}_h - \overline{\mathbf{v}}_h\right| \leq \sqrt{\frac{\ln(2K_b/\beta)}{2n_b}}, \ \forall k \in [K_b], \ \forall h \in [H],
\tag{97}
$$

from which we can derive the bound of difference for $\ell_1$ norm distance as:

$$
\left\|\mathbf{v}^{(k)} - \overline{\mathbf{v}}\right\|_1 \leq r(\beta) := H\sqrt{\frac{\ln(2K_b/\beta)}{2n_b}}, \ \forall k \in [K_b],
\tag{98}
$$

where $r(\beta)$ is the perturbation radius of random vector $\mathbf{v}$ given confidence level $1 - \beta$. $\forall k_1, k_2 \in [K_b]$, the following holds with probability $1 - \beta$ due to the triangular inequality:

$$
\left\|\mathbf{v}^{(k_1)} - \mathbf{v}^{(k_2)}\right\|_1 \leq \left\|\mathbf{v}^{(k_1)} - \overline{\mathbf{v}}\right\|_1 + \left\|\mathbf{v}^{(k_2)} - \overline{\mathbf{v}}\right\|_1 \leq 2r(\beta).
\tag{99}
$$

Furthermore, due to the fact that $\|\mathbf{v}\|_p \leq \|\mathbf{v}\|_1$ for any integer $p \geq 1$, the following holds with probability $1 - \beta$:

$$
\left\|\mathbf{v}^{(k)} - \overline{\mathbf{v}}\right\|_p \leq \left\|\mathbf{v}^{(k)} - \overline{\mathbf{v}}\right\|_1 \leq r(\beta),
\tag{100}
$$

$$
\left\|\mathbf{v}^{(k_1)} - \mathbf{v}^{(k_2)}\right\|_p \leq \left\|\mathbf{v}^{(k_1)} - \mathbf{v}^{(k_2)}\right\|_1 \leq 2r(\beta).
\tag{101}
$$

Then following the part (b) and (c) in the proof of Theorem 1, we can finally conclude that:

$$
\begin{aligned}
\mathbb{P}\left[Y_{\text{test}} \in \hat{C}_\alpha(X_{\text{test}})\right] &\geq 1 - \alpha - \frac{\epsilon n_b + 1}{n_b + K_b} - H\sqrt{\frac{\ln(2K_b/\beta)}{2n_b}}\left(1 + \frac{N_m}{n_b}\frac{2}{1-\tau}\right), \\
\mathbb{P}\left[Y_{\text{test}} \in \hat{C}_\alpha(X_{\text{test}})\right] &\leq 1 - \alpha + \epsilon + \frac{K_b}{n_b + K_b} + H\sqrt{\frac{\ln(2K_b/\beta)}{2n_b}}\left(1 + \frac{N_m}{n_b}\frac{2}{1-\tau}\right),
\end{aligned}
\tag{102}
$$

$\square$

### E.2 IMPROVEMENT OF COROLLARY 1 WITH DKW INEQUALITY

**Corollary 3** (Improvement of Corollary 1 with DKW inequality)**.** *Under the same definitions and conditions in Theorem 1 and with Assumption 3.1, the following holds with probability* $1 - \beta$:

$$\mathbb{P}\left[Y_{test} \in \hat{C}_\alpha(X_{test})\right] \geq 1 - \alpha - \frac{\epsilon n_b + 1}{n_b + K_b} - H\sqrt{\frac{\ln(2K_b/\beta)}{2n_b}}\left(1 + \frac{N_m}{n_b}\frac{2}{1 - \tau}\right) - \frac{N_m}{n_b}\frac{\sigma}{1 - \tau},$$

$$\mathbb{P}\left[Y_{test} \in \hat{C}_\alpha(X_{test})\right] \leq 1 - \alpha + \epsilon + \frac{K_b}{n_b + K_b} + H\sqrt{\frac{\ln(2K_b/\beta)}{2n_b}}\left(1 + \frac{N_m}{n_b}\frac{2}{1 - \tau}\right) + \frac{N_m}{n_b}\frac{\sigma}{1 - \tau}. \tag{103}$$

*Proof.* We conclude the proof by leveraging the concentration analysis in the proof of Theorem 5 and part (b) and part (c) in the proof of Corollary 1. □

## F ANALYSIS OF ROB-FCP WITH AN OVERESTIMATED NUMBER OF BENIGN CLIENTS $K_b'$

**Theorem 6** (Theorem 1 with an overestimated number of benign clients)**.** *For $K$ clients including $K_b$ benign clients and $K_m := K - K_b$ malicious clients, each client reports a characterization vector $\mathbf{v}^{(k)} \in \Delta^H$ ($k \in [K]$) and a quantity $n_k \in \mathbb{Z}^+$ ($k \in [K]$) to the server. Suppose that the reported characterization vectors of benign clients are sampled from the same underlying multinomial distribution $\mathcal{D}$, while those of malicious clients can be arbitrary. Let $\epsilon$ be the estimation error of the data sketching by characterization vectors as illustrated in Equation (3). **Let $K_b' > K_b$ be the overestimated number of benign clients.** We also assume benign clients and malicious clients have the same sample sizes. Under the assumption that $K_m < K_b$, the following holds with probability $1 - \beta$:*

$$\mathbb{P}\left[Y_{test} \in \hat{C}_\alpha(X_{test})\right] \geq 1 - \alpha - \frac{\epsilon n_b + 1}{n_b + K_b} - \left[1 - \frac{K_b}{K_b'}\left(1 - \frac{H\Phi^{-1}(1 - \beta/2HK_b)}{2\sqrt{n_b}}\right)\right],$$

$$\mathbb{P}\left[Y_{test} \in \hat{C}_\alpha(X_{test})\right] \leq 1 - \alpha + \epsilon + \frac{K_b}{n_b + K_b} + \left[1 - \frac{K_b}{K_b'}\left(1 - \frac{H\Phi^{-1}(1 - \beta/2HK_b)}{2\sqrt{n_b}}\right)\right], \tag{104}$$

*where $\tau = K_m/K_b$ is the ratio of the number of malicious clients and the number of benign clients, $N_m := \sum_{k \in [K] \setminus [K_b]} n_k$ is the total sample size of malicious clients, $n_b := \min_{k' \in [K_b]} n_{k'}$ is the minimal sample size of benign clients, and $\Phi^{-1}(\cdot)$ denotes the inverse of the cumulative distribution function (CDF) of standard normal distribution.*

*Proof.* The proof consists of 3 parts: (a) concentration analysis of the characterization vectors $\mathbf{v}^{(k)}$ for benign clients ($1 \leq k \leq K_b$), (b) analysis of the algorithm of the identification of malicious clients, and (c) analysis of the error of the coverage bound. Part (a) and (c) follow that of Theorem 1, and thus, we provide the details of part (b) here. Let $N(k, n)$ be the set of the index of $n$ nearest clients to the $k$-th client based on the metrics of $\ell_p$ norm distance in the space of characterization vectors. Then the maliciousness scores $M(k)$ for the $k$-th client ($k \in [K]$) can be defined as:

$$M(k) := \frac{1}{K_b - 1}\sum_{k' \in N(k, K_b - 1)}\left\|\mathbf{v}^{(k)} - \mathbf{v}^{(k')}\right\|_p. \tag{105}$$

Let $\mathcal{B}$ be the set of the index of benign clients identified by Algorithm 1 by selecting the clients associated with the lowest $K_b'$ maliciousness scores. We will consider the following cases separately:

(1) $\mathcal{B}$ contains exactly $K_b$ benign clients, and (2) $\mathcal{B}$ contains at least one malicious client indexed by $m$. *Case (1)*: $\mathcal{B}$ $(|\mathcal{B}| = K_b')$ contains all $K_b$ benign clients. We can derive as follows:

$$\left\| \sum_{k \in \mathcal{B}} \frac{n_k}{N_{\mathcal{B}}} \mathbf{v}^{(k)} - \overline{\mathbf{v}} \right\|_p \leq \sum_{k \in \mathcal{B}} \frac{n_k}{N_{\mathcal{B}}} \left\| \mathbf{v}^{(k)} - \overline{\mathbf{v}} \right\|_p \tag{106}$$

$$\leq \sum_{k \in \mathcal{B}, k \in [K_b]} \frac{n_k}{N_{\mathcal{B}}} \left\| \mathbf{v}^{(k)} - \overline{\mathbf{v}} \right\|_p + \sum_{k \in \mathcal{B}, k \in [K] \setminus [K_b]} \frac{n_k}{N_{\mathcal{B}}} \left\| \mathbf{v}^{(k)} - \overline{\mathbf{v}} \right\|_p \tag{107}$$

$$\leq \sum_{k \in \mathcal{B}, k \in [K_b]} \frac{n_k}{N_{\mathcal{B}}} r(\beta) + \sum_{k \in \mathcal{B}, k \in [K] \setminus [K_b]} \frac{n_k}{N_{\mathcal{B}}} \times 1 \tag{108}$$

$$= \frac{K_b}{K_b'} r(\beta) + \left( 1 - \frac{K_b}{K_b'} \right) \tag{109}$$

$$= 1 - \frac{K_b}{K_b'} \left( 1 - r(\beta) \right) \tag{110}$$

*Case (2)*: $\mathcal{B}$ $(|\mathcal{B}| = K_b')$ does not contain all benign clients, which implicates that for any malicious client $m \in \mathcal{B}$, we can derive the lower bound of the maliciousness score for the $m$-th client $M(m)$ as:

$$M(m) = \frac{1}{K_b' - 1} \sum_{k' \in N(m, K_b' - 1)} \left\| \mathbf{v}^{(m)} - \mathbf{v}^{(k')} \right\|_p \tag{111}$$

$$\geq \frac{1}{K_b' - 1} \sum_{k' \in N(m, K_b' - 1), k' \in [K_b]} \left\| \mathbf{v}^{(m)} - \mathbf{v}^{(k')} \right\|_p . \tag{112}$$

Since there are at least $K_b' - K_m$ benign clients in $\mathcal{B}$ (there are at most $K_m$ malicious clients in $\mathcal{B}$), there exists one client indexed by $b_b$ $(b_b \in \mathcal{B})$ such that:

$$\left\| \mathbf{v}^{(m)} - \mathbf{v}^{(b_b)} \right\|_p \leq \frac{(K_b' - 1) M(m)}{K_b' - K_m} \tag{113}$$

We can derive the upper bound of the maliciousness score for the $b$-th benign client (one benign client not in $\mathcal{B}$) $M(b)$ as:

$$M(b) = \frac{1}{K_b' - 1} \sum_{k' \in N(b, K_b' - 1)} \left\| \mathbf{v}^{(b)} - \mathbf{v}^{(k')} \right\|_p \tag{114}$$

$$\leq \frac{K_b - 1}{K_b' - 1} 2r(\beta) + \frac{K_b - K_b'}{K_b' - 1} \tag{115}$$

Since the $m$-th client is included in $\mathcal{B}$ and identified as a benign client, while the $b$-th client is not in $\mathcal{B}$, the following holds according to the procedure in Algorithm 1:

$$M(b) \geq M(m), \tag{116}$$

Then, we can derive the upper bound of $\left\| \mathbf{v}^{(m)} - \overline{\mathbf{v}} \right\|_p$, $\forall m \in \mathcal{B}$ and $K_b < m \leq K$ as follows:

$$\left\| \mathbf{v}^{(m)} - \overline{\mathbf{v}} \right\|_p \leq \left\| \mathbf{v}^{(m)} - \mathbf{v}^{(b_b)} \right\|_p + \left\| \mathbf{v}^{(b_b)} - \overline{\mathbf{v}} \right\|_p \tag{117}$$

$$\leq \frac{(K_b - 1) 2r(\beta) + K_b - K_b'}{K_b' - K_m} \tag{118}$$

Finally, we can derive as follows:

$$\left\| \sum_{k \in \mathcal{B}} \frac{n_k}{N_{\mathcal{B}}} \mathbf{v}^{(k)} - \overline{\mathbf{v}} \right\|_p \leq \sum_{k \in \mathcal{B}} \frac{n_k}{N_{\mathcal{B}}} \left\| \mathbf{v}^{(k)} - \overline{\mathbf{v}} \right\|_p \tag{119}$$

$$\leq \sum_{k \in \mathcal{B}, k \in [K_b]} \frac{n_k}{N_{\mathcal{B}}} \left\| \mathbf{v}^{(k)} - \overline{\mathbf{v}} \right\|_p + \sum_{k \in \mathcal{B}, k \in [K] \setminus [K_b]} \frac{n_k}{N_{\mathcal{B}}} \left\| \mathbf{v}^{(k)} - \overline{\mathbf{v}} \right\|_p \tag{120}$$

$$\leq \sum_{k \in \mathcal{B}, k \in [K_b]} \frac{n_k}{N_{\mathcal{B}}} r(\beta) + \sum_{k \in \mathcal{B}, k \in [K] \setminus [K_b]} \frac{n_k}{N_{\mathcal{B}}} \frac{(K_b - 1)2r(\beta) + K_b - K_b'}{K_b' - K_m} \tag{121}$$

$$\leq \frac{K_b}{K_b'} r(\beta) + \frac{K_b' - K_b}{K_b'} \frac{(K_b - 1)2r(\beta) + K_b - K_b'}{K_b' - K_m} \tag{122}$$

Combining *case (1)* and *case (2)*, we can conclude that:

$$\left\| \sum_{k \in \mathcal{B}} \frac{n_k}{N_{\mathcal{B}}} \mathbf{v}^{(k)} - \overline{\mathbf{v}} \right\|_p \leq \max \left\{ 1 - \frac{K_b}{K_b'} \left(1 - r(\beta)\right), \frac{K_b}{K_b'} r(\beta) + \frac{K_b' - K_b}{K_b'} \frac{(K_b - 1)2r(\beta) + K_b - K_b'}{K_b' - K_m} \right\}$$

$$= 1 - \frac{K_b}{K_b'} \left(1 - r(\beta)\right) \tag{123}$$

Finally, by applying the analysis of part (a) and (c) in the proof of Theorem 1, we can conclude that:

$$\mathbb{P} \left[ Y_{\text{test}} \in \hat{C}_\alpha(X_{\text{test}}) \right] \geq 1 - \alpha - \frac{\epsilon n_b + 1}{n_b + K_b} - \left[ 1 - \frac{K_b}{K_b'} \left( 1 - \frac{H \Phi^{-1}(1 - \beta/2HK_b)}{2\sqrt{n_b}} \right) \right],$$

$$\mathbb{P} \left[ Y_{\text{test}} \in \hat{C}_\alpha(X_{\text{test}}) \right] \leq 1 - \alpha + \epsilon + \frac{K_b}{n_b + K_b} + \left[ 1 - \frac{K_b}{K_b'} \left( 1 - \frac{H \Phi^{-1}(1 - \beta/2HK_b)}{2\sqrt{n_b}} \right) \right], \tag{124}$$

$\square$

# G ALGORITHM OF ROB-FCP

We provide the complete pseudocodes of malicious client identification in Rob-FCP in Algorithm 1. First, we characterize the conformity scores $\{s_j^{(k)}\}_{j \in [n_k]}$ with a vector $\mathbf{v}^{(k)} \in \mathbb{R}^H$ for client $k$ ($k \in [K]$) via histogram statistics as Equation (4). Then, we compute the pairwise $\ell_p$-norm ($p \in \mathbb{Z}^+$) vector distance and the maliciousness scores for clients, which are the averaged vector distance to the clients in the $K_b - 1$ nearest neighbors, where $K_b$ is the number of benign clients. Finally, the benign set identified by Rob-FCP $\mathcal{B}_{\text{Rob-FCP}}$ is the set of the index of the clients with the lowest $K_b$ maliciousness scores in $\{M(k)\}_{k=1}^K$.

# H EXPERIMENTS

## H.1 EXPERIMENT SETUP

**Datasets**. We evaluate Rob-FCP on computer vision datasets including MNIST (Deng, 2012), CIFAR-10 (Krizhevsky et al.), and Tiny-ImageNet (T-ImageNet) (Le & Yang, 2015). We additionally evaluate Rob-FCP on two realistic healthcare datasets, including SHHS (Zhang et al., 2018) and PathMNIST (Yang et al., 2023). The MNIST dataset consists of a collection of 70,000 handwritten digit images, each of which is labeled with the corresponding digit (0 through 9) that the image represents. CIFAR-10 consists of 60,000 32x32 color images, each belonging to one of the following 10 classes: airplane, automobile, bird, cat, deer, dog, frog, horse, ship, and truck. Tiny-ImageNet consists of 200 different classes, each represented by 500 training images, making a total of 100,000 training images. Additionally, it has 10,000 validation images and 10,000 test images, with 50 images per class for both validation and test sets. Each image in Tiny-ImageNet is a 64x64 colored image. SHHS (the Sleep Heart Health Study) is a large-scale multi-center study to determine

---

**Algorithm 1** Malicious client identification

---

1: **Input:** number of clients $K$, number of benign clients $K_b$, sets of scores for $K$ clients $\left\{s_j^{(k)}\right\}_{j \in [n_k], k \in [K]}$, parameter $p$ in $\ell_p$ norm distance.

2: **Output:** set of benign clients $\mathcal{B}_{\text{Rob-FCP}}$.

3: **for** $k = 1$ **to** $K$ **do**

4:     Characterize the conformity score observations $\left\{s_j^{(k)}\right\}_{j \in [n_k]}$ with a vector $\mathbf{v}^{(k)}$ for client $k$ as Equation (4).

5: **end for**

6: **for** $k_1 = 1$ **to** $K$ **do**

7:     **for** $k_2 = 1$ **to** $K$ **do**

8:         Compute the vector distance $d_{k_1,k_2} \leftarrow \|\mathbf{v}^{(k_1)} - \mathbf{v}^{(k_2)}\|_p$.

9:     **end for**

10: **end for**

11: **for** $k = 1$ **to** $K$ **do**

12:     Compute the set of index of $K_b - 1$ nearest neighbors for client $k$: $N_{ear}(k, K_b - 1)$.

13:     Compute maliciousness scores of client $k$ as $M(k) \leftarrow \dfrac{1}{K_b - 1} \sum_{k' \in N_{ear}(k, K_b - 1)} d_{k,k'}$.

14: **end for**

15: Compute the index set of benign clients $\mathcal{B}_{\text{Rob-FCP}}$ as the associated index of the lowest $K_b$ maliciousness scores in $\{M(k)\}_{k=1}^{K}$.

---

consequences of sleep-disordered breathing. We use the EEG recordings from SHHS for the sleep-staging task, where every 30-second-epoch is classified into Wake, N1, N2, N3 and REM stages. 2,514 patients (2,545,869 samples) were used for training the DNN, and 2,514 patients (2,543,550 samples) were used for calibration and testing. PathMNIST is a 9-class classification dataset consisting of 107,180 hematoxylin and eosin stained histological images. 89,996 images were used to train the DNN and 7,180 were used for calibration and testing.

**Training and evaluation strategy.** In the IID setting, we randomly partitioned the datasets into local datasets of multiple clients and further split them into a local training set and a conformal calibration set. In the non-IID setting, except for SHHS, we partition the datasets by sampling the proportion of each label from Dirichlet distribution for every agent, following the literature (Li et al., 2022a). For SHHS, we assign the patients to different clients according to the proportion of their time being awake. The parameter of the Dirichlet distribution is fixed as $0.5$ across the evaluations. We pretrain the models with standard FedAvg algorithm (McMahan et al., 2016). We use the same collaboratively pretrained model for conformal prediction for different methods for fair comparisons. We perform conformal prediction with nonconformity scores LAC (Sadinle et al., 2019) and APS (Romano et al., 2020). Without specification, we use the LAC score by default across evaluations. Given a pretrained estimator $\hat{\pi} : \mathbb{R}^d \mapsto \Delta^C$ with $d$-dimensional input and $C$ classes, the LAC non-conformity score is formulated as:

$$S_{\hat{\pi}_y}^{\text{LAC}}(x, y) = 1 - \hat{\pi}_y(x). \tag{125}$$

The APS non-conformity score is formulated as:

$$S_{\hat{\pi}_y}^{\text{APS}}(x, y) = \sum_{j \in \mathcal{Y}} \hat{\pi}_j(x) \mathbb{I}[\hat{\pi}_j(x) > \hat{\pi}_y(x)] + \hat{\pi}_y(x) u, \tag{126}$$

where $\mathbb{I}[\cdot]$ is the indicator function and $u$ is uniformly sampled over the interval $[0, 1]$.

**Byzantine attacks.** To evaluate the robustness of Rob-FCP in the Byzantine setting, we compare Rob-FCP with the baseline FCP (Lu et al., 2023) under three types of Byzantine attacks: (1) *coverage attack* (CovAttack) which reports the largest conformity scores to induce a larger conformity score at the desired quantile and a lower coverage accordingly, (2) *efficiency attack* (EffAttack) which reports the smallest conformity scores to induce a lower conformity score at the quantile and a larger prediction set, and (3) Gaussian Attack (GauAttack) which injects random Gaussian noises to the scores to perturb the conformal calibration. The gaussian noises are sampled from a univariate Gaussian $\mathcal{N}(0, 0.5)$ with zero mean and $0.5$ variance.

## H.2 Additional evaluation results

Table 2: Marginal coverage / average set size with different non-IID imbalance levels $\beta$, the parameter of Dirichlet distribution where the label ratios of clients are sampled from. The evaluation is done under different Coverage attack with $40\%$ ($K_m/K = 40\%$) malicious clients. The desired marginal coverage is 0.9.

|  | $\beta = 0.1$ | $\beta = 0.3$ | $\beta = 0.5$ | $\beta = 0.7$ | $\beta = 0.9$ |
|---|---|---|---|---|---|
| FCP (MNIST) | 0.780 / 1.173 | 0.817 / 1.318 | 0.833 / 1.384 | 0.805 / 1.265 | 0.828 / 1.363 |
| Rob-FCP (MNIST) | 0.899 / 1.806 | 0.905 / 1.809 | 0.903 / 1.827 | 0.898 / 1.781 | 0.893 / 1.768 |
| FCP (CIFAR-10) | 0.806 / 1.641 | 0.821 / 1.717 | 0.836 / 1.791 | 0.823 / 1.744 | 0.824 / 1.723 |
| Rob-FCP (CIFAR-10) | 0.899 / 2.260 | 0.907 / 2.405 | 0.892 / 2.243 | 0.904 / 2.396 | 0.910 / 2.416 |
| FCP (T-ImageNet) | 0.840 / 28.625 | 0.830 / 28.192 | 0.833 / 28.340 | 0.821 / 27.140 | 0.8308 / 28.7507 |
| Rob-FCP (T-ImageNet) | 0.913 / 45.872 | 0.910 / 44.972 | 0.898 / 42.571 | 0.887 / 41.219 | 0.898 / 43.298 |
| FCP (PathMNIST) | 0.850 / 1.106 | 0.839 / 1.065 | 0.837 / 1.055 | 0.839 / 1.065 | 0.832 / 1.043 |
| Rob-FCP (PathMNIST) | 0.895 / 1.311 | 0.900 / 1.355 | 0.900 / 1.355 | 0.899 / 1.354 | 0.901 / 1.363 |

Table 3: Marginal coverage / average set size on SHHS with non-IID data partition based on different attributes: wake time, N1, N2, N3, REM. The evaluation is done under different Coverage attack with $40\%$ ($K_m/K = 40\%$) malicious clients. The desired marginal coverage is 0.9.

|  | wake time | N1 | N2 | N3 | REM |
|---|---|---|---|---|---|
| FCP (SHHS) | 0.835 / 1.098 | 0.841 / 1.104 | 0.841 / 1.104 | 0.837 / 1.105 | 0.840 / 1.107 |
| Rob-FCP (SHHS) | 0.901 / 1.367 | 0.902 / 1.358 | 0.902 / 1.355 | 0.902 / 1.375 | 0.900 / 1.356 |

Table 4: Runtime of RobFCP quantile computation with $40\%$ malicious clients.

|  | MNIST | CIFAR-10 | Tiny-ImageNet | SHHS | PathMNIST |
|---|---|---|---|---|---|
| Runtime (seconds) | 0.5284 | 0.5169 | 0.5563 | 0.2227 | 0.3032 |

**Results with different non-IID data construction**. We evaluate Rob-FCP with different imbalance levels $\beta$ to show the effectiveness of Rob-FCP under different non-IID settings. The results in Table 2 demonstrate that under multiple degrees of data imbalance in non-IID federated conformal prediction, Rob-FCP consistently outperforms FCP in achieving a nominal marginal coverage level in the existence of Byzantine clients. We also consider alternative approaches to construct non-IID data with demographic differences in federated learning. The results in Table 3 show that in this type of non-IID partition, Rob-FCP still demonstrates robustness and effectiveness compared to FCP.

**Runtime of Rob-FCP**. We evaluate the runtime of quantile computation in Rob-FCP in Table 4, which indicates the efficiency of federated conformal prediction with our method.

**Results with an overestimate or underestimate of the number of malicious clients.** In Table 5, we provided evaluations of Rob-FCP with incorrect numbers of malicious clients. The results show that either underestimated numbers or overestimated numbers would harm the performance to different extents. Specifically, an underestimate of the number of malicious clients will definitely lead to the inclusion of malicious clients in the identified set $\mathcal{B}$ and downgrade the quality of conformal prediction. On the other hand, an overestimated number will lead to the exclusion of some benign clients. The neglect of non-conformity scores of those clients will lead to a distribution shift from the true data distribution in the calibration process, breaking the data exchangeability assumption of conformal prediction, and a downgraded performance. Therefore, correctly estimating the num-

Table 5: Marginal coverage / average set size under different Coverage attack with underestimated and overestimated numbers of malicious clients on TinyImageNet. The true ratio of malicious clients is $40\%$ ($K_m/K = 25\%$), while we evaluate Rob-FCP with different ratios of malicious clients $K'_m/K$ ranging from $5\%$ to $45\%$. The desired marginal coverage is $0.9$.

| $K'_m/K$ | 5% | 10% | 15% | 20% | 25% | 30% | 35% | 40% | 45% |
|---|---|---|---|---|---|---|---|---|---|
| Coverage | 0.8682 | 0.8756 | 0.8812 | 0.8884 | 0.9078 | 0.8936 | 0.8921/ | 0.8834 | 0.8803 |
| Set Size | 35.875 | 37.130 | 38.372 | 40.643 | 44.578 | 42.173 | 42.023 | 38.346 | 38.023 |

ber of malicious clients is of significance, and this is why we propose the malicious client number estimator, which is sound both theoretically and empirically to achieve the goal.

**The benign setting.** The benign conformal prediction performance (marginal coverage / average set size) without any malicious clients is provided in Table 6. As expected, the coverage of the prediction sets is very close to the target (0.9). In the non-IID setting, the predictive performance of the base global model is typically worse, leading to a larger average size of the prediction sets.

**Byzantine robustness of Rob-FCP in the IID and non-IID settings with known $K_m$.** We evaluate the marginal coverage and average set size of Rob-FCP under coverage, efficiency, and Gaussian attack and compare the results with the baseline FCP. We present results of FCP and Rob-FCP in existence of $10\%, 20\%, 30\%$ ($K_m/K = 10\%, 20\%, 30\%$) malicious clients on MNIST, CIFAR-10, Tiny-ImageNet (T-ImageNet), SHHS, and PathMNIST in Table 7. The coverage of FCP deviates drastically from the desired coverage level 0.9 under Byzantine attacks, along with a deviation from the benign set size. In contrast, Rob-FCP achieves comparable marginal coverage and average set size in both IID and non-IID settings.

**Byzantine robustness of Rob-FCP in the IID and non-IID settings with unknown $K_m$.** Similar to above, we evaluate the marginal coverage and average set size of Rob-FCP under verious attacks and compare the results with the FCP. We present results in existence of $10\%, 20\%, 30\%, 40\%$ ($K_m/K = 10\%, 20\%, 30\%, 40\%$) malicious clients in Table 8, where the number of the malicious clients is unknown to the algorithm. Again, the coverage of FCP as well as the size of the prediction set deviates drastically from the benign set setting, but Rob-FCP achieves comparable marginal coverage and average set size in both IID and non-IID settings.

**Robustness of Rob-FCP with different conformity scores.** Besides applying LAC nonconformity scores, we also evaluate Rob-FCP with APS scores (Romano et al., 2020). The results in Figures 5 to 10 demonstrate the Byzantine robustness of Rob-FCP with APS scores.

**Ablation study of different conformity score distribution characterization.** One key step in Rob-FCP is to characterize the conformity score distribution based on empirical observations. We adopt the histogram statistics approach as Equation (4). Rob-FCP also flexibly allows for alternative approaches to characterizing the empirical conformity score samples with a real-valued vector $\mathbf{v}$. We can fit a parametric model (e.g., Gaussian model) to the empirical scores and concatenate the parameters as the characterization vector $\mathbf{v}$. Another alternative is to characterize the score samples with exemplars approximated by clustering algorithms such as KMeans. We empirically compare different approaches in Figures 11 and 12 and show that the histogram statistics approach achieves the best performance.

**Ablation study of the distance measurement.** In Rob-FCP, we need to compute the distance between characterization vectors with measurement $d(\cdot, \cdot)$. We evaluate Rob-FCP with $\ell_1, \ell_2, \ell_\infty$-norm based vector distance as Equation (5) and an alternative Cosine similarity in Figure 13. The results show that the effectiveness of Rob-FCP is agnostic to these commonly used distance measurements. We adopt $\ell_2$-norm vector distance for consistency across evaluations.

Table 6: Benign conformal prediction results (marginal coverage / average set size) without any malicious clients.

| Dataset | IID setting | non-IID setting |
|---|---|---|
| MNIST | 0.898 / 0.900 | 0.902 / 1.828 |
| CIFAR-10 | 0.901 / 1.597 | 0.898 / 2.308 |
| Tiny-ImageNet | 0.901 / 21.92 | 0.899 / 42.35 |
| SHHS | 0.898 / 1.352 | 0.897 / 1.351 |
| PathMNIST | 0.904 / 1.242 | 0.901 / 1.361 |

Table 7: Marginal coverage / average set size under different Byzantine attacks with 10%, 20%, and 30% malicious clients. Rob-FCP consistently recovers the coverage (and average size of prediction set) of benign conformal prediction (Table 6), while the performance of FCP generally deteriorates as the percentage of malicious clients increases.

| Attack Method | Coverage Attack FCP | Coverage Attack Rob-FCP | Efficiency Attack FCP | Efficiency Attack Rob-FCP | Gaussian Attack FCP | Gaussian Attack Rob-FCP |
|---|---|---|---|---|---|---|
| **$K_m/K = 10\%$** | | | | | | |
| MNIST (IID) | 0.896 / 0.898 | **0.899 / 0.900** | 0.999 / 4.034 | **0.904 / 0.909** | 0.947 / 0.960 | **0.905 / 0.910** |
| CIFAR-10 (IID) | 0.887 / 1.499 | **0.900 / 1.588** | 1.000 / 7.991 | **0.892 / 1.556** | 0.906 / 1.633 | **0.892 / 1.565** |
| T-ImageNet (IID) | 0.873 / 18.44 | **0.901 / 22.36** | 0.999 / 148.7 | **0.895 / 21.28** | 0.916 / 23.98 | **0.909 / 23.80** |
| SHHS (IID) | 0.889 / 1.303 | **0.900 / 1.359** | 0.999 / 5.338 | **0.900 / 1.359** | 0.909 / 1.409 | **0.900 / 1.360** |
| PathMNIST (IID) | 0.892 / 1.184 | **0.905 / 1.249** | 1.000 / 6.271 | **0.902 / 1.235** | 0.941 / 1.504 | **0.903 / 1.240** |
| MNIST (non-IID) | 0.892 / 1.747 | **0.897 / 1.813** | 1.000 / 9.319 | **0.896 / 1.813** | 0.892 / 1.798 | **0.902 / 1.794** |
| CIFAR-10 (non-IID) | 0.887 / 1.209 | **0.894 / 2.287** | 1.000 / 8.808 | **0.908 / 2.347** | 0.918 / 2.515 | **0.911 / 2.378** |
| T-ImageNet (non-IID) | 0.892 / 41.03 | **0.905 / 44.81** | 0.997 / 146.7 | **0.902 / 44.29** | 0.917 / 47.47 | **0.900 / 44.74** |
| SHHS (non-IID) | 0.889 / 1.304 | **0.900 / 1.358** | 1.000 / 5.981 | **0.900 / 1.359** | 0.909 / 1.412 | **0.901 / 1.361** |
| PathMNIST (non-IID) | 0.892 / 1.290 | **0.902 / 1.361** | 0.996 / 5.149 | **0.900 / 1.348** | 0.938 / 1.739 | **0.904 / 1.374** |
| **$K_m/K = 20\%$** | | | | | | |
| MNIST (IID) | 0.873 / 0.876 | **0.893 / 0.897** | 1.000 / 10.00 | **0.895 / 0.899** | 0.967 / 0.988 | **0.900 / 0.905** |
| CIFAR-10 (IID) | 0.869 / 1.398 | **0.888 / 1.532** | 1.000 / 10.00 | **0.913 / 1.659** | 0.916 / 1.725 | **0.903 / 1.633** |
| T-ImageNet (IID) | 0.874 / 17.787 | **0.900 / 22.23** | 1.000 / 200.0 | **0.903 / 22.50** | 0.908 / 23.12 | **0.904 / 22.94** |
| SHHS (IID) | 0.876 / 1.243 | **0.900 / 1.359** | 1.000 / 5.984 | **0.900 / 1.356** | 0.918 / 1.467 | **0.900 / 1.360** |
| PathMNIST (IID) | 0.880 / 1.134 | **0.905 / 1.251** | 1.000 / 8.335 | **0.904 / 1.244** | 0.983 / 2.434 | **0.903 / 1.236** |
| MNIST (non-IID) | 0.857 / 1.534 | **0.896 / 1.765** | 1.000 / 9.089 | **0.902 / 1.836** | 0.915 / 1.945 | **0.912 / 1.904** |
| CIFAR-10 (non-IID) | 0.866 / 2.038 | **0.896 / 2.314** | 1.000 / 10.00 | **0.908 / 2.366** | 0.938 / 2.895 | **0.892 / 2.256** |
| T-ImageNet (non-IID) | 0.860 / 33.99 | **0.902 / 44.69** | 1.000 / 199.0 | **0.904 / 44.72** | 0.922 / 49.44 | **0.912 / 48.27** |
| SHHS (non-IID) | 0.874 / 1.236 | **0.901 / 1.363** | 1.000 / 5.985 | **0.901 / 1.363** | 0.917 / 1.463 | **0.900 / 1.358** |
| PathMNIST (non-IID) | 0.876 / 1.210 | **0.901 / 1.355** | 1.000 / 7.395 | **0.902 / 1.366** | 0.980 / 2.905 | **0.900 / 1.348** |
| **$K_m/K = 30\%$** | | | | | | |
| MNIST (IID) | 0.851 / 0.854 | **0.908 / 0.914** | 1.000 / 10.00 | **0.911 / 0.917** | 0.977 / 1.009 | **0.900 / 0.905** |
| CIFAR-10 (IID) | 0.852 / 1.307 | **0.895 / 1.583** | 1.000 / 10.00 | **0.894 / 1.563** | 0.909 / 1.672 | **0.903 / 1.602** |
| T-ImageNet (IID) | 0.862 / 15.66 | **0.904 / 22.61** | 1.000 / 200.0 | **0.907 / 22.85** | 0.907 / 23.89 | **0.906 / 24.15** |
| SHHS (IID) | 0.859 / 1.176 | **0.901 / 1.364** | 1.000 / 6.000 | **0.900 / 1.356** | 0.926 / 1.526 | **0.900 / 1.359** |
| PathMNIST (IID) | 0.863 / 1.064 | **0.906 / 1.252** | 1.000 / 9.000 | **0.903 / 1.241** | 1.000 / 6.531 | **0.906 / 1.255** |
| MNIST (non-IID) | 0.849 / 1.451 | **0.913 / 1.890** | 1.000 / 10.00 | **0.875 / 1.650** | 0.925 / 2.010 | **0.919 / 1.958** |
| CIFAR-10 (non-IID) | 0.844 / 1.870 | **0.900 / 2.294** | 1.000 / 10.00 | **0.912 / 2.408** | 0.950 / 3.152 | **0.901 / 2.327** |
| T-ImageNet (non-IID) | 0.864 / 33.41 | **0.895 / 43.12** | 1.000 / 200.0 | **0.906 / 43.46** | **0.923 / 52.23** | 0.932 / 55.78 |
| SHHS (non-IID) | 0.857 / 1.169 | **0.900 / 1.358** | 1.000 / 6.000 | **0.900 / 1.358** | 0.927 / 1.530 | **0.898 / 1.350** |
| PathMNIST (non-IID) | 0.860 / 1.141 | **0.900 / 1.344** | 1.000 / 9.000 | **0.903 / 1.368** | 1.000 / 6.287 | **0.903 / 1.373** |

Table 8: Marginal coverage / average set size under different Byzantine attacks with 10%, 20%, 30% and 40% malicious clients with unknown numbers of malicious clients. Rob-FCP consistently recovers the coverage (and average size of prediction set) of benign conformal prediction (Table 6), while the performance of FCP generally deteriorates as the percentage of malicious clients increases.

| Attack Method | Coverage Attack FCP | Rob-FCP | Efficiency Attack FCP | Rob-FCP | Gaussian Attack FCP | Rob-FCP |
|---|---|---|---|---|---|---|
| $K_m/K = 10\%$ | | | | | | |
| **IID** | | | | | | |
| MNIST | 0.896 / 0.898 | **0.901 / 0.905** | 0.999 / 4.034 | **0.890 / 0.895** | 0.947 / 0.960 | **0.895 / 0.900** |
| CIFAR-10 | 0.887 / 1.499 | **0.903 / 1.612** | 1.000 / 7.991 | **0.920 / 1.689** | 0.906 / 1.633 | **0.890 / 1.543** |
| T-ImageNet | 0.873 / 18.44 | **0.908 / 22.52** | 0.999 / 148.7 | **0.890 / 20.93** | 0.916 / 23.98 | **0.897 / 21.64** |
| SHHS | 0.889 / 1.303 | **0.902 / 1.365** | 0.999 / 5.338 | **0.903 / 1.368** | 0.909 / 1.409 | **0.902 / 1.367** |
| PathMNIST | 0.892 / 1.184 | **0.899 / 1.237** | 1.000 / 6.271 | **0.905 / 1.253** | 0.905 / 1.253 | **0.901 / 1.239** |
| **non-IID** | | | | | | |
| MNIST | 0.892 / 1.747 | **0.895 / 1.798** | 1.000 / 9.319 | **0.900 / 1.780** | 0.892 / 1.798 | **0.896 / 1.800** |
| CIFAR-10 | 0.887 / 1.209 | **0.890 / 2.221** | 1.000 / 8.808 | **0.900 / 2.304** | 0.918 / 2.515 | **0.905 / 2.418** |
| T-ImageNet | 0.892 / 41.03 | **0.903 / 43.94** | 0.997 / 146.7 | **0.898 / 43.01** | 0.917 / 47.47 | **0.915 / 47.35** |
| SHHS | 0.889 / 1.304 | **0.902 / 1.367** | 1.000 / 5.981 | **0.902 / 1.364** | 0.909 / 1.412 | **0.900 / 1.357** |
| PathMNIST | **0.892** / 1.290 | 0.909 / **1.394** | 0.996 / 5.149 | **0.901 / 1.376** | **0.905** / 1.387 | 0.907 / **1.375** |
| $K_m/K = 20\%$ | | | | | | |
| **IID** | | | | | | |
| MNIST | 0.873 / 0.876 | **0.898 / 0.903** | 1.000 / 10.00 | **0.906 / 0.912** | 0.967 / 0.988 | **0.904 / 0.908** |
| CIFAR-10 | 0.869 / 1.398 | **0.888 / 1.512** | 1.000 / 10.00 | **0.902 / 1.603** | 0.916 / 1.725 | **0.905 / 1.623** |
| T-ImageNet | 0.874 / 17.787 | **0.904 / 22.47** | 1.000 / 200.0 | **0.907 / 22.76** | 0.908 / 23.12 | **0.904 / 22.88** |
| SHHS | 0.876 / 1.243 | **0.902 / 1.365** | 1.000 / 5.984 | **0.902 / 1.366** | 0.918 / 1.467 | **0.902 / 1.363** |
| PathMNIST | 0.880 / 1.134 | **0.900 / 1.229** | 1.000 / 8.335 | **0.902 / 1.241** | 0.909 / 1.273 | **0.898 / 1.229** |
| **non-IID** | | | | | | |
| MNIST | 0.857 / 1.534 | **0.901 / 1.832** | 1.000 / 9.089 | **0.881 / 1.713** | 0.915 / 1.945 | **0.908 / 1.889** |
| CIFAR-10 | 0.866 / 2.038 | **0.900 / 2.344** | 1.000 / 10.00 | **0.897 / 2.312** | 0.938 / 2.895 | **0.929 / 2.702** |
| T-ImageNet | 0.860 / 33.99 | **0.905 / 44.38** | 1.000 / 199.0 | **0.894 / 42.30** | 0.922 / 49.44 | **0.906 / 46.38** |
| SHHS | 0.874 / 1.236 | **0.901 / 1.362** | 1.000 / 5.985 | **0.903 / 1.369** | 0.917 / 1.463 | **0.902 / 1.365** |
| PathMNIST | 0.876 / 1.210 | **0.907 / 1.388** | 1.000 / 7.395 | **0.903 / 1.362** | 0.905 / 1.382 | **0.902 / 1.362** |
| $K_m/K = 30\%$ | | | | | | |
| **IID** | | | | | | |
| MNIST | 0.851 / 0.854 | **0.905 / 0.912** | 1.000 / 10.00 | **0.907 / 0.913** | 0.977 / 1.009 | **0.903 / 0.908** |
| CIFAR-10 | 0.852 / 1.307 | **0.904 / 1.612** | 1.000 / 10.00 | **0.891 / 1.544** | 0.909 / 1.672 | **0.903 / 1.578** |
| T-ImageNet | 0.862 / 15.66 | **0.902 / 21.92** | 1.000 / 200.0 | **0.903 / 22.19** | 0.907 / 23.89 | **0.906 / 23.77** |
| SHHS | 0.859 / 1.176 | **0.903 / 1.372** | 1.000 / 6.000 | **0.902 / 1.366** | 0.926 / 1.526 | **0.903 / 1.368** |
| PathMNIST | 0.863 / 1.064 | **0.902 / 1.239** | 1.000 / 9.000 | **0.898 / 1.221** | 0.907 / 1.263 | **0.905 / 1.246** |
| **non-IID** | | | | | | |
| MNIST | 0.849 / 1.451 | **0.920 / 1.947** | 1.000 / 10.00 | **0.900 / 1.779** | 0.925 / 2.010 | **0.911 / 1.943** |
| CIFAR-10 | 0.844 / 1.870 | **0.899 / 2.360** | 1.000 / 10.00 | **0.891 / 2.264** | 0.950 / 3.152 | **0.896 / 2.300** |
| T-ImageNet | 0.864 / 33.41 | **0.895 / 42.79** | 1.000 / 200.0 | **0.908 / 44.74** | 0.923 / 52.23 | **0.920 / 50.70** |
| SHHS | 0.857 / 1.169 | **0.902 / 1.368** | 1.000 / 6.000 | **0.904 / 1.374** | 0.927 / 1.530 | **0.903 / 1.370** |
| PathMNIST | 0.860 / 1.141 | **0.895 / 1.337** | 1.000 / 9.000 | **0.902 / 1.376** | 0.910 / 1.418 | **0.899 / 1.352** |
| $K_m/K = 40\%$ | | | | | | |
| **IID** | | | | | | |
| MNIST | 0.832 / 0.834 | **0.891 / 0.892** | 1.000 / 10.00 | **0.895 / 0.901** | 0.979 / 1.025 | **0.899 / 0.904** |
| CIFAR-10 | 0.831 / 1.189 | **0.913 / 1.666** | 1.000 / 10.00 | **0.902 / 1.608** | 0.916 / 1.733 | **0.905 / 1.612** |
| T-ImageNet | 0.830 / 12.97 | **0.888 / 21.45** | 1.000 / 200.0 | **0.905 / 22.99** | 0.918 / 25.69 | **0.903 / 23.42** |
| SHHS | 0.834 / 1.093 | **0.902 / 1.363** | 1.000 / 6.000 | **0.903 / 1.369** | 0.937 / 1.611 | **0.902 / 1.368** |
| PathMNIST | 0.840 / 0.997 | **0.901 / 1.246** | 1.000 / 9.000 | **0.898 / 1.237** | 0.914 / 1.302 | **0.899 / 1.250** |
| **non-IID** | | | | | | |
| MNIST | 0.805 / 1.284 | **0.911 / 1.929** | 1.000 / 10.00 | **0.910 / 1.906** | 0.941 / 2.227 | **0.929 / 2.084** |
| CIFAR-10 | 0.829 / 1.758 | **0.893 / 2.270** | 1.000 / 10.00 | **0.888 / 2.203** | 0.970 / 3.863 | **0.923 / 2.635** |
| T-ImageNet | 0.825 / 27.84 | **0.906 / 45.18** | 1.000 / 200.0 | **0.903 / 42.62** | 0.942 / 61.50 | **0.937 / 59.61** |
| SHHS | 0.835 / 1.095 | **0.902 / 1.364** | 1.000 / 6.000 | **0.904 / 1.375** | 0.937 / 1.609 | **0.903 / 1.371** |
| PathMNIST | 0.837 / 1.055 | **0.903 / 1.378** | 1.000 / 9.000 | **0.909 / 1.398** | 0.915 / 1.464 | **0.914 / 1.488** |

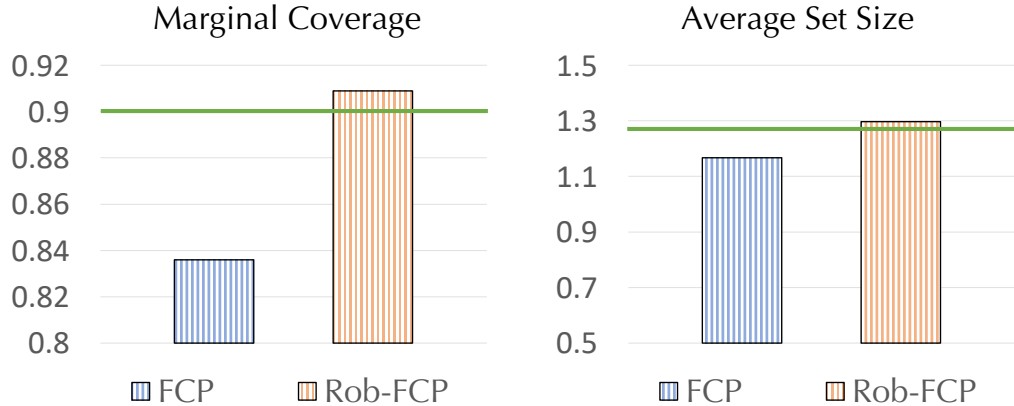

Figure 5: Marginal coverage / average set size under coverage attack with 40% malicious clients in the IID Byzantine setting on CIFAR-10. The green horizontal line represents the benign marginal coverage and average set size without any malicious clients.

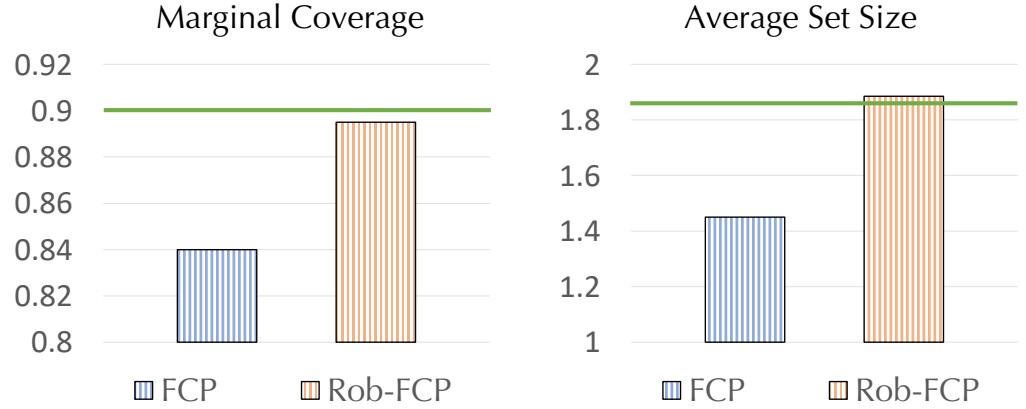

Figure 6: Marginal coverage / average set size under coverage attack with 40% malicious clients in the non-IID Byzantine setting on CIFAR-10. The green horizontal line represents the benign marginal coverage and average set size without any malicious clients.

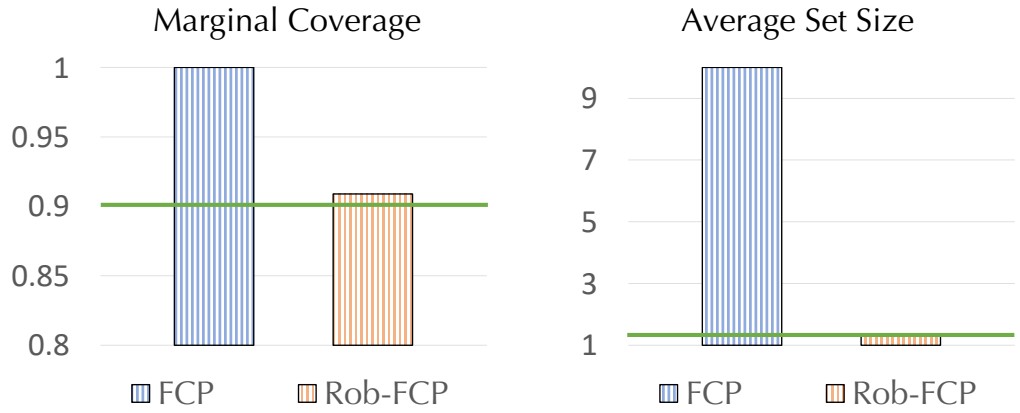

Figure 7: Marginal coverage / average set size under efficiency attack with 40% malicious clients in the IID Byzantine setting on CIFAR-10. The green horizontal line represents the benign marginal coverage and average set size without any malicious clients.

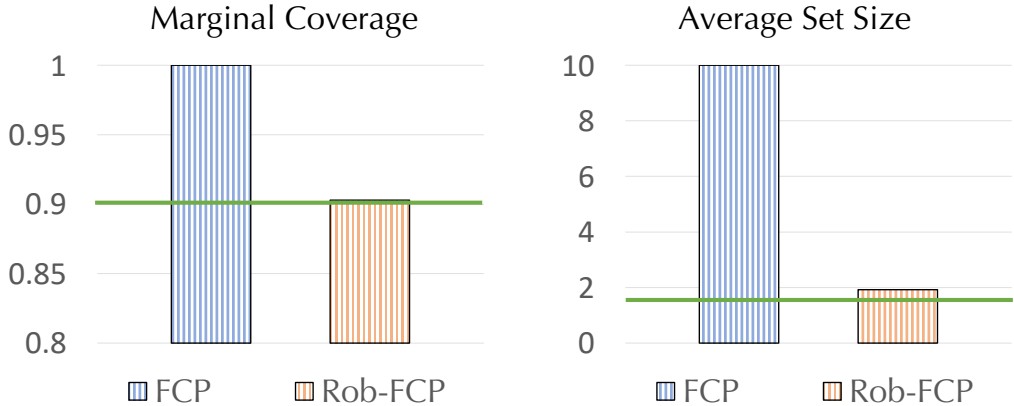

Figure 8: Marginal coverage / average set size under efficiency attack with 40% malicious clients in the non-IID Byzantine setting on CIFAR-10. The green horizontal line represents the benign marginal coverage and average set size without any malicious clients.

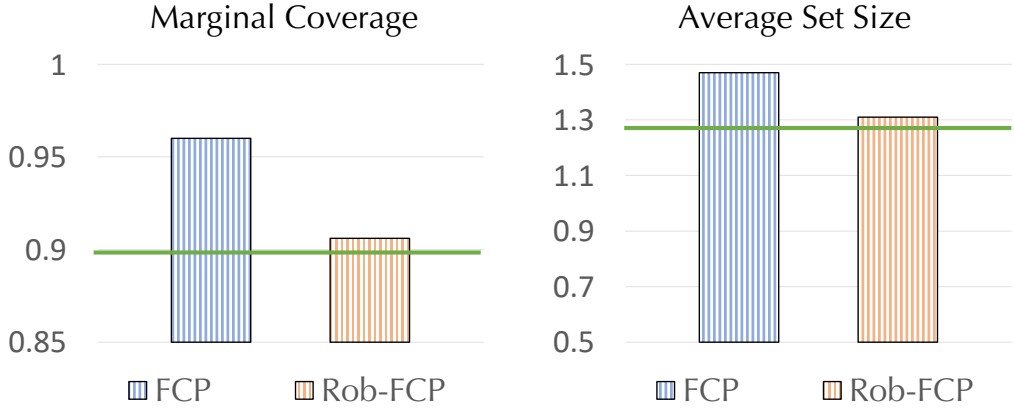

Figure 9: Marginal coverage / average set size under Gaussian attack with 40% malicious clients in the IID Byzantine setting on CIFAR-10. The green horizontal line represents the benign marginal coverage and average set size without any malicious clients.

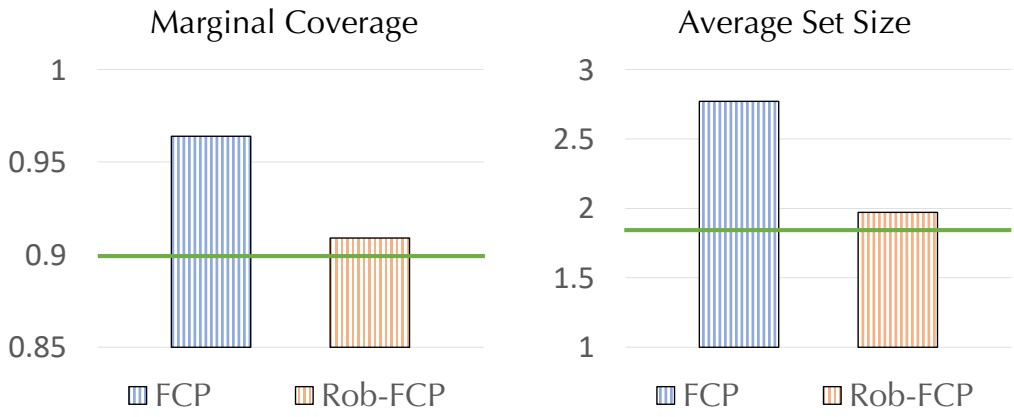

Figure 10: Marginal coverage / average set size under Gaussian attack with 40% malicious clients in the non-IID Byzantine setting on CIFAR-10. The green horizontal line represents the benign marginal coverage and average set size without any malicious clients.

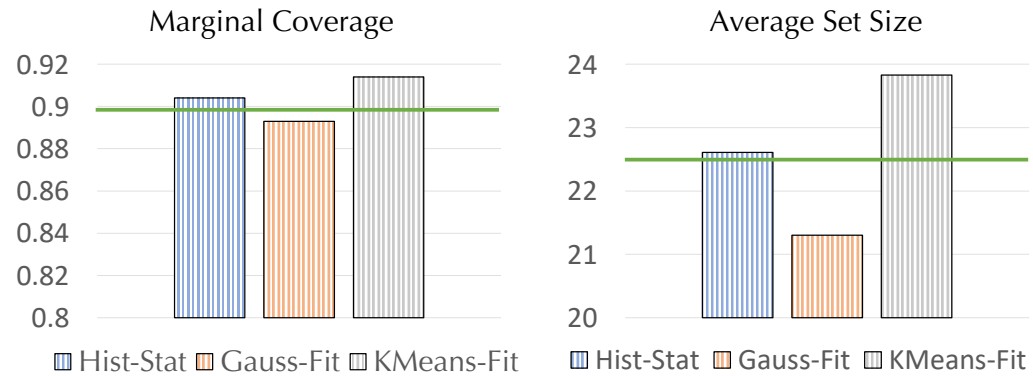

Figure 11: Marginal coverage / average set size under coverage attack with 40% malicious clients in the IID Byzantine setting on Tiny-ImageNet. The green horizontal line represents the benign marginal coverage and average set size without any malicious clients.

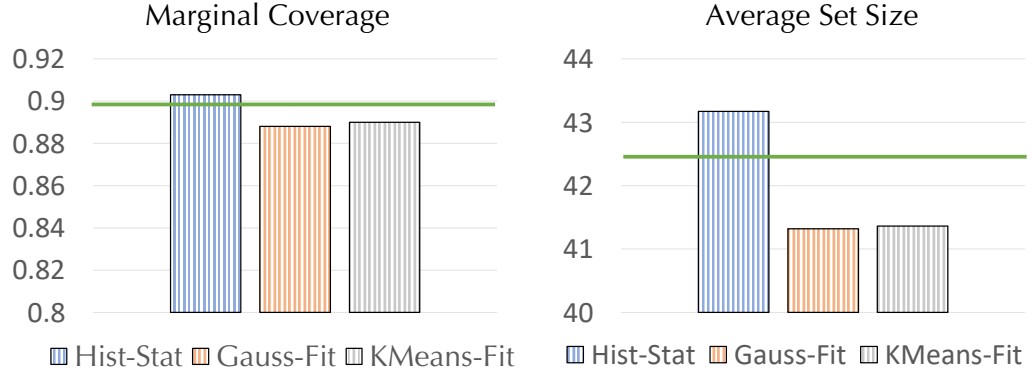

Figure 12: Marginal coverage / average set size under coverage attack with 40% malicious clients in the non-IID Byzantine setting on Tiny-ImageNet. The green horizontal line represents the benign marginal coverage and average set size without any malicious clients.

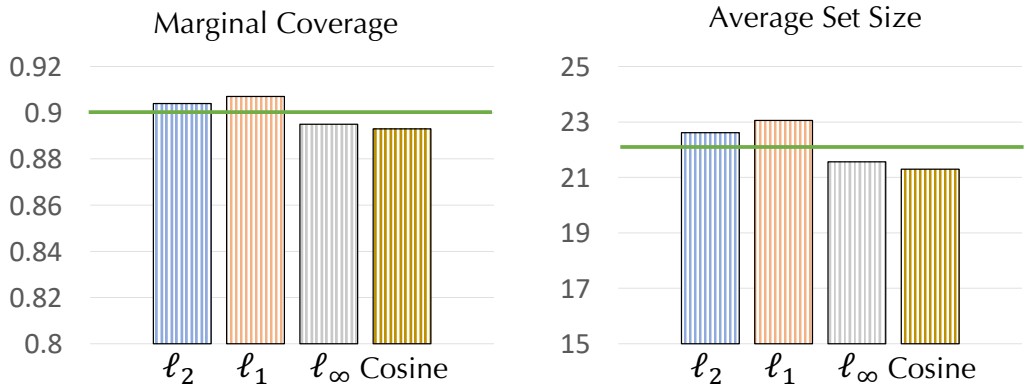

Figure 13: Marginal coverage / average set size under coverage attack with 40% malicious clients in the IID Byzantine setting on Tiny-ImageNet. The green horizontal line represents the benign marginal coverage and average set size without any malicious clients.

