# OpenReview forum: "Certifiably Byzantine-Robust Federated Conformal Prediction"
_ICLR.cc/2024/Conference — Submitted to ICLR 2024_

### Official Review · Reviewer_LS6w · 2023-10-29

**Soundness:** 3 good
**Presentation:** 3 good
**Contribution:** 3 good
**Rating:** 5
**Confidence:** 4

**Summary:**

The authors study a robust conformal prediction problem in a federated learning setting. Since local datasets share some privacy properties, some agents may not truthfully report their non-conformity score.  In this scenario, making a prediction interval based on this contaminated data would be harmful and lack the correct coverage probability. To solve such an issue, the authors propose a novel algorithm and obtain a coverage bound close to the desired level under both iid and non-iid settings.

**Strengths:**

1. The authors study a realistic setting where the agents may not be truthful in reporting their non-conformity score, and they developed an ``outlier detection" algorithm to ensure the prediction sets are made based on truthful agents.

2. Rigorous theoretical guarantees are provided on the coverage probability lower and upper bounds.

3. The methodology is applied to various types of datasets to measure its performance, which is great.

**Weaknesses:**

1. It seems the results are derived based on either knowing the number of truthful agents or being able to consistently estimate this quantity.  This could be a relatively strong assumption in reality.

2. The assumption on bounded distribution disparity could be relatively strong. There would be the cases where the underlying distributions of different group of agents are not the same. In that case, this assumption can be easily violated.

**Questions:**

1. Could the authors establish some theoretical guarantees where the number of truthful agents is not correctly estimated? What will happen if they are underestimated or overestimated?

2. Could the authors also provide some numerical results to illustrate the sensitivity if the number of truthful agents is not correctly identified?

3. Again, as I mentioned in the weakness part, if the distributions of underlying agents are no-i.i.d., it is very likely that assumption 3.1 will be violated, even if the agents report the true non-conformity score. It seems the current algorithm may not handle this point very well.

---

> ### Author Response · Authors · 2023-11-21
> **Author response to Reviewer LS6w [1/3]**
>
> > Q1: It seems the results are derived based on either knowing the number of truthful agents or being able to consistently estimate this quantity. Could the authors establish some theoretical guarantees where the number of truthful agents is not correctly estimated? What will happen if they are underestimated or overestimated?
>
> We thank you for the valuable comment. If the number of benign clients is overestimated, then there must be a portion of malicious clients in the output set of RobFCP $\mathcal{B}$. Those malicious clients will participate in the conformal calibration and distort the computed quantile value. In Theorem 6 in Appendix F, we added the analysis of this scenario. The theoretical results show that (1) compared to the bound with the correct number $K\_b$, the overestimate $K’\_b$ asymptotically looses the bound by a factor $1-K\_b/K’\_b$, and (2) a larger overestimate $K’\_b$ leads to worse coverage guarantees. If the number of benign clients is underestimated, we can always find a benign client not in the set $\mathcal{B}$, which makes the original proof smooth. Therefore, Theorem 1 and Corollary 1 still hold by plugging a smaller $K’\_b$ into the bound, but the coverage guarantee also becomes worse due the monotonicity regarding the number of benign clients $K\_b$.
>
> > Q2: The assumption on bounded distribution disparity could be relatively strong. There would be cases where the underlying distributions of different group of agents are not the same. In that case, this assumption can be easily violated.
>
> We thank you for your valuable comments and suggestions. We added clarifications and improvements of Rob-FCP for the non-IID setting from both the theoretical perspective and the evaluation perspective.
>
> From the perspective of theoretical analysis, we provide the coverage guarantees of Rob-FCP in the non-IID setting in Corollary 1 based on the assumption of bounded disparity among benign clients in Assumption 3.1. We added clarifications that the assumption is quite standard, typical, and not simplified across the literature of non-IID federated learning analysis [1] and Byzantine analysis [2,3]. Concretely, Li et al. [1] quantify the non-iid degree (i.e., disparity) of local loss of clients with $\Gamma$ in Section 3.1 of [1] and accordingly provide the convergence guarantee of federated optimization in Theorem 1 of [1], which is a function of the disparity $\Gamma$. Park et al. [2] assume a bounded difference $\rho_1$ between the local loss of benign clients and global loss in Assumption 2 of [2] and provide the convergence analysis in federated Byzantine optimization in Theorem 1 of [2] as a function of $\rho_1$. Data et al. [3] have an assumption of bounded gradient dissimilarity $\kappa$ in Assumption 2 of [3] and prove a convergence guarantee in Theorem 1 of [3] regarding $\kappa$. Similarly, in our paper, we assume a bounded disparity of score statistics quantified with $\sigma$ in Assumption 3.1 and accordingly provide the coverage guarantees of Rob-FCP in the non-IID setting in Corollary 1 as a function of $\sigma$. As we can see, the type of assumption is quite standard in the analysis setup of the line of non-IID analysis and Byzantine analysis. The assumption is also essential since without a quantity to restrict the disparity in local behavior, the local behavior can vary arbitrarily, and we can not control global performance quantitatively. From the results in Corollary 1, we can also see that as long as the disparity bound $\sigma$ is finite, the coverage bound in the non-IID setting is asymptotically valid and tight with sufficiently large benign sample sizes $n_b$. We added these clarifications and details in Section 3.3 in the revised manuscript.

---

> ### Author Response · Authors · 2023-11-21
> **Author response to Reviewer LS6w [2/3]**
>
> > To continue the answer of Q1
>
> From the evaluation perspective, we added more clarifications and illustrations on how we construct non-IID data of different levels and also added more evaluations on various approaches to non-IID data construction. We first added clarifications that we followed the standard evaluation setup of non-IID federated learning by sampling different label ratios for different clients from the Dirichlet distribution as the literature [4,5,6,7]. Concretely, we sample $p_{c,j} \sim \text{Dir}(\beta)$ and allocate a $p_{c,j}$ proportion of the instances with class $c$ to the client $j$. Here $\text{Dir}(·)$ denotes the Dirichlet distribution and $\beta$ is a concentration parameter ($\beta$ > 0). Dirichlet distribution is commonly used as the prior distribution in Bayesian statistics and is an appropriate
> choice to simulate real-world data distribution. An advantage of this approach is that we can flexibly change the imbalance level by varying the concentration parameter $\beta$. If $\beta$
> is set to a smaller value, then the partition is more unbalanced. We fixed the imbalance level $\beta$ as $0.5$ in the evaluations before the rebuttal period. Following the suggestion, we added evaluations with different imbalance levels $\beta$ to show the effectiveness of Rob-FCP under different non-IID settings. The results in the following Table 1 demonstrate that under multiple degrees of data imbalance in non-IID federated conformal prediction, Rob-FCP consistently outperforms FCP in achieving a nominal marginal coverage level in the existence of Byzantine clients. We also consider alternative approaches to construct non-IID data with demographic differences in federated learning. Concretely, we split the SHHS dataset by sorting five different attributes: wake time, N1, N2, N3, REM. For example, we first sort the instances according to the wake time of patients and then assign them to 100 clients in sequence. Therefore, some clients are assigned patients with an early wake time, and some are assigned instances with late wake time. The results in Table 2 show that in this type of non-IID partition, Rob-FCP still demonstrates robustness and effectiveness compared to FCP.
>
> Table 1. Marginal coverage / average set size with different non-IID imbalance level $\beta$, the parameter of Dirichlet distribution where the label ratios of clients are sampled from. The evaluation is done under different Coverage attack with $40\%~(K_m/K=40\%)$ malicious clients. The desired marginal coverage is $0.9$.
>
> | | $\beta=0.1$ | $\beta=0.3$ | $\beta=0.5$ | $\beta=0.7$ | $\beta=0.9$ |
> | - | - | - | - | - | - |
> | FCP on MNIST | 0.780 / 1.173 |  0.817 / 1.318 | 0.833 / 1.384 | 0.805 / 1.265 |  0.828 / 1.363 |
> | Rob-FCP on MNIST | **0.899** / 1.806 | **0.905** / 1.809 | **0.903** / 1.827 |  **0.898** / 1.781 | **0.893** / 1.768 |
> | FCP on CIFAR-10 | 0.806 / 1.641 | 0.821 / 1.717 | 0.836 / 1.791 | 0.823 / 1.744 | 0.824 / 1.723 |
> | Rob-FCP on CIFAR-10 | **0.899** / 2.260 |  **0.907** / 2.405 | **0.892** / 2.243 | **0.904** / 2.396 | **0.910** / 2.416 |
> | FCP on Tiny-ImageNet | 0.840 / 28.625 | 0.830 / 28.192 |  0.833 / 28.340 |  0.821 / 27.140 | 0.8308 / 28.751 |
> | Rob-FCP on Tiny-ImageNet | **0.913** / 45.872 | **0.910** / 44.972 | **0.898** / 42.571 | **0.887** / 41.219 | **0.898** / 43.298 |
> | FCP on PathMNIST | 0.850 / 1.106 | 0.839 / 1.065 | 0.837 / 1.055 | 0.839 / 1.065	 | 0.832 / 1.043 |
> | Rob-FCP on PathMNIST | **0.895** / 1.311 | **0.900** / 1.355 | **0.900** / 1.355 | **0.899** / 1.354 | **0.901** / 1.363 |
>
>
> Table 2. Marginal coverage / average set size on SHHS with non-IID data partition based on different attributes: wake time, N1, N2, N3, REM. The evaluation is done under different Coverage attack with $40\%~(K_m/K=40\%)$ malicious clients. The desired marginal coverage is $0.9$.
>
> | | wake time | N1 | N2 | N3 | REM |
> | - | - | - | - | - | - |
> | FCP on SHHS | 0.835 / 1.098 | 0.841 / 1.104 | 0.841 / 1.104 | 0.837 / 1.105 | 0.840 / 1.107 |
> | Rob-FCP on SHHS | *0.901* / 1.367 | *0.902* / 1.358 | *0.902* / 1.355 | *0.902* / 1.375 | *0.900* / 1.356 |
>
> *[1] Li, Xiang, et al. "On the convergence of fedavg on non-iid data." ICLR 2020.*
>
> *[2] Park, Jungwuk, et al. "Sageflow: Robust federated learning against both stragglers and adversaries." NeurIPS 2021.*
>
> *[3] Data, Deepesh, et al. "Byzantine-resilient high-dimensional SGD with local iterations on heterogeneous data." ICML 2021.*
>
> *[4] Yurochkin, Mikhail, et al. "Bayesian nonparametric federated learning of neural networks." International conference on machine learning. ICML 2019.*
>
> *[5] Lin, Tao, et al. "Ensemble distillation for robust model fusion in federated learning." NeurIPS 2020.*
>
> *[6] Wang, Hongyi, et al. "Federated learning with matched averaging." ICLR 2020.*
>
> *[7] Gao, Liang, et al. "Feddc: Federated learning with non-iid data via local drift decoupling and correction." CVPR 2022.*

---

> ### Author Response · Authors · 2023-11-21
> **Author response to Reviewer LS6w [3/3]**
>
> > Q3: Could the authors also provide some numerical results to illustrate the sensitivity if the number of truthful agents is not correctly identified?
>
> We thank you for the valuable suggestion. In the following Table 3, we provided evaluations of Rob-FCP with incorrect numbers of malicious clients. The results show that either underestimated numbers or overestimated numbers would harm the performance to different extents. Specifically, an underestimate of the number of malicious clients will definitely lead to the inclusion of malicious clients in the identified set $\mathcal{B}$ and downgrade the quality of conformal prediction. On the other hand, an overestimated number will lead to the exclusion of some benign clients. The neglect of non-conformity scores of those clients will lead to a distribution shift from the true data distribution in the calibration process, breaking the data exchangeability assumption of conformal prediction, and a downgraded performance. Therefore, correctly estimating the number of malicious clients is of significance, and this is why we propose the malicious client number estimator, which is sound both theoretically and empirically to achieve the goal.
>
> Table 3. Marginal coverage / average set size under different Coverage attack with underestimated and overestimated numbers of malicious clients on TinyImageNet. The true ratio of malicious clients is $40\%~(K_m/K=25\%)$, while we evaluate Rob-FCP with different ratios of malicious clients $K’_m/K$ ranging from $5\%$ to $45\%$. The desired marginal coverage is $0.9$.
>
> | | $K’_m/K=5\\%$ | $K’_m/K=10\\%$ | $K’_m/K=15\\%$ | $K’_m/K=20\\%$ | $K’_m/K=25\\%$ |  $K’_m/K=30\\%$ | $K’_m/K=35\\%$ | $K’_m/K=40\\%$ | $K’_m/K=45\\%$ |
> | - | - | - | - | - | - | - | - | - | - |
> | Marginal coverage / average set size | 0.8682 / 35.8754 | 0.8756 / 37.1307 | 0.8812 / 38.3729 | 0.8884 / 40.6433 | 0.9078 / 44.5781 | 0.8936 / 42.1735 | 0.8921 / 42.0232 | 0.8834 / 38.3463 | 0.8803 / 38.0232 |
>
>
> > Q4: If the distributions of underlying agents are no-i.i.d., it is very likely that assumption 3.1 will be violated, even if the agents report the true non-conformity score. It seems the current algorithm may not handle this point very well.
>
> We thank you for the interesting question. We can actually think about the effectiveness of Rob-FCP in the non-IID setting from two perspectives. (1) If the malicious clients attempt to disturb the quantile value of conformity scores to a large extent, they need to report a score vector far away from the cluster of score vectors of benign clients. Although the benign score vectors have some heterogeneity, the malicious score vector in this case is still distinguishable in the vector space and effectively excluded during the calibration. (2) If the malicious clients report score vectors close to the cluster of benign score vectors, then even if the Rob-FCP algorithm does not identify malicious clients exactly, the malicious score vectors can also characterize the structure of the benign score cluster, and thus, the computed quantile value would not deviate much. The two situations are exactly the important intuitions in the proof of Theorem 1 and Corollary 1, which rigorously formulates the coverage guarantee of Rob-FCP as a function of the non-IID degree $\sigma$.

---

### Official Review · Reviewer_bwYM · 2023-10-30

**Soundness:** 3 good
**Presentation:** 2 fair
**Contribution:** 2 fair
**Rating:** 5
**Confidence:** 4

**Summary:**

The paper studies federated conformal prediction (i.e. perform conformal prediction when the calibration dataset is distributed among multiple agents) where there may be Byzantine agents who may act maliciously. The paper shows how to detect such malicious behaviors by recording their deviation from the rest of the population; for each agent, they record the empirical histogram over H bins, calculate the average distance away from their closest neighbors where the distance is measured by the Lp distance between their H histogram bins, and mark agents with high distance away from their neighbors as malicious agents.

They show that under a few assumptions (iid or non-iid + some other assumptions), their algorithm can identify malicious agents and provide a target coverage guarantee (Theorem 1 and Corollary 1). In the case the number of malicious agents is not known, they show a way to estimate such number via some approximation (approximating the multinomial with multivariate normal).

Finally, they evaluate their algorithm on a few different datasets (Section 5).

**Strengths:**

-The idea to identify the malicious agents behavior via their deviation from the non-malicious agents in terms of the non-conformal score distribution seems novel.
-Their algorithm seem to perform quite well in their experiments.

**Weaknesses:**

-The main reason why the algorithm in the paper works seems to be due to the homogeneity of the non-malicious agents. Even in the experiments, the clients are partitioned randomly and hence their distributions will be pretty similar. However, as discussed even in the intro of the paper, there can be many settings where there is quite a bit of heterogeneity among the agents not due to their Byzantine and malicious behaviors but the underlying distributions are just inherently different. In fact, this heterogeneity seemed to me the motivation to studying this problem — i.e. the bolded sentences in the second paragraph of the intro. But the paper seems to study cases when there isn’t much heterogeneity?

**Questions:**

-It’s not clear to me what is exactly meant by the non-iid setting. Is it just that the agents’s non-conformal scores don’t come from the same distribution? However, the assumption that the v^(k) values (the histogram values) aren’t too different across agents in the non-iid setting essentially gives you an iid setting, right? I’m a little confused on the difference so it would be helpful to compare exactly the iid setting and the non-iid setting (along with other assumptions made for each setting) and how exactly they are really different.


-In proving the main result (Theorem 1), the appearance of the inverse of the CDF of the standard normal distribution seems surprising as there was no normality assumption before; it would be good to cite a reference for what is referred to as “the binomial proportion confidence interval” or eqn (13) in the proof of Theorem 1. Also, can’t one just apply the DKW inequality (https://en.wikipedia.org/wiki/Dvoretzky%E2%80%93Kiefer%E2%80%93Wolfowitz_inequality) here? It should be sufficient to show concentration of the empirical CDF (which exactly characterizes the empirical histogram values v(k)_h) toward the true CDF (which also characterizes the true bar(v) values). This would avoid the need to union bound over H values too as DKW tells you that over all possible h values (i.e. hth cut point), the empirical and the true CDF value is close with high probability.


 -How’s the final (1-alpha)-quantile being calculated after identifying the benign clients? Is it simply combining all the clients non-conformal scores altogether and then calculating the (1-alpha)-quantile? If that’s the case, I’m not understanding the federated nature of this problem except for the fact that some of the clients data are being ignored due to their apparent distributional difference to other clients?  How can one tell if this distributional difference of some clients is due to their malicious behavior or truly inherent distributional difference?

---

> ### Author Response · Authors · 2023-11-21
> **Author response to Reviewer bwYM [1/3]**
>
> > Q1: More clarification of the consideration of heterogeneity in analysis and evaluations.
>
> We thank you for your valuable comments and suggestions. We added clarifications and improvements of Rob-FCP for the non-IID setting from both the theoretical perspective and the evaluation perspective.
>
> From the perspective of theoretical analysis, we provide the coverage guarantees of Rob-FCP in the non-IID setting in Corollary 1 based on the assumption of bounded disparity among benign clients in Assumption 3.1. We added clarifications that the assumption is quite standard, typical, and not simplified across the literature of non-IID federated learning analysis [1] and Byzantine analysis [2,3]. Concretely, Li et al. [1] quantify the non-iid degree (i.e., disparity) of local loss of clients with $\Gamma$ in Section 3.1 of [1] and accordingly provide the convergence guarantee of federated optimization in Theorem 1 of [1], which is a function of the disparity $\Gamma$. Park et al. [2] assume a bounded difference $\rho_1$ between the local loss of benign clients and global loss in Assumption 2 of [2] and provide the convergence analysis in federated Byzantine optimization in Theorem 1 of [2] as a function of $\rho_1$. Data et al. [3] have an assumption of bounded gradient dissimilarity $\kappa$ in Assumption 2 of [3] and prove a convergence guarantee in Theorem 1 of [3] regarding $\kappa$. Similarly, in our paper, we assume a bounded disparity of score statistics quantified with $\sigma$ in Assumption 3.1 and accordingly provide the coverage guarantees of Rob-FCP in the non-IID setting in Corollary 1 as a function of $\sigma$. As we can see, the type of assumption is quite standard in the analysis setup of the line of non-IID analysis and Byzantine analysis. The assumption is also essential since without a quantity to restrict the disparity in local behavior, the local behavior can vary arbitrarily, and we can not control global performance quantitatively. From the results in Corollary 1, we can also see that as long as the disparity bound $\sigma$ is finite, the coverage bound in the non-IID setting is asymptotically valid and tight with sufficiently large benign sample sizes $n_b$. We added these clarifications and details in Section 3.3 in the revised manuscript.
>
> From the evaluation perspective, we added more clarifications and illustrations on how we construct non-IID data of different levels and also added more evaluations on various approaches to non-IID data construction. We first added clarifications that we followed the standard evaluation setup of non-IID federated learning by sampling different label ratios for different clients from the Dirichlet distribution as the literature [4,5,6,7]. Concretely, we sample $p_{c,j} \sim \text{Dir}(\beta)$ and allocate a $p_{c,j}$ proportion of the instances with class $c$ to the client $j$. Here $\text{Dir}(·)$ denotes the Dirichlet distribution and $\beta$ is a concentration parameter ($\beta$ > 0). Dirichlet distribution is commonly used as the prior distribution in Bayesian statistics and is an appropriate
> choice to simulate real-world data distribution. An advantage of this approach is that we can flexibly change the imbalance level by varying the concentration parameter $\beta$. If $\beta$
> is set to a smaller value, then the partition is more unbalanced. We fixed the imbalance level $\beta$ as $0.5$ in the evaluations before the rebuttal period. Following the suggestion, we added evaluations with different imbalance levels $\beta$ to show the effectiveness of Rob-FCP under different non-IID settings. The results in the following Table 1 demonstrate that under multiple degrees of data imbalance in non-IID federated conformal prediction, Rob-FCP consistently outperforms FCP in achieving a nominal marginal coverage level in the existence of Byzantine clients. We also consider alternative approaches to construct non-IID data with demographic differences in federated learning. Concretely, we split the SHHS dataset by sorting five different attributes: wake time, N1, N2, N3, REM. For example, we first sort the instances according to the wake time of patients and then assign them to 100 clients in sequence. Therefore, some clients are assigned patients with an early wake time, and some are assigned instances with late wake time. The results in Table 2 show that in this type of non-IID partition, Rob-FCP still demonstrates robustness and effectiveness compared to FCP.

---

> ### Author Response · Authors · 2023-11-21
> **Author response to Reviewer bwYM [2/3]**
>
> > Q1 (to continue): More clarification of the consideration of heterogeneity in analysis and evaluations.
>
> Table 1. Marginal coverage / average set size with different non-IID imbalance level $\beta$, the parameter of Dirichlet distribution where the label ratios of clients are sampled from. The evaluation is done under different Coverage attack with $40\%~(K_m/K=40\%)$ malicious clients. The desired marginal coverage is $0.9$.
>
> | | $\beta=0.1$ | $\beta=0.3$ | $\beta=0.5$ | $\beta=0.7$ | $\beta=0.9$ |
> | - | - | - | - | - | - |
> | FCP on MNIST | 0.780 / 1.173 |  0.817 / 1.318 | 0.833 / 1.384 | 0.805 / 1.265 |  0.828 / 1.363 |
> | Rob-FCP on MNIST | **0.899** / 1.806 | **0.905** / 1.809 | **0.903** / 1.827 |  **0.898** / 1.781 | **0.893** / 1.768 |
> | FCP on CIFAR-10 | 0.806 / 1.641 | 0.821 / 1.717 | 0.836 / 1.791 | 0.823 / 1.744 | 0.824 / 1.723 |
> | Rob-FCP on CIFAR-10 | **0.899** / 2.260 |  **0.907** / 2.405 | **0.892** / 2.243 | **0.904** / 2.396 | **0.910** / 2.416 |
> | FCP on Tiny-ImageNet | 0.840 / 28.625 | 0.830 / 28.192 |  0.833 / 28.340 |  0.821 / 27.140 | 0.8308 / 28.751 |
> | Rob-FCP on Tiny-ImageNet | **0.913** / 45.872 | **0.910** / 44.972 | **0.898** / 42.571 | **0.887** / 41.219 | **0.898** / 43.298 |
> | FCP on PathMNIST | 0.850 / 1.106 | 0.839 / 1.065 | 0.837 / 1.055 | 0.839 / 1.065	 | 0.832 / 1.043 |
> | Rob-FCP on PathMNIST | **0.895** / 1.311 | **0.900** / 1.355 | **0.900** / 1.355 | **0.899** / 1.354 | **0.901** / 1.363 |
>
>
> Table 2. Marginal coverage / average set size on SHHS with non-IID data partition based on different attributes: wake time, N1, N2, N3, REM. The evaluation is done under different Coverage attack with $40\%~(K_m/K=40\%)$ malicious clients. The desired marginal coverage is $0.9$.
>
> | | wake time | N1 | N2 | N3 | REM |
> | - | - | - | - | - | - |
> | FCP on SHHS | 0.835 / 1.098 | 0.841 / 1.104 | 0.841 / 1.104 | 0.837 / 1.105 | 0.840 / 1.107 |
> | Rob-FCP on SHHS | *0.901* / 1.367 | *0.902* / 1.358 | *0.902* / 1.355 | *0.902* / 1.375 | *0.900* / 1.356 |
>
> *[1] Li, Xiang, et al. "On the convergence of fedavg on non-iid data." ICLR 2020.*
>
> *[2] Park, Jungwuk, et al. "Sageflow: Robust federated learning against both stragglers and adversaries." NeurIPS 2021.*
>
> *[3] Data, Deepesh, et al. "Byzantine-resilient high-dimensional SGD with local iterations on heterogeneous data." ICML 2021.*
>
> *[4] Yurochkin, Mikhail, et al. "Bayesian nonparametric federated learning of neural networks." International conference on machine learning. ICML 2019.*
>
> *[5] Lin, Tao, et al. "Ensemble distillation for robust model fusion in federated learning." NeurIPS 2020.*
>
> *[6] Wang, Hongyi, et al. "Federated learning with matched averaging." ICLR 2020.*
>
> *[7] Gao, Liang, et al. "Feddc: Federated learning with non-iid data via local drift decoupling and correction." CVPR 2022.*
>
> > Q2: More clarifications of the comparisons between IID setting and non-IID setting.
>
> We consider the standard IID and non-IID settings in federated learning literature. Specifically, the data of local clients are sampled from the same data distribution in the IID setting, while the data of clients are sampled from different data distributions in the non-IID setting. In the context of federated conformal prediction, IID data samples among clients can imply the IID non-conformity scores of clients, which implies that the characterization vectors of scores are sampled from the same multinomial distribution. Similarly, non-IID data samples among clients can also imply the non-IID non-conformity scores of clients, which implies that the characterization vectors are sampled from different multinomial distributions. In Assumption 3.1, we quantify the maximum disparity of different multinomial distributions with $\sigma>0$, and a larger $\sigma$ can induce a larger non-IID degree. Note that $\sigma=0$ is equivalent to the IID setting because $\sigma=0$ implies that the parameters of multinomial distributions are the same: $\overline{\rm{v}}^{(k\_1)} = \overline{\rm{v}}^{(k\_2)}, \forall k\_1,k\_2$. To conclude, $\sigma=0$ in Assumption 3.1 formally defines the IID setting in the context of federated conformal prediction, while Assumption 3.1 with $\sigma>0$ defines the non-IID setting with non-IID degree $\sigma$.

---

> ### Author Response · Authors · 2023-11-21
> **Author response to Reviewer bwYM [3/3]**
>
> > Q3: Improvement of the concentration bound of the empirical CDF and true CDF.
>
> We thank you for the valuable suggestion! We added the underlying assumption of the binomial proportion confidence interval [1] in Theorem 1 and the proof. We also leveraged the more advanced DKW inequality [2] to bound the error between the empirical CDF and population CDF, and provided the results in Theorem 5 and Corollary 3 in Appendix E. With the DKW inequality, we can basically improve the concentration factor from $\dfrac{H\Phi^{-1}({1-\beta/2HK\_b})}{2 \sqrt{n\_b}}$ to $H\sqrt{\dfrac{\ln(2K\_b/\beta)}{2n\_b}} \left( 1+\dfrac{N\_m}{n\_b}\dfrac{2}{1-\tau} \right)$, which removes the necessity of applying union bound over $H$ subintervals. We referred to the results with DKW inequality in the remarks of Theorem 1 and Corollary 1 in the main text.
>
> *[1] Wallis, Sean. "Binomial confidence intervals and contingency tests: mathematical fundamentals and the evaluation of alternative methods." Journal of Quantitative Linguistics 20.3 (2013): 178-208.*
>
> *[2] Dvoretzky, Aryeh, Jack Kiefer, and Jacob Wolfowitz. "Asymptotic minimax character of the sample distribution function and of the classical multinomial estimator." The Annals of Mathematical Statistics (1956): 642-669.*
>
> > Q4: How is the final $(1-\alpha)$ quantile computed? Is it simply combining all the clients non-conformal scores altogether and then calculating the (1-alpha)-quantile?
>
> We added clarifications that we did not combine all the non-conformal scores of clients  $\\{s_j^{(k)}\\}~(k \in {1,2,..,K})$ because sending all the conformity scores to the server is expensive and leak much information about local data samples in the federated setting. Therefore, we characterize the scores $\\{s_j^{(k)}\\}$ with a histogram statistics vector $v^{(k)}$, which computes score frequency in $H$ equally partitioned subintervals. The characterization vector $v^{(k)}$ has a much smaller dimension than the universal scores $\\{s_j^{(k)}\\}$ ($H \ll n_k$) and thus leaks much less private information during the communication between the server and clients.
> The characterization vector is an approximation of the universal scores and can induce an approximation error of federated calibration, quantified by $\epsilon$ in Section 2.2. We also consider the influence of $\epsilon$ on the coverage guarantee in the Byzantine setting in Theorem 1 and Corollary 1, indicating a tradeoff between the tightness of conformal guarantee bound and the private information leakage/communication costs in federated settings.
>
> > Q5: How can one tell if this distributional difference of some clients is due to their malicious behavior or truly inherent distributional difference?
>
> We thank you for the interesting question. We can actually think about the effectiveness of Rob-FCP in the non-IID setting from two perspectives. (1) If the malicious clients attempt to disturb the quantile value of conformity scores to a large extent, they need to report a score vector far away from the cluster of score vectors of benign clients. Although the benign score vectors have some heterogeneity, the malicious score vector in this case is still distinguishable in the vector space and effectively excluded during the calibration. (2) If the malicious clients report score vectors close to the cluster of benign score vectors, then even if the Rob-FCP algorithm does not identify malicious clients exactly, the malicious score vectors can still characterize the structure of the benign score cluster, and thus, the computed quantile value would not deviate much. The two situations are exactly the important intuitions in the proof of Theorem 1 and Corollary 1, which rigorously formulates the coverage guarantee of Rob-FCP as a function of the non-IID degree $\sigma$.

---

### Official Review · Reviewer_eCJ4 · 2023-10-31

**Soundness:** 3 good
**Presentation:** 3 good
**Contribution:** 2 fair
**Rating:** 5
**Confidence:** 4

**Summary:**

This paper introduces an algorithm called Rob-FCP to perform Conformal Prediction in a federated learning setting where some agents are potentially malicious. This algorithm first discards the malicious agents and then performs a standard conformal prediction algorithm. Authors theoretically provide the conformal coverage bound of Rob-FCP in the Byzantine setting and show that the coverage of Rob-FCP is asymptotically close to the desired coverage level under mild conditions in both IID and non-IID settings. An algorithm for automatically determining the number of malicious agents is also provided. Finally, empirical experiments demonstrate that Rob-FCP is effective.

**Strengths:**

1\ This is a very interesting problem for the conformal prediction community.

2\ The paper treats both the iid setting and the non-iid setting.

3\ The paper is well-written, clear, and easy to follow.

4\ The experience shows that the method performs well in this federated learning setting with malicious agents.

**Weaknesses:**

1\ A major weakness is that to calculate the vector distance $d_{k_1, k_2}$ in step 8 of the algorithm ("Algorithm 1 Identifying the malicious client"), we need to send all the vectors $v^{(k)}$ to the server. It seems to me that this step is very problematic in a federated learning context.

2\ Another important weakness is that the bounds of Theorem 1 and Corollary 1 are in $1/(\min n_i)$. Therefore, if a non-malicious agent has only one data point, the bound does not improve, even if the other agents have an increasing number of data. This may just be due to an artifact of the proof or maybe we really can't do any better. This has to be proven (or at least discussed).

3\ Citations are not always appropriate. For instance, the split conformal prediction method is attributed to Lei et al., 2018 but the citation should be Papadopoulos et al., 2002.  The same applies to the FCP which does not cite the first paper due to Lu et al., 2021, and the more recent one Humbert et al., 2023, and to related work on federated learning, which only cites papers from 2019 and above.

4\ Although the experiences are well explained and in large quantities, I think that the parameters of the experiments are not always given (maybe I am wrong). For example, we do not know how malicious data are generated.

Minor:
1\ "marginal prediction coverage: ..." the definition is with an "inclusion"
2\ Mixture coefficients in the definition of $Q_{\lambda}$ are missing.
3\ Problem in the definition of $N_m$.
4\ In the abstract and introduction, the federated learning framework is also justified by privacy concerns. In general, it is not true that federated learning guarantees privacy.

**Questions:**

1\ Is it possible to compute the vector distance $d_{k_1, k_2}$ with a federated algorithm ?

2\ The $\min{n_i}$ in the bound of Theorem 1 and Corollary 1 cannot be improved or is it just an artifact of the proof?

3\ In Theorem 1 and Corollary 1, $\varepsilon$ appears in the bound. Is it possible to control it?

4\ In addition to the previous question, in the experiment how much time it takes to compute the quantile? (and for wich $\varepsilon$ ?)

5\ Regarding my remark on privacy, is it possible/easy to extend Rob-FCP in order to have differential privacy guarantees?

---

> ### Author Response · Authors · 2023-11-21
> **Author response to Reviewer eCJ4 [1/2]**
>
> > Q1: More clarifications of sending all the characterization vectors $v^{(k)}$ to the server in the federated setting. Is it possible to compute the vector distance with a federated algorithm?
>
> We added clarifications that the characterization vector $v^{(k)}$ is proposed exactly for the federated setting to reduce the privacy leakage of local scores and reduce communication costs. Concretely, in the federated conformal prediction context, each client $k$ computes the conformity scores of its $n_k$ local data samples $\\{s_j^{(k)}\\}$. However, sending all the conformity scores $\\{s_j^{(k)}\\}$ to the server is expensive and leaks much information about local data samples. Therefore, we characterize the scores $\\{s_j^{(k)}\\}$ with a histogram statistics vector $v^{(k)}$, which computes score frequency in $H$ equally partitioned subintervals. The characterization vector $v^{(k)}$ has a much smaller dimension than the universal scores $\\{s_j^{(k)}\\}$ ($H \ll n\_k$) and thus leaks much less private information during the communication between the server and clients.
> Thus, the characterization vector distance computation on the server side also obeys the principle of federated learning for privacy-preserving and communication efficiency.
> The characterization vector is an approximation of the universal scores and can induce an approximation error of federated calibration quantified by $\epsilon$ in Section 2.2. We also consider the influence of $\epsilon$ on the coverage guarantee in the Byzantine setting in Theorem 1 and Corollary 1, indicating a tradeoff between the tightness of conformal guarantee bound and the private information leakage/communication costs in federated settings.
>
> > Q2: More discussions of the factor of the minimal sample size of benign clients $1/\min n_i$.
>
> We thank you for the valuable comment. The appearance of $1/\min n_i$ is indeed due to the artifact of the proof. In the derivation from Equation (36) to Equation (37) in the proof of Theorem 1, we lower bound the total sample sizes in $\mathcal{B}$ (benign client set identified by Rob-FCP) with the minimal benign sample size $\min_{k’ \in [K_b]} n_{k’}$. We can definitely replace $\min_{k’ \in [K_b]} n_{k’}$ with the total sample sizes of $K_b-K_m$ benign clients with the smallest sample sizes, which leads to a tighter bound in many scenarios. This will introduce more complex notations, but it is only meaningful for some extreme cases. In practice, if a benign client has an extremely small sample size, then the characterization vector is almost useless for malicious client identification, and the influence on the conformal calibrated quantile is also negligible as FCP performs a weighted calibration based on sample sizes of clients. Therefore, for the extreme case of some benign clients with extremely small sample sizes (e.g., less than 10), it is more meaningful to discard their scores and apply Rob-FCP to the remaining clients, which is more beneficial to malicious client detection and only leads to negligible error for conformal calibration. We added the discussions to the remark of Theorem 1, which is similarly applied to Corollary 1.
>
> > Q3: Citations are not always appropriate.
>
> We follow the suggestions to fix citations of split conformal prediction in Section 2.1, citations of FCP in Sections 1,6, and citations of federated learning in Section 1.
>
> > Q4: More details of experiment settings.
>
> We included more details of used parameters in the main text for a better understanding in Section 5.1. We evaluate on three types of malicious data of Byzantine clients (i.e., Byzantine attacks): (1) *coverage attack* (CovAttack) with which malicious clients report the upper bound th conformity scores (e.g., $1$ for the mostly used LAC score) to induce a larger conformity score at the desired quantile and a lower coverage accordingly, (2) *efficiency attack* (EffAttack) with which malicious clients report the lower bound of the conformity scores (e.g., $0$ for the mostly used LAC score) to induce a lower conformity score at the quantile and a larger prediction set, and (3) *Gaussian Attack* (GauAttack) with which malicious clients inject random Gaussian noises with standard deviation $0.5$ to the scores to perturb the conformal calibration.
>
> We also included more details about the non-IID setting construction in the main text. we sample $p_{c,j} \sim \text{Dir}(\beta)$ and allocate a $p_{c,j}$ proportion of the instances of class $c$ to the client $j$. Here $\text{Dir}(·)$ denotes the Dirichlet distribution and $\beta$ is a concentration parameter ($\beta$ > 0). We select $\beta=0.5$ in evaluations without specification.
>
> > Q5: Notation errors.
>
> We thank you for pointing out the notation issues. We fixed the notation error of marginal coverage, $\mathcal{Q}_\lambda$, and $N_m$ in the revised manuscript.

---

> ### Author Response · Authors · 2023-11-21
> **Author response to Reviewer eCJ4 [2/2]**
>
> > Q6: In Theorem 1 and Corollary 1, $\epsilon$ appears in the bound. Is it possible to control it?
>
> We follow FCP to introduce the notation $\epsilon$ to quantify the approximation error of the estimated quantile $\hat{q}\_\alpha$ by data sketching such that the rank of $\hat{q}\_\alpha$ is between $(1-\alpha-\epsilon)(N+K)$ and $(1-\alpha+\epsilon)(N+K)$. The approximation error $\epsilon$ affects the coverage bound of FCP as Equation (3) and accordingly affects the coverage bound in the Byzantine setting as Theorem 1. Since $\epsilon$ is a quantity dependent on the score distribution, we cannot explicitly formulate it without further assumptions. However, we can control the approximation error $\epsilon$ with the dimension of the characterization vector $H$. A large $H$ indicates a large number of subintervals for the histogram statistics and a fine-grained partition, which will lead to a more precise approximation and a low $\epsilon$. Therefore, we can empirically adjust $H$ to control the approximation error $\epsilon$. We included the discussions in the remarks of Theorem 1 in the revised manuscript.
>
> *[1] Lu, Charles, et al. "Federated conformal predictors for distributed uncertainty quantification." ICML 2023.*
>
> > Q7: In the experiment how much time does it take to compute the quantile? (and for which $\epsilon$?)
>
> We follow the suggestion to add the runtime of quantile computation in Rob-FCP in the following Table 1, which indicates the efficiency of federated conformal prediction with our method.
>
> Table 1. Runtime of RobFCP quantile computation with $40\%$ malicious clients.
> | | MNIST | CIFAR-10 | Tiny-ImageNet | SHHS | PathMNIST |
> | - | - | - | - | - | - |
> | Runtime (seconds) | 0.5284 | 0.5169 | 0.5563 | 0.2227 | 0.3032 |
>
> > Q8: Is it possible/easy to extend Rob-FCP in order to have differential privacy guarantees?
>
> We thank you for the valuable question. To provide differential privacy guarantees of Rob-FCP, one practical approach is to add privacy-preserving noises to the characterization vectors before uploading them to the server. Essentially, we can view the characterization vector as the gradient in the setting of FL with differential privacy (DP) and add Gaussian noises to the characterization vector with differential privacy guarantees as a function of the scale of noises, which can be achieved by drawing analogy from the FL with DP setting [1,2,3]. Therefore, practically implementing the differential-private version of Rob-FCP is possible and straightforward.
>
> *[1] Zheng, Qinqing, et al. "Federated f-differential privacy." AISTATS 2021.*
>
> *[2] Andrew, Galen, et al. "Differentially private learning with adaptive clipping." NeurIPS 2021.*
>
> *[3] Zhang, Xinwei, et al. "Understanding clipping for federated learning: Convergence and client-level differential privacy." ICML 2022.*

---

> > ### Comment · Reviewer_eCJ4 · 2023-11-22
> >
> > I would like to thank the authors for their very detailed responses.
> >
> > -- Regarding the bound with $1/\min{n_i}$, if I understand correctly, it is possible to replace $1/\min{n_i}$ with $1/N_b$ with $N_b$ the number of data of all the benign clients? If this is true, I suggest at least mentioning the fact that it could be improved in thiw way. In FL, having a bound in the total number of data is desirable.
> >
> > -- "we can control the approximation error  with the dimension of the characterization vector"
> >
> > Is it possible to have theoretical bounds on the error?

---

> > > ### Author Response · Authors · 2023-11-22
> > > **Follow-up Discussion with Reviewer eCJ4**
> > >
> > > We thank you for the valuable comments and questions!
> > >
> > > > Q1: Can we reply $1/\min n_i$ with $1/N_b$, where $N_b$ is the number of data of all the benign clients?
> > >
> > > In the proof of Theorem 1, we bound $N\_{\mathcal{B}}$ (number of data samples in the index set $\mathcal{B}$ output by Rob-FCP) with a lower bound of the minimal sample size of benign clients $\min_{k' \in [K_b]} n_{k'}$. Let $K_b$ be the number of benign clients and $K_m~(K_m<K_b)$ be the number of malicious clients. Since we have $|\mathcal{B}|=K_b$, there are at least $K_b-K_m$ benign clients in set $\mathcal{B}$. Therefore, we can only replace the quantity $\min_{k' \in [K_b]} n_{k'}$ with the total sample sizes of $K_b-K_m$ benign clients with the smallest $K_b-K_m$ sample sizes. It is not possible to replace it with a summation of data samples for all benign clients. We made it more clear in the revised version.
> > >
> > > > Q2: Is it possible to have theoretical bounds on the approximation error?
> > >
> > > It is not feasible to have theoretical bounds of the error ($\epsilon$) without further assumptions on the specific distributions of the non-conformity scores. $\epsilon$ is the error of the rank of the estimate $\hat{q}\_\alpha$ with the characterization vectors on the calibration samples. The rank is a distribution-dependent quantity and is hard to analyze without a specification of the concrete distribution of the scores. We think it is interesting for future work to explore the rank approximation error bound in the FCP framework for different families of score distributions.

---

### Official Review · Reviewer_YYE4 · 2023-11-01

**Soundness:** 3 good
**Presentation:** 3 good
**Contribution:** 1 poor
**Rating:** 3
**Confidence:** 3

**Summary:**

The authors consider the conformal prediction interval construction in the federated learning setting where (1) each client reports conformality scores, and (2) a subset of clients are malicious and can report distorted scores to mess up with the interval construction. To alleviate the influence of malicious clients, the authors (1) identify malicious clients by comparing the score distributions from different clients under the assumption that conformity scores for benign clients are sampled from the same underlying distribution (IID setting), or from, essentially, distributions with bounded distance (non IID setting); (2) construct conformal prediction interval using the estimated benign clients.

**Strengths:**

The paper is overall easy to follow; and robust prediction intervals against malicious clients are an important question.

**Weaknesses:**

It seems to be that the assumptions considered seem over-simplified and may lead to less robustness and under-coverage of difficulty cases when violated: the entire paper is based on the assumption that benign clients have similar conformity score distributions, in both IID settings (identical) and non-IID settings (close), and a client whose conformity score is far from its K_b "neighbors" is claimed malicious.  However, in practice, benign clients can have data with different local characteristics, due to, e.g., demographic differences, differences in medical practice guidelines, etc.

**Questions:**

More details of the nonIID setting + different levels of being nonIID can be helpful.

---

> ### Author Response · Authors · 2023-11-21
> **Author response to Reviewer YYE4 [1/2]**
>
> > Q1: The assumptions considered are over-simplified and may lead to failure in difficult cases. More details of the nonIID setting and different levels of being nonIID can be helpful.
>
> We thank you for your valuable comments and suggestions. We added clarifications and improvements of Rob-FCP for the non-IID setting from both the theoretical perspective and the evaluation perspective.
>
> From the perspective of theoretical analysis, we provide the coverage guarantees of Rob-FCP in the non-IID setting in Corollary 1 based on the assumption of bounded disparity among benign clients in Assumption 3.1. We added clarifications that the assumption is quite standard, typical, and not simplified across the literature of non-IID federated learning analysis [1] and Byzantine analysis [2,3]. Concretely, Li et al. [1] quantify the non-iid degree (i.e., disparity) of local loss of clients with $\Gamma$ in Section 3.1 of [1] and accordingly provide the convergence guarantee of federated optimization in Theorem 1 of [1], which is a function of the disparity $\Gamma$. Park et al. [2] assume a bounded difference $\rho_1$ between the local loss of benign clients and global loss in Assumption 2 of [2] and provide the convergence analysis in federated Byzantine optimization in Theorem 1 of [2] as a function of $\rho_1$. Data et al. [3] have an assumption of bounded gradient dissimilarity $\kappa$ in Assumption 2 of [3] and prove a convergence guarantee in Theorem 1 of [3] regarding $\kappa$. Similarly, in our paper, we assume a bounded disparity of score statistics quantified with $\sigma$ in Assumption 3.1 and accordingly provide the coverage guarantees of Rob-FCP in the non-IID setting in Corollary 1 as a function of $\sigma$. As we can see, the type of assumption is quite standard in the analysis setup of the line of non-IID analysis and Byzantine analysis. The assumption is also essential since without a quantity to restrict the disparity in local behavior, the local behavior can vary arbitrarily, and we can not control global performance quantitatively. From the results in Corollary 1, we can also see that as long as the disparity bound $\sigma$ is finite, the coverage bound in the non-IID setting is asymptotically valid and tight with sufficiently large benign sample sizes $n_b$. We added these clarifications and details in Section 3.3 in the revised manuscript.
>
> From the evaluation perspective, we added more clarifications and illustrations on how we construct non-IID data of different levels and also added more evaluations on various approaches to non-IID data construction. We first added clarifications that we followed the standard evaluation setup of non-IID federated learning by sampling different label ratios for different clients from the Dirichlet distribution as the literature [4,5,6,7]. Concretely, we sample $p_{c,j} \sim \text{Dir}(\beta)$ and allocate a $p_{c,j}$ proportion of the instances with class $c$ to the client $j$. Here $\text{Dir}(·)$ denotes the Dirichlet distribution and $\beta$ is a concentration parameter ($\beta$ > 0). Dirichlet distribution is commonly used as the prior distribution in Bayesian statistics and is an appropriate
> choice to simulate real-world data distribution. An advantage of this approach is that we can flexibly change the imbalance level by varying the concentration parameter $\beta$. If $\beta$
> is set to a smaller value, then the partition is more unbalanced. We fixed the imbalance level $\beta$ as $0.5$ in the evaluations before the rebuttal period. Following the suggestion, we added evaluations with different imbalance levels $\beta$ to show the effectiveness of Rob-FCP under different non-IID settings. The results in the following Table 1 demonstrate that under multiple degrees of data imbalance in non-IID federated conformal prediction, Rob-FCP consistently outperforms FCP in achieving a nominal marginal coverage level in the existence of Byzantine clients. We also consider alternative approaches to construct non-IID data with demographic differences in federated learning. Concretely, we split the SHHS dataset by sorting five different attributes: wake time, N1, N2, N3, REM. For example, we first sort the instances according to the wake time of patients and then assign them to 100 clients in sequence. Therefore, some clients are assigned patients with an early wake time, and some are assigned instances with late wake time. The results in Table 2 show that in this type of non-IID partition, Rob-FCP still demonstrates robustness and effectiveness compared to FCP.

---

> ### Author Response · Authors · 2023-11-21
> **Author response to Reviewer YYE4 [2/2]**
>
> > Q1 (to continue):  The assumptions considered are over-simplified and may lead to failure in difficult cases. More details of the nonIID setting and different levels of being nonIID can be helpful.
>
> Table 1. Marginal coverage / average set size with different non-IID imbalance level $\beta$, the parameter of Dirichlet distribution where the label ratios of clients are sampled from. The evaluation is done under different Coverage attack with $40\%~(K_m/K=40\%)$ malicious clients. The desired marginal coverage is $0.9$.
>
> | | $\beta=0.1$ | $\beta=0.3$ | $\beta=0.5$ | $\beta=0.7$ | $\beta=0.9$ |
> | - | - | - | - | - | - |
> | FCP on MNIST | 0.780 / 1.173 |  0.817 / 1.318 | 0.833 / 1.384 | 0.805 / 1.265 |  0.828 / 1.363 |
> | Rob-FCP on MNIST | **0.899** / 1.806 | **0.905** / 1.809 | **0.903** / 1.827 |  **0.898** / 1.781 | **0.893** / 1.768 |
> | FCP on CIFAR-10 | 0.806 / 1.641 | 0.821 / 1.717 | 0.836 / 1.791 | 0.823 / 1.744 | 0.824 / 1.723 |
> | Rob-FCP on CIFAR-10 | **0.899** / 2.260 |  **0.907** / 2.405 | **0.892** / 2.243 | **0.904** / 2.396 | **0.910** / 2.416 |
> | FCP on Tiny-ImageNet | 0.840 / 28.625 | 0.830 / 28.192 |  0.833 / 28.340 |  0.821 / 27.140 | 0.8308 / 28.751 |
> | Rob-FCP on Tiny-ImageNet | **0.913** / 45.872 | **0.910** / 44.972 | **0.898** / 42.571 | **0.887** / 41.219 | **0.898** / 43.298 |
> | FCP on PathMNIST | 0.850 / 1.106 | 0.839 / 1.065 | 0.837 / 1.055 | 0.839 / 1.065	 | 0.832 / 1.043 |
> | Rob-FCP on PathMNIST | **0.895** / 1.311 | **0.900** / 1.355 | **0.900** / 1.355 | **0.899** / 1.354 | **0.901** / 1.363 |
>
>
> Table 2. Marginal coverage / average set size on SHHS with non-IID data partition based on different attributes: wake time, N1, N2, N3, REM. The evaluation is done under different Coverage attack with $40\%~(K_m/K=40\%)$ malicious clients. The desired marginal coverage is $0.9$.
>
> | | wake time | N1 | N2 | N3 | REM |
> | - | - | - | - | - | - |
> | FCP on SHHS | 0.835 / 1.098 | 0.841 / 1.104 | 0.841 / 1.104 | 0.837 / 1.105 | 0.840 / 1.107 |
> | Rob-FCP on SHHS | **0.901** / 1.367 | **0.902** / 1.358 | **0.902** / 1.355 | **0.902** / 1.375 | **0.900** / 1.356 |
>
> *[1] Li, Xiang, et al. "On the convergence of fedavg on non-iid data." ICLR 2020.*
>
> *[2] Park, Jungwuk, et al. "Sageflow: Robust federated learning against both stragglers and adversaries." NeurIPS 2021.*
>
> *[3] Data, Deepesh, et al. "Byzantine-resilient high-dimensional SGD with local iterations on heterogeneous data." ICML 2021.*
>
> *[4] Yurochkin, Mikhail, et al. "Bayesian nonparametric federated learning of neural networks." International conference on machine learning. ICML 2019.*
>
> *[5] Lin, Tao, et al. "Ensemble distillation for robust model fusion in federated learning." NeurIPS 2020.*
>
> *[6] Wang, Hongyi, et al. "Federated learning with matched averaging." ICLR 2020.*
>
> *[7] Gao, Liang, et al. "Feddc: Federated learning with non-iid data via local drift decoupling and correction." CVPR 2022.*

---

> ### Comment · Reviewer_YYE4 · 2023-11-23
>
> I am sorry but I feel this is a very dangerous assumption, and leads to increased prediction disparity when leaving out minority samples that are not intentional attack. What makes it really dangerous is that this decision is  based on prediction quality, as a result, there is no way that you can even assess which samples during test are belonging to the left-out minority populations.
>
> I am willing to increase my score if the authors can explain to me why this is not a concern in the settings considered.

---

> > ### Author Response · Authors · 2023-11-23
> > **Follow-up discussions with Reviewer YYE4**
> >
> > Thank you for the valuable response and feedback! We can actually think about the effectiveness of Rob-FCP in the non-IID setting from two perspectives. (1) If the malicious clients attempt to disturb the quantile value of conformity scores to a large extent, they need to report a score vector far away from the cluster of score vectors of benign clients. Although the benign score vectors have some heterogeneity, the malicious score vector in this case is still distinguishable in the vector space and effectively excluded during the calibration. (2) If the malicious clients report score vectors close to the cluster of benign score vectors, then even if the Rob-FCP algorithm does not identify malicious clients exactly, the malicious score vectors can also characterize the structure of the benign score cluster, and thus, the computed quantile value would not deviate much. The two situations are exactly the important intuitions in the proof of Theorem 1 and Corollary 1, which rigorously formulates the coverage guarantee of Rob-FCP as a function of the non-IID degree $\sigma$. We also would like to emphasize that the non-IID assumption and setting are standard in the literature of Byzantine robustness [1,2,3].
> >
> > *[1] Li, Xiang, et al. "On the convergence of fedavg on non-iid data." ICLR 2020.*
> >
> > *[2] Park, Jungwuk, et al. "Sageflow: Robust federated learning against both stragglers and adversaries." NeurIPS 2021.*
> >
> > *[3] Data, Deepesh, et al. "Byzantine-resilient high-dimensional SGD with local iterations on heterogeneous data." ICML 2021.*

---

### Author Response · Authors · 2023-11-21
**Revision Summary**

We thank all the reviewers for their valuable comments and feedback. We are glad that the reviewers find our work novel with sound theoretical and empirical results. Based on the reviews, we have made the following updates to further improve our work. All the updates were already incorporated into the revised manuscript with a highlighted blue color.

1. We added clarifications and improvements of Rob-FCP for the non-IID setting from both the theoretical perspective and the evaluation perspective, following the suggestion of Reviewer YYE4,  bwYM, and LS6w.

2. We added more discussions of certain factors in the theorems, following the suggestion of Reviewer eCJ4.

3. We added clarifications of the rationale of Rob-FCP to the federated learning setting, following suggestions of Reviewer eCJ4, and bwYM.

4. We added evaluations of the runtime of Rob-FCP, following suggestions of Reviewer eCJ4.

5. We provided certification results with more advanced concentration bounds, following the suggestions of Reviewer bwYM.

6. We included more discussions of the effectiveness of Rob-FCP in the non-IID settings, following the suggestions of Reviewer bwYM, and LS6w.

7. We theoretically and empirically show Rob-FCP with an overestimate or underestimate of the number of benign clients, following the suggestions of Reviewer LS6w.

---

### Meta-Review · Area_Chair_F4ak · 2023-12-18

**Metareview:**

The paper is about conformal prediction in a distributed setting where adversarial users could potentially be present. An algorithm, called Rob-FCP, is proposed to execute robust federated conformal prediction, effectively countering malicious clients capable of reporting arbitrary statistics during the conformal calibration process.

I thank the authors for their thorough responses. The reviewers had raised several concerns, some alleviated by the responses, but some others have remained. I encourage the authors to incorporate the comments from the reviewers (the main assumptions, some details of the algorihms and the proofs, etc -- please see the reviews and discussions afterwards). The paper will then be an interesting contribution to the CP+FL communities.

**Justification For Why Not Higher Score:**

I recommended rejecting -- the decision is based on the reviews and the discussions afterwards.

**Justification For Why Not Lower Score:**

--

---

### Decision · Program_Chairs · 2024-01-16

Reject